# RL MAKES MLLMS SEE BETTER THAN SFT

**Junha Song[1,2], Sangdoo Yun[2], Dongyoon Han[2], Jaegul Choo[1], Byeongho Heo[2†]**

[1]KAIST, [2]NAVER AI Lab
[†]Corresponding author

## ABSTRACT

A dominant assumption in Multimodal Language Model (MLLM) research is that its performance is largely inherited from the LLM backbone, given its immense parameter scale and remarkable capabilities. This has created a void in the understanding of the vision encoder, which determines '*how MLLMs perceive images*'. The recent shift in MLLM training paradigms, from Supervised Finetuning (SFT) to Reinforcement Learning (RL), magnifies this oversight—namely, the significant lack of analysis on how such training reshapes the vision encoder as well as the MLLM. To address this, we first investigate the impact of training strategies on MLLMs, where RL shows a clear advantage over SFT in strongly vision-related VQA benchmarks. Motivated by this, we conduct a critical yet under-explored analysis of the vision encoder of MLLMs through diverse and in-depth experiments, ranging from ImageNet classification and segmentation to gradient visualization. Our results demonstrate that MLLM's post-training strategy (*i.e.*, SFT or RL) not only leads to distinct outcomes on MLLM downstream tasks, but also fundamentally reshapes MLLM's underlying visual representations. Specifically, our main finding is that **RL produces stronger and more localized visual representations compared to SFT, boosting the ability of the vision encoder for MLLM.** We then reframe our findings into a simple recipe for building strong vision encoders for MLLMs, Preference-Instructed Vision OpTimization (`PIVOT`). When integrated into MLLMs, a `PIVOT`-trained vision encoder outperforms even larger and more heavily-trained counterparts, despite requiring less than 1% of the computational cost of standard vision pretraining. This result opens an effective and efficient path for advancing the vision backbones of MLLMs.

## 1 INTRODUCTION

Human knowledge is acquired through multiple sensory experiences, with vision playing a dominant role in understanding the environment and accumulating knowledge, beyond finding food and avoiding predators (Piaget et al. 1952; Tong et al. 2024a). Inspired by this principle, recent advances in Large Language Models (LLMs) (Dubey et al. 2024; Yang et al. 2025b; Brown et al. 2020) naturally extend toward Multimodal LLMs (MLLMs) (Achiam et al. 2023; Team et al. 2023; 2024a). Especially, large vision language models[1] have been recently and preferentially investigated as a pathway to foster visual intelligence in LLMs (Liu et al. 2023a; Li et al. 2025a; Chen et al. 2024).

The combination of independently pretrained LLMs and vision models enabled MLLMs to reach strong initial capabilities (Mokady et al. 2021; Li et al. 2023a). Further advances have been driven by larger and stronger architectures, along with higher-quality datasets, as shown in LLaVA (Liu et al. 2024a; Li et al. 2025a), QwenVL (Bai et al. 2023b), and DINO-MLLM (Fan et al. 2025). Building on this, current research now seeks improvements via reinforcement learning (RL), moving beyond the standard supervised finetuning (SFT), paralleling the shift that RL brought to LLMs (Christiano et al. 2017; Ouyang et al. 2022). For instance, several studies demonstrate that incorporating human preference data via RL enhances MLLM performance (Sun et al. 2024a; Wang et al. 2024b) and mitigates hallucination (Yang et al. 2025c; Yu et al. 2024; Fu et al. 2025b). Other research has expanded the scope of RL to include contrastive image pairs (Wang et al. 2024a; Fu et al. 2025a).

Despite the efficacy of RL in the **MLLM**, a comprehensive understanding of its effects compared to SFT—and critically, its influence on the **vision encoder**—remains largely absent from the literature.

---

[1]Following recent works (Tong et al. 2024a;b; Fan et al. 2025), we refer to LLMs with visual capabilities as MLLMs.

Specifically, the field lacks a systematic comparison within MLLMs between SFT for instruction-following and RL for preference alignment, including an analysis of model scaling in common benchmarks. The lack of understanding is especially notable for another under-investigated dimension: the vision encoder. Indeed, research has progressed little beyond the preliminary finding that fine-tuning the vision encoder (Tong et al. 2024a; Li et al. 2024b) yields better outcomes than keeping it frozen (Liu et al. 2023a; Li et al. 2023a; Driess et al. 2023; Karamcheti et al. 2024). Such oversight can be attributed to an implicit, LLM-centric assumption about the source of MLLM capabilities, leaving a significant void in our understanding of how SFT and RL differ in reshaping visual representations.

We present a timely exploration of both the MLLM and its vision encoder under different training strategies. We focus our RL analysis on Direct Preference Optimization (DPO) for simplicity, which is a common recipe for recent MLLMs (Yu et al. 2024; Yang et al. 2025c; Fu et al. 2025a). We begin with a fundamental analysis in Section 3, comparing the effects of SFT and RL on MLLMs across broad vision-language (VL) benchmarks. Our analysis reveals that RL yields significant gains on vision-

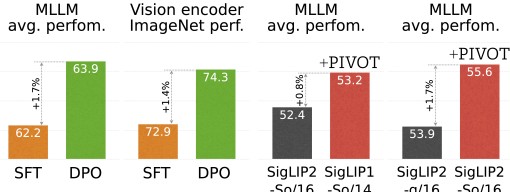

Figure 1: **TD;LR.** We study how SFT and RL (*e.g.*, DPO) affect not only MLLMs but also their vision encoders, and formulate a simple recipe, PIVOT, for evolving vision models for use in MLLM.

centric tasks, a finding that motivates a deeper investigation into the vision encoder itself. Subsequently, in Section 4, we conduct a unique and critical analysis of the vision encoder, providing key insights for the visual encoder development. Our results reveal that MLLM post-training rewrites the visual representations, with RL driving stronger representation than SFT. The finding is supported by gradient visualizations that trace how optimization signals propagate to the vision encoder.

The foregoing analysis establishes that RL reshapes visual representations, motivating a critical question we explore in Section 5: *Can RL-trained models surpass SOTA vision models for MLLM?*. To this end, we re-formalize RL training as an auxiliary training process for vision encoder, termed Preference-Instructed Vision OpTimization (PIVOT), and evaluate its efficacy on diverse encoders, including CLIP (Radford et al. 2021), DINO (Oquab et al. 2024), and MAE (He et al. 2022). The results reveal a remarkable impact of PIVOT when the enhanced encoders are used within MLLMs; a vision model trained with PIVOT not only outperforms its original counterpart but also surpasses a substantially larger model (*e.g.*, SigLIP2-So/16 + PIVOT > SigLIP2-g/16) and even a subsequent-generation encoder (*e.g.*, SigLIP1-So/14[2] + PIVOT > SigLIP2-So/16). Notably, this enhancement is achieved with just 18 hours of training on 8 H100 GPUs using a Qwen2.5-1.5B LLM-head. This amounts to fewer than **1%** of GPUs of standard vision pre-training, with SigLIP2 trained on up to 2K TPUv5e chips. Taken together, the evidence indicates that even state-of-the-art encoders have substantial room for MLLM evolution, and PIVOT is a promising direction for future exploration.

## 2  MLLMs ON RL: WHERE DO WE STAND?

The initial paradigm for training LLMs involves auto-regressive pre-training followed by SFT to promote instruction-following capabilities (Radford et al. 2018; Dai et al. 2019; Yang et al. 2019; Brown et al. 2020). A subsequent breakthrough occurs with Reinforcement Learning from Human Feedback (RLHF), which demonstrates that utilizing RL to align LLM outputs with human preferences enables chat-oriented LLMs (Christiano et al. 2017; Ouyang et al. 2022; Touvron et al. 2023b). The use of RL has become a cornerstone of modern LLM development, with advanced methods like DPO (Rafailov et al. 2023) and GRPO (Shao et al. 2024) being widely implemented in recent models such as LLaMA-3 (Dubey et al. 2024) and Qwen-2.5 (Yang et al. 2025a).

MLLMs have adopted the LLM training advances to leverage prior experiences. Early MLLMs such as LLaVA-Next (Li et al. 2024b) and Cambrian (Tong et al. 2024a) combine a pre-trained LLM with a pre-trained vision model, then align the LLM to vision representation through **SFT** on vision-language data like captioning and visual question answering. Recent works, as summarized in Table A, demonstrated that applying **RL** as an auxiliary process can further boost MLLM's downstream performance (Yu et al. 2025; Wang et al. 2024b; Sun et al. 2024a). Other studies have proposed advanced DPO variants for multimodal contexts, for instance by incorporating visual preference data (Fu et al. 2025a; Wang et al. 2024a) or modifying the objective to mitigate hallucinations (Yu et al. 2024; Yang et al. 2025c). Further studies highlight RL's advantages over SFT in adapting an

---

[2] We use SigLIP1-So/14, as the weights for SigLIP1-So/16 are not publicly available.

MLLM's knowledge to specialized environments, such as map navigation (Chu et al. 2025) and robot action planning (Li et al. 2025b).

These studies reveal a clear trend in the application of RLHF to MLLMs. They rely on RL using either PPO (Sun et al. 2024a) or DPO, with the predominant choice becoming DPO (Yu et al. 2024; 2025; Wang et al. 2024a; Yang et al. 2025c; Fu et al. 2025a), as shown in Table A. In line with this trend, our main paper uses DPO as the primary RL representative for a controlled comparison with SFT. Results for other RL algorithms (PPO, GRPO (Shao et al. 2024)) and DPO variants (Wang et al. 2024b) are also presented in Section B.1.

# 3    HOW DO SFT AND RL AFFECT MLLMS?

Despite the advances of RL described in Section 2, existing studies lack a comprehensive analysis, offering limited insight and intuition into following questions: *How do SFT and DPO affect MLLM on diverse VQA tasks?*, *Is DPO actually superior to SFT?*, And *does this trend hold with model scaling?* To address them, we establish a controlled training setup and conduct a deep investigation.

## 3.1    EXPERIMENTAL SETUP & PREREQUISITE

**Model scaling.** The standard MLLM architecture, which integrates an LLM with a vision encoder via a multimodal projector, has proven effective, achieving superior performance on VL tasks (Liu et al. 2024a; Lei et al. 2025; Shukor et al. 2025). Our model is implemented using the popular open-source MLLM repository, LLaVA-OneVision (Li et al. 2025a). Following their setup, we conduct a study across various cases by adopting four scales of the Qwen2.5 LLM (0.5B, 1.5B, 3B, 7B) (Yang et al. 2025a) and four SigLIP2 384px sizes (B/16, L/16, So/16, g/16) (Tschannen et al. 2025), with a 2-layer MLP serving as the projector.

**Training procedure.** Our MLLM development process consists of two stages: *Stage 1* pre-training and *Stage 2* post-training. In *Stage 1*, we first align the visual and language embedding spaces by conducting multimodal projector-only training. And then, a base MLLM is established by training all model parameters on diverse VL datasets, including Visual Question Answering (VQA), vision-grounded dialogue, and image captioning (Li et al. 2025a). *Stage 2* indicates post-training, which involves a full-parameter update of the base model according to SFT or DPO, detailed below. Further details, including hyperparameters, are included in Section E.1 and the source code[3].

**Post-training strategies.** Our analysis compares two post-training approaches: SFT and DPO. Prior works like MPO (Wang et al. 2024b) typically focus on comparing a pre-trained model (*Stage 1*) against the same model further trained with DPO, which does not provide a fair evaluation of DPO versus SFT. On the other hand, we conduct a controlled comparison in *Stage 2*, using the *same* number of 'image-query-response' pairs across the two algorithms. Specifically, we define the post-training dataset as $X_{\text{PT}} = \{x_0, x_1, \ldots, x_T\}$, with each element $x_i = \{I_i, q_i, y_i^c, y_i^r\}$ representing an image $I_i$, a query $q_i$, and the corresponding chosen and rejected responses $y_i^c$ and $y_i^r$. The optimization objectives using this dataset is defined as follows:

$$L_{\text{SFT}} = -\mathbb{E}_{i \sim X_{\text{PT}}} \log \pi_\theta(y_i^c \mid I_i, q_i); \; L_{\text{DPO}} = -\mathbb{E}_{i \sim X_{\text{PT}}} \log \sigma \left( \beta \left( \log \frac{\pi_\theta(y_i^c|I_i,q_i)}{\pi_{\text{ref}}(y_i^c|I_i,q_i)} - \log \frac{\pi_\theta(y_i^r|I_i,q_i)}{\pi_{\text{ref}}(y_i^r|I_i,q_i)} \right) \right), \quad (1)$$

where $\pi_\theta$ represents the MLLM; $\pi_{\text{ref}}$ is the reference model; and $\beta$ is the temperature controlling the strength of preference alignment. In short, we compare SFT (*Stage 2*) with DPO (*Stage 2*) with the same number of training samples. A more detailed description is given in Section D.1.

**Data & Evaluation.** To ensure reproducibility, we utilize publicly available datasets provided in the LLaVA-OneVision and MPO repositories. To be more specific, in *Stage 1*, we apply projector-only pre-training on the LAION/CC/SBU-558K dataset (Liu et al. 2024a) and perform end-to-end pre-training on the LLaVA-OneVision-3.2M dataset (Li et al. 2025a). As the post-training dataset $X_{\text{pt}}$ in *Stage 2*, we utilize the MPO (Wang et al. 2024b) data and randomly sample 20K instances, a scale comparable to recent DPO studies for MLLMs (Yu et al. 2024; Yang et al. 2025c). It is worth noting that this two-stage strategy and the proportion of training data closely resemble the training paradigm of LLMs such as InstructGPT (Ouyang et al. 2022), where RLHF is applied after instruction-following pre-training. For evaluation, we adapt the benchmark suite introduced in Cambrian, which covers 16 tasks across four categories of VQA: General, Knowledge, OCR & Chart, and Vision-Centric. This provides a broader and more common comparison than prior studies that mainly focus only on vision (Yang et al. 2025c; Fu et al. 2025b) or specialized tasks (Chu et al. 2025).

---

[3]https://github.com/junha1125/PIVOT

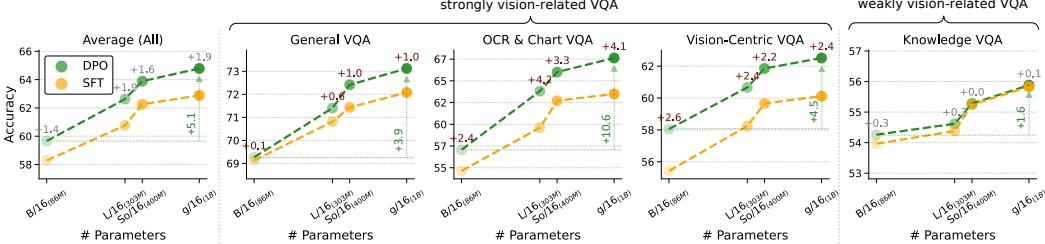

Figure 2: **Scaling the vision encoder in MLLMs.** We analyze the impact of the vision encoder sizes, ranging from 86M (B/16) to 1B (g/16) parameters, in Qwen2.5-3B combined with SigLIP2 on vision–language benchmarks. Interestingly, DPO yields particularly stronger gains over SFT in vision-intensive VQA.

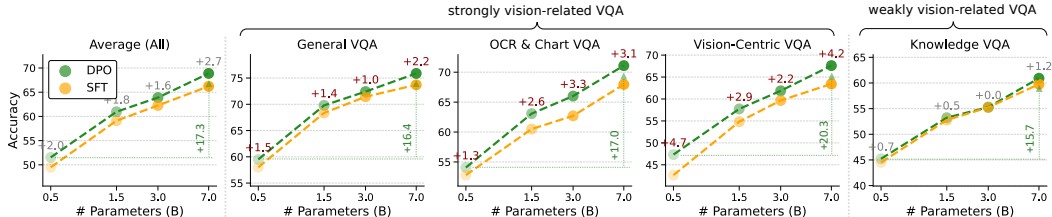

Figure 3: **Scaling the language model in MLLMs.** Using SigLIP2-So/16 as the vision encoder, we vary the language model size (Qwen2.5) and evaluate performance. Consistent with Figure 2, DPO substantially outperforms SFT on vision-related tasks, while they show comparable results in the Knowledge VQA.

## 3.2 ANALYSIS AND FINDINGS

We compare the performance of MLLMs trained with two post-training approaches across different model scales. First, Figure 2 reports results as the vision model, SigLIP2, scales from 86M to 1B, with the language model fixed to Qwen2.5-3B. Next, Figure 3 shows performances as the language model size increases from 0.5B to 7B, while keeping the vision encoder fixed to SigLIP2-So/16.

Before comparing SFT and DPO, we analyze the impacts of model scaling on MLLM benchmarks. As shown in Figure 2, the performance improves with the size of the vision encoder, confirming the importance of the visual representation capacity within MLLMs. Replacing SigLIP2-B/16 with SigLIP2-g/16 encoder yields significantly better performance on strongly vision-related tasks. For the DPO-tune MLLM, the gap between the B/16 and g/16 models reaches +4.5%p in Vision-Centric and strikingly +10.6%p in OCR & Chart VQA. In contrast, the improvement is relatively minor at +1.9%p in the weakly vision-related task, Knowledge VQA. These results show that the vision model plays a crucial role in vision-related tasks, even though the language model scaling in Figure 3 exhibits a large performance gap.

> ***Finding 1:*** Increasing the capacity of the vision encoder in MLLMs is particularly important for tasks requiring fine-grained visual understanding.

A central focus of our analysis is the comparative efficacy of DPO and SFT for MLLM post-training. The results in Figure 2 show that DPO achieves a superior performance compared to SFT, particularly on tasks that require deep visual comprehension rather than those primarily relying on the LLM's knowledge. For instance, on Knowledge VQA benchmarks such as ScienceQA (Lu et al. 2022) and MathVista (Lu et al. 2023), where models rely on scientific or mathematical backgrounds in LLMs, the improvement is only marginal (*e.g.*, +0.3%p). On the other hand, DPO's superiority becomes evident in strongly vision-related benchmarks like ChartQA (Masry et al. 2022), DocVQA (Mathew et al. 2021), MMVP (Tong et al. 2024b), and CV-bench (Tong et al. 2024a). Quantitatively, with the SigLIP2-L/16, DPO builds a model with +4.2%p and +2.4%p higher performance on OCR & Chart VQA and Vision-Centric VQA, respectively.

The trend of DPO's superiority holds firm even when scaling the language model, as shown in Figure 3. Even as the language model's size increases, the DPO-tuned MLLM consistently surpasses the SFT model, maintaining significant gaps of +3.1%p in OCR & Chart VQA and +4.2%p in Vision-Centric VQA with SigLIP2-g/16. It highlights the superiority of DPO, particularly on tasks requiring detailed visual understanding, and further implies that preference alignment impacts the model's

visual processing capabilities, beyond the language model. This observation motivates an in-depth analysis of visual representation in MLLMs.

> **Finding 2:** Preference alignment (DPO) produces MLLMs with superior performance to SFT, especially on strongly vision-related tasks.

As a final analysis, we investigate the effect of data scaling on the *Stage 2* post-training. The training data is scaled from 3K to 40K, whereas the model sizes are fixed to Qwen2.5-1.5B and SigLIP2-So/16. The results are shown in Figure 4. While SFT's performance improves gradually with more data, DPO achieves high performance rapidly, even with a small number of samples. We also observe that a DPO-trained model outperforms an SFT-trained counterpart even with a data disadvantage. For example, DPO with 3K samples achieves a score of 60.4%p, surpassing the 59.5%p score of an SFT model trained on 40K samples. Additional results, including performance on distinct domains, are in Section C.1.

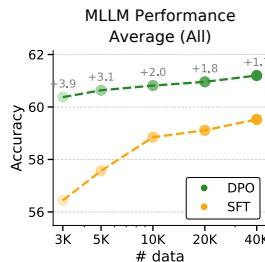

Figure 4: **Impact of data scales** on MLLM tasks.

## 4 HOW DOES MLLM TRAINING AFFECT VISUAL REPRESENTATIONS?

The previous section demonstrates DPO's superiority over SFT on MLLM benchmarks, with impressive gains on vision-related tasks. The finding suggests that DPO impacts not only the language module but the model's visual processing capabilities. Several studies have investigated the vision encoder in MLLMs, focusing primarily on architectural adjustments such as enabling vision encoder updates (Bai et al. 2025; Li et al. 2024b), applying all image grids (Li et al. 2025a; Marafioti et al. 2025), and utilizing multiple vision encoders (Tong et al. 2024b;a). In this section, we move beyond these approaches to conduct a deeper analysis of the vision encoder within MLLMs. To the best of our knowledge, this is the first work to conduct an in-depth analysis of the vision encoder in MLLMs.

### 4.1 EXPERIMENTAL SETUP

We begin with the MLLMs used in Section 3, which are trained with *Stage 1* pre-training and either SFT or DPO *Stage 2* post-training. After separating the vision components from the MLLM (*i.e.*, detach the vision encoder and projector), we assess their standalone performance on classic vision tasks, including ImageNet classification and semantic segmentation. Performance is measured using image features generated from the vision encoder, or from the combined encoder-projector. In this analysis, we disentangle the impact on the visual representations by isolating the vision encoder from the LLM. More details are available in Section E and the source code.

### 4.2 EVALUATING VISION ENCODERS BEYOND MLLM BENCHMARKS

**ImageNet Classification.** We conduct model scaling experiments on ImageNet classification, performing a linear-probe evaluation with the features extracted from the visual components. Note that the features are originally used as the visual token inputs in the MLLM. As shown in Figure 6, our investigation highlights the following key points. (*i*) The MLLM post-training actually reshapes the visual representations. (*ii*) DPO consistently outperforms SFT in the vision-only benchmark. DPO outperforms SFT in ImageNet Top-1 accuracy by +1.83%p for SigLIP2-So/16 coupled with a Qwen-3B head, and by +1.96%p for SigLIP2-L/16 with a Qwen-1.5B head. We claim this as a novel finding: DPO—a preva-

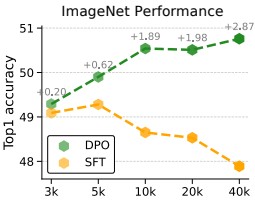

Figure 5: **Impact of data scales** on ImageNet.

lent RL method in the LLM community (Yang et al. 2025b; Dubey et al. 2024)—is more effective than SFT, not only for **aligning LLMs** but also for **learning visual representations**. (*iii*) MLLM training with larger LLMs yields a high-performing vision encoder. For instance, when trained on DPO, the SigLIP2-So/16 coupled with a 7B LLM exhibits a +4.4%p increase in ImageNet accuracy compared to when coupled with a 0.5B LLM. It supports the hypothesis that larger-capacity LLMs provide more informative optimization signals to the vision encoder.

Additionally, we investigate how the data scale of *Stage 2* post-training affects visual representations, using the MLLM architecture described in Section 3 (Qwen2.5-1.5B and SigLIP2-So/16). The results in Figure 5 show a notable difference from those observed in Figure 4. While performance on MLLM

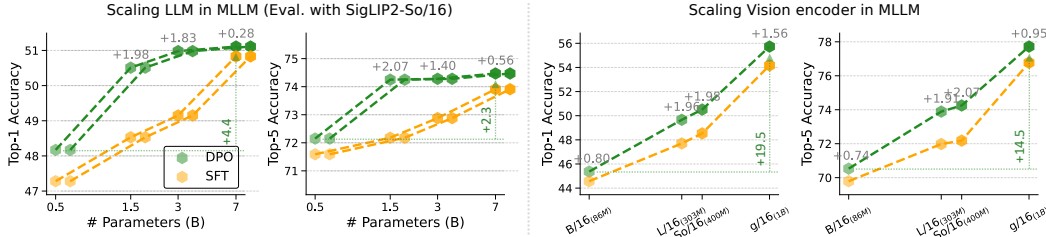

Figure 6: **ImageNet accuracy of vision encoder.** MLLM post-training is conducted with either SFT or DPO, then the vision encoder is detached from LLM and its vision-only performance evaluated via linear probing. We scale the LLM with a fixed SigLIP2-So/16 (left), or the vision encoder with a fixed Qwen2.5-1.5B (right).

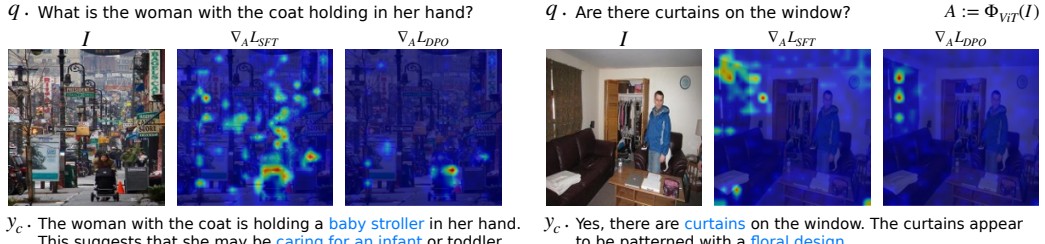

Figure 7: **Gradient visualization.** Using Grad-CAM (Selvaraju et al. 2017), we visualize the gradient signals received by the vision encoder features ($A := \Phi_{VE}(I)$) under MLLM post-training strategies. We observe that the gradient signals from DPO align more strongly with question-relevant regions than those from SFT.

benchmarks improves for both SFT and DPO with more data, only DPO benefits from data scaling in the quality of visual representation. This finding suggests that the choice of MLLM training strategy fundamentally alters *how the model sees an image*.

> **Finding 3:** MLLM training not only adapts the language model but also reshapes the visual representations that determine how the model sees an image.

**Gradient Visualization.** To understand DPO's effectiveness on vision, we investigate how DPO differs in the gradient signals to the vision encoder compared to SFT in the post-training stage. We use Grad-CAM (Selvaraju et al. 2017): we compute the loss for a specific sample $x_i$ as defined in Equation (1) and perform a backward pass with the sample loss. During the backward pass, we obtain the gradients with respect to the feature activations of the vision encoder, measure the gradient magnitude of each token, and visualize the results. Interestingly, as shown in Figure 7, large gradients primarily occur in question-relevant regions, supporting **Finding 3**. Moreover, the SFT signal tends to be scattered, while the signal from DPO is precisely focused on semantically relevant regions. We hypothesize that the contrastive nature of the DPO objective enables fine-grained gradient signals for the visual representations when differentiating between chosen and rejected responses. Additional results are available in Section C.4.

**Image Segmentation.** Assuming that DPO enhances the fine-grained training of visual representations, we expect it to be connected with improved localization ability. To measure the localization ability, we perform segmentation probing evaluation with the ADE20K (Zhou et al. 2017) dataset, following the protocol of Covert et al. (2025). First, we utilize MLLM-tuned vision encoders from Section 3. Then, we freeze the vision encoder and attach a two-layer MLP, training it as a patch-level classifier for segmentation. We utilize various vision encoders, based on CLIP (Radford et al. 2021), SigLIP1 (Zhai et al. 2023), and SigLIP2 (Tschannen et al. 2025), all of which are tuned with either SFT or DPO using a Qwen-1.5B LLM. The results in Figure 8 show that the MLLM-tuned vision encoder with DPO consistently outperforms those with SFT on segmentation task; for example, DPO-tuned yields a 1.08%p increase in patch-level recall when using a CLIP-L/14 336px encoder. The superiority of DPO is also supported by the qualitative results in Figure 9 and Figure G, showing DPO-tuned vision encoders generate segmentation maps with closer alignment with the ground truth.

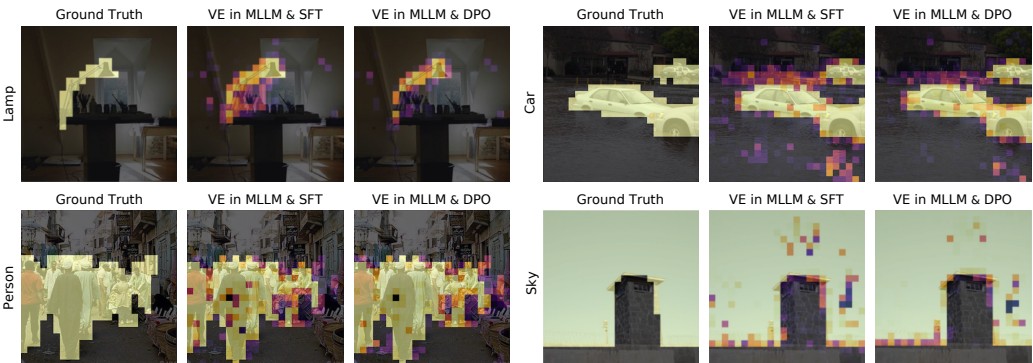

**Figure 8: Segmentation probing results.** We evaluate segmentation performance via two-layer MLP probing across 6 encoders, each MLLM-trained with a Qwen2.5-1.5B LLM head. The y-axis shows the mean patch-level recall over six random seeds. DPO consistently outperforms over SFT, with the gain shown above the DPO bar.

**Figure 9: Qualitative results of segmentation.** We visualize results from probing on the CLIP-L/14 336px encoder, post-trained with SFT and DPO in MLLMs. The DPO-trained vision encoder (VE) yields more accurate segmentation maps that closely align with the ground truth. More results are in Figure G.

> **Finding 4:** DPO steers the vision encoder toward a more fine-grained analysis of visual information, improving its object localization capabilities.

**Vision & Language alignment.** Huh et al. (2024) proposed a representation alignment metric to evaluate representation similarity between models trained on different modalities, such as vision and language; typically, larger and stronger vision models show higher alignment with LLMs. We adopt this metric to evaluate the representations of a vision encoder. As shown in Figure 10, vision encoders trained with DPO show stronger alignment scores. Additionally, pairing with a larger LLM leads to consistently higher alignment scores, which supports our aforementioned hypothesis that larger LLMs transmit more useful signals to the vision encoder during backpropagation.

> **Finding 5:** The vision encoder benefits from a larger LLM, which provides more informative backward signals for visual representation within an MLLM.

## 5  WHAT'S NEXT: UNLOCKING VISION MODEL POTENTIAL VIA RL

Our analysis has shown that training a vision model with an LLM via DPO builds more fine-grained visual representations than SFT. We now reframe this training process into an effective strategy for evolving vision models, which we term Preference-Instructed Vision OpTimization (PIVOT). In this section, we apply PIVOT to existing vision models that are widely adopted as vision encoders in MLLMs. These include encoders pretrained with image-language supervision[4] (*e.g.*, CLIP and SigLIP) or with vision-only self-supervision (*e.g.*, MAE (He et al. 2022) and DINOv2 (Oquab et al. 2024)). Our objective is to investigate how much these vision models can be improved by PIVOT for use in MLLM.

### 5.1  EXPERIMENTAL SETUP

The process begins with a vision encoder commonly used in MLLMs, such as CLIP and SigLIP1. The encoder is attached to an LLM and optimized through both pre-training and post-training with DPO or SFT—on 3M instruction-following samples and 20K preference pairs, as described in Section 3.1.

---

[4]Following Cambrian (Tong et al. 2024a), we consider CLIP training as strongly supervised, as language provides richer supervision than class labels.

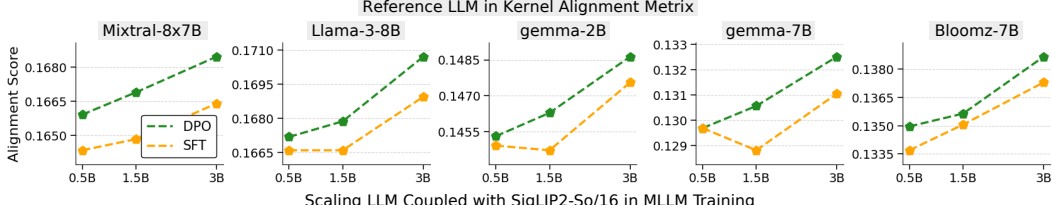

Figure 10: **representational alignment.** We measure alignment (Huh et al. 2024) between reference LLMs and vision encoders trained within MLLMs. SigLIP2-So/16, paired with three different LLM scales (x-axis), is trained with DPO or SFT and then used to compute alignment scores against five reference LLMs.

We refer to this training procedure as `PIVOT`. Afterward, the vision encoder is detached from the LLM, its weights are frozen, and the resulting model is termed the `PIVOT`-enhanced encoder. We evaluate the performance of `PIVOT`-enhanced encoder by combining it with Qwen2.5-1.5B and build an MLLM. The combined model is optimized with projector-only pretraining on LAION/CC/SBU-558K (Liu et al. 2024a), followed by instruction finetuning of the projector and LLM on Cambrian's 737K dataset. This design allows us to isolate the encoder's capability and assess the effectiveness of `PIVOT` representations within MLLMs. Note that we follow the same evaluation protocol as prior works such as Cambrian (Tong et al. 2024a), DINO-MLLM (Fan et al. 2025), and MLLM-data (Han et al. 2025), which has been demonstrated to allows us to study visual representations efficiently. More details are in Figure E.

The idea of `PIVOT` is simple yet effective: training vision models with LLM-head using DPO. We highlight the contributions of `PIVOT`: (*i*) positioning `PIVOT` *not* as a new method, but as an under-explored training regime. (*ii*) showing that it can develop significantly better MLLMs than those using original vision models, revealing substantial room for improvement in state-of-the-art vision models. (*iii*) presenting the first evidence that DPO reshapes visual features with more positive effects than SFT on standard vision benchmarks as well as on multimodal tasks.

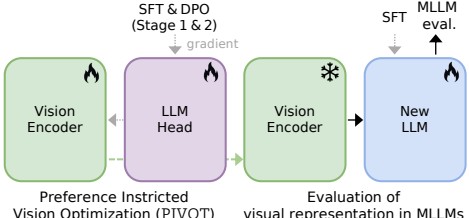

Figure 11: Evaluation setup for the `PIVOT`-enhanced vision encoder within MLLMs.

## 5.2 RESULTS

The results are presented in Table 1. In the following, we describe the main comparisons in detail.

**SigLIP1 → SigLIP2.** We compare an MLLM using the original SigLIP2 encoder against a `PIVOT`-enhanced SigLIP1. SigLIP2 is a more recent model, developed with substantially larger datasets and an advanced training scheme compared to its predecessor. An MLLM leveraging the SigLIP2-So/16 encoder achieves an average VQA score of 52.4%p. However, by enhancing SigLIP1-So/14 with the `PIVOT` process, we obtain an MLLM that achieves an average VQA score of 53.2%p, surpassing those with SigLIP2-So/16.

**SigLIP2–So/16 → SigLIP2-g/16.** SigLIP2-g/16 is considered to have the strongest representations in its family due to its large scale. We compare its MLLM performance against a `PIVOT`-enhanced SigLIP2-So/16. Despite having 2.5 times fewer parameters, the So/16 model outperforms the g/16 model, achieving a score of 55.6%p versus 53.9%p. This shows the considerable potential for enhancing popular vision backbones for optimal performance within MLLMs.

**DPO vs. SFT on `PIVOT`.** In Section 4, we show that DPO during post-training benefits even vision encoders within MLLM. Similarly, a vision encoder enhanced by DPO (*i.e.*, `PIVOT`) provides a 1.3%p advantage over one enhanced with SFT (56.7%p vs. 55.4%p) in the MLLM application when using SigLIP2-g/16. Here, SFT can be seen as similar to the language alignment of (Bolya et al. 2025). This result indicates that DPO's advantage over SFT continues in the context of `PIVOT`. Thus, we adopt DPO as the default choice for `PIVOT`.

**Classic vision encoders + `PIVOT`.** We investigate the effect of `PIVOT` on diverse vision encoders and find that all five models improve MLLM performance. An interesting observation is that this improvement holds not only for vision-only self-supervised models such as MAE (He et al. 2022)

| Evolving vision encoder for MLLM applications | | | | MLLM combining the vision encoder with Qwen2.5-1.5B | | | | |
|---|---|---|---|---|---|---|---|---|
| Model | Size | # Params | # Samples seen | Average (All) | General | OCR&Chart | Vision-Cent. | Knowledge |
| SigLIP 1 (2023) | So400m | 400M | 30B | 50.9 | 65.4 | 42.3 | 49.8 | 46.0 |
| + SFT | | | 30B + 0.003B | 52.2 | 66.5 | 45.2 | 50.8 | 46.3 |
| + `PIVOT` | | | 30B + 0.003B | **53.2** | **67.7** | **46.8** | **51.7** | **46.6** |
| SigLIP 2 (2025) | So400m | 400M | 40B | 52.4 | 66.2 | 46.6 | 50.6 | 46.1 |
| + SFT | | | 40B + 0.003B | 54.6 | 66.9 | 52.2 | 51.7 | 47.7 |
| + `PIVOT` | | | 40B + 0.003B | **55.6** | **68.1** | **53.9** | **52.4** | **48.1** |
| SigLIP 2 (2025) | giant | 1000M | 40B | 53.9 | 66.5 | 50.8 | 51.9 | 46.4 |
| + SFT | | | 40B + 0.003B | 55.4 | 67.4 | 52.8 | 53.1 | 48.5 |
| + `PIVOT` | | | 40B + 0.003B | **56.7** | **68.5** | **54.7** | **54.2** | **49.3** |
| **Classical vision encoders** | | | | Average (All) | General | OCR&Chart | Vision-Cent. | Knowledge |
| Model | Size | # Params | # Samples seen | | | | | |
| CLIP (2021) | large | 303M | 32B | 46.3 | 62.1 | 35.1 | 43.0 | 45.0 |
| + `PIVOT` | | | 32B + 0.003B | **49.5** | **64.6** | **37.8** | **48.6** | **47.1** |
| DINOv2 (2024) | giant | 1000M | 2B | 40.9 | 58.4 | 17.6 | 45.1 | 42.6 |
| + `PIVOT` | | | 2B + 0.003B | **43.6** | **62.1** | **18.7** | **49.2** | **44.3** |
| MAE (2022) | huge | 632M | 2B | 36.8 | 47.6 | 17.3 | 40.2 | 42.0 |
| + `PIVOT` | | | 2B + 0.003B | **39.7** | **52.5** | **18.2** | **43.3** | **44.6** |
| MOCO (2020) | base | 86M | 1.4B | 35.3 | 42.5 | 17.1 | 39.6 | 42.1 |
| + `PIVOT` | | | 1.4B + 0.003B | **37.5** | **47.4** | **17.6** | **41.0** | **44.1** |
| ImageNet-Sup (2021) | huge | 632M | N/A | 35.5 | 44.6 | 17.2 | 38.2 | 42.1 |
| + `PIVOT` | | | N/A | **37.7** | **47.3** | **18.1** | **40.3** | **45.1** |
| **Model ensemble** (Tong et al. 2024b) | | | | Average (All) | General | OCR&Chart | Vision-Cent. | Knowledge |
| Model | | # Params | | | | | | |
| SigLIP 1-So400m+ DINOv2-L | | 700M | | 49.4 | 64.5 | 41.5 | 46.5 | 45.1 |
| SigLIP 1-So400m+ ConvNeXt-XXL | | 1.25B | | 51.4 | 65.9 | 44.6 | 49.1 | 45.9 |
| SigLIP 1-So400m_`PIVOT` + ConvNeXt-XXL | | 1.25B | | **53.6** | **67.3** | **48.5** | **52.5** | **46.0** |

Table 1: **Influence of `PIVOT` on existing vision models.** We apply `PIVOT` to reveal the potential for improving existing vision models for MLLMs. Following the setup in Section 3.1, vision model is trained with a Qwen2.5-1.5B LLM-head on 3M samples, and then finetuned with either SFT (+ SFT) or DPO (+ `PIVOT`) on 20K data. '# samples seen' refers number samples used for whole training as in Cherti et al. (2023); Zhai et al. (2023).

and MOCO (He et al. 2020), but also for the supervised encoder (Dosovitskiy et al. 2021) trained solely with an image classification loss on the ImageNet dataset.

**Model ensemble.** The idea of model ensemble utilizing multiple vision encoders for a single MLLM has been explored in prior works (Tong et al. 2024b;a). The experiments show that combining SigLIP1-So/14 and ConvNeXt-XXL increases the average score from 50.9%p to 51.4%p, although it requires a greater number of parameters. We show that SigLIP1-So/14+ `PIVOT` alone achieves a superior score of 53.2%p without increasing parameters. Furthermore, combining this SigLIP1+ `PIVOT` with ConvNeXt-XXL results in an additional performance gain, reaching a score of 53.6%p.

> **Finding 6:** Existing vision models possess substantial potential for improvement within MLLMs, which can be unlocked by `PIVOT`.

We provide additional experimental results in Section B.7, including the impact of training data scale and different usage strategies for the `PIVOT`-enhanced projector.

# 6 CONCLUSION

In this work, we investigated the differential impacts of SFT and RL on both MLLMs and their vision encoders. Our experiments first demonstrated that DPO, a form of RL, achieves superior MLLM performance over SFT, particularly on tasks requiring detailed visual comprehension. A subsequent, focused analysis of the vision encoder revealed that DPO induces stronger and more localized visual features. We then consolidated these findings into `PIVOT`, a practical recipe, and validated its efficacy across a diverse range of vision encoders. We hope this research contributes to the broader goal of enabling MLLMs to better perceive and interpret visual information.

Beyond our main paper, the supplementary material contains additional analyses, including other RL algorithms (PPO, GRPO, and MPO) vs. SFT (Section B.1), more SFT-friendly settings(Section B.4), text-only benchmarks (Section B.6), and `PIVOT` ablations(Section B.7).

## ACKNOWLEDGMENTS

We thank NAVER AI Lab for its generous support. We are also grateful to Heejin Do, Hyesong Choi, Taekyung Kim, and Jaehui Hwang for their valuable feedback. This work was also supported by Institute of Information & communications Technology Planning & Evaluation (IITP) grant funded by the Korea government(MSIT) (RS-2022-00143911, AI Excellence Global Innovative Leader Education Program).

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

# Appendix

## Table of Contents

## A    RELATED WORK

### A.1    MLLMS.

Building on the success of LLMs, the development of MLLMs has become a prominent research direction for equipping LLMs with visual understanding (Team et al. 2024a; Achiam et al. 2023; Bai et al. 2023b). The standard paradigm involves connecting a pretrained vision encoder to an LLM via a multimodal projector, creating a strong baseline (Liu et al. 2023a; Li et al. 2023a; Jaegle et al. 2022). Subsequent advancements have been achieved by employing larger components (Li et al. 2024b), training on higher-quality conversational data (Li et al. 2025a; Bai et al. 2025; Han et al. 2025), or introducing new techniques for stronger visual understanding (Kar et al. 2024; Lin et al. 2024; Tong et al. 2024b). The dominant training strategy for these models has been SFT (Touvron et al. 2023a; Brown et al. 2020; Dai et al. 2019), where the model learns to generate a ground-truth response for a given visual input and query. As noted in Cambrian (Tong et al. 2024a), while SFT has been effective, RL is emerging as a promising alternative to potentially surpass the performance ceilings of current methods.

### A.2    LLMS WITH RL.

Following the development of various Transformer-based language models (Raffel et al. 2020; Lewis et al. 2020; Radford et al. 2018; Touvron et al. 2023a; Bai et al. 2023a), trained with objectives such as masked modeling (Devlin et al. 2019) and SFT, a major breakthrough was achieved by aligning LLMs with human preferences through RLHF (Christiano et al. 2017; Ouyang et al. 2022; Touvron et al. 2023b). The foundational method involved using PPO (Schulman et al. 2017) to optimize an SFT model against a reward model trained on preference data. This paradigm has since evolved: DPO (Rafailov et al. 2023) directly instills preference alignment by optimizing on pairwise preferences, and GRPO (Shao et al. 2024) updates the policy using group-wise relative rankings of candidate responses. This line of research, which also includes methods like IPO (Azar et al. 2024), KTO (Ethayarajh et al. 2024), and ORPO (Hong et al. 2024), has consistently demonstrated the power of RL. Whereas prior works, RL's Razor (Shenfeld et al. 2025) and RL-Squeezes (Matsutani et al. 2025), compared RL and SFT in the context of LLM adaptation to new tasks, we conduct a parallel investigation into how these distinct trainings impact MLLMs.

### A.3    MLLMS WITH RL

The MLLM field is increasingly adopting RL to push beyond the limitations of SFT, mirroring the evolution of LLMs. We provide a comprehensive list in Table A. Several studies (Yu et al. 2025; Xiong et al. 2025), including LLaVA-RLHF (Sun et al. 2024a) and MPO (Wang et al. 2024b), have reported that applying additional preference alignment to an SFT-trained MLLM can boost its performance. In parallel, other works have proposed DPO extensions for MLLMs: RLHF-V (Yu et al. 2024), OPA-DPO (Yang et al. 2025c), and HDPO (Fu et al. 2025b). These approaches reweight token-level losses on disagreement tokens between the chosen and rejected responses, or combine SFT with DPO for joint training. Some studies (Zadeh et al. 2025; Xie et al. 2024), such as CHiP (Fu et al. 2025a) and mDPO (Wang et al. 2024a), have shown that incorporating visual preference data reduces perceptual errors in MLLMs. Finally, Chu et al. (2025) and Li et al. (2025b) have indicated that RL is advantageous for adapting MLLM's inherent knowledge to special environments, like card games, map navigation, or robot action planning. Our work conducts a controlled comparison between SFT and DPO (Section 3.1) and, unlike RL-vs.-SFT studies (Chu et al. 2025; Li et al. 2025b), evaluates on common benchmarks rather than specialized settings.

### A.4    VISION-CENTRIC PRE-TRAININGS

The pretraining of vision models has largely followed two paths: image-only self-supervised learning and image-language supervised learning. The former, encompassing contrastive (He et al. 2020; Chen et al. 2020; Caron et al. 2020; Chen & He 2021; Caron et al. 2021) learning and masked-image-modeling (Bao et al. 2022; He et al. 2022), has proven effective for creating visual representation models for classic vision tasks like image classification and segmentation. The latter, as in CLIP (Radford et al. 2021), SigLIP (Zhai et al. 2023; Tschannen et al. 2025), and EvaCLIP (Sun et al. 2023),

| Abbreviation | Title | Venue | Year | RL |
|---|---|---|---|---|
| RLHF-V (Yu et al. 2024) | Towards Trustworthy MLLMs via Behavior Alignment from Fine-grained Correctional Human Feedback | CVPR | 2024 | DPO |
| RLAIF-V (Yu et al. 2025) | Open-Source AI Feedback Leads to Super GPT-4V Trustworthiness | CVPR | 2025 | DPO |
| LLaVA-RLHF (Sun et al. 2024a) | Aligning Large Multimodal Models with Factually Augmented RLHF | ACL | 2024 | PPO |
| LLaVA-Critic (Xiong et al. 2025) | Learning to Evaluate Multimodal Models | CVPR | 2025 | DPO |
| OPA-DPO (Yang et al. 2025c) | Mitigating Hallucinations in Large Vision-Language Models via DPO: On-Policy Data Hold the Key | CVPR | 2025 | DPO |
| HDPO (Fu et al. 2025b) | Mitigating Hallucination in Multimodal Large Language Model via Hallucination-targeted Direct Preference Optimization | ACL | 2025 | DPO |
| CHiP (Fu et al. 2025a) | Cross-modal Hierarchical Direct Preference Optimization for Multimodal LLMs | ICLR | 2025 | DPO |
| mDPO (Wang et al. 2024a) | Conditional Preference Optimization for Multimodal Large Language Models | EMNLP | 2024 | DPO |
| LPOI (Zadeh et al. 2025) | Listwise Preference Optimization for Vision Language Models | ACL | 2025 | DPO |
| V-DPO (Xie et al. 2024) | Mitigating Hallucination in Large Vision Language Models via Vision-Guided Direct Preference Optimization | EMNLP | 2024 | DPO |
| MPO (Wang et al. 2024b) | Enhancing the Reasoning Ability of Multimodal Large Language Models via Mixed Preference Optimization | arXiv | 2024 | DPO |
| RL Generalizes (Chu et al. 2025) | SFT Memorizes, RL Generalizes: A Comparative Study of Foundation Model Post-training | ICML | 2025 | PPO |
| SimpleVLA-RL (Li et al. 2025b) | SimpleVLA-RL: Scaling VLA Training via Reinforcement Learning | arXiv | 2025 | GRPO |
| LongPerceptualThoughts (Liao et al. 2025) | LongPerceptualThoughts: Distilling System-2 Reasoning for System-1 Perception | arXiv | 2025 | DPO |

Table A: **List of RL-based MLLM works.** We provide an overview of methods with their venues, years, and RL optimization strategies, and note that most of the previous studies have adopted DPO (Rafailov et al. 2023) as one of their RL baselines.

aligns vision and language, enabling strong zero-shot recognition and making these models popular backbones for MLLMs (Li et al. 2025a). Our `PIVOT` is a CLIP-style alternative for training vision encoders, as both use language-aligned supervision. Applied to existing encoders, it evolve into MLLM-ready encoders with <1% of the GPUs and data relative to SigLIP2 training.

Recently, Perception Encoder (Bolya et al. 2025) explored improved recipes for building powerful vision encoders through vision-language pre-training. Its language alignment stage follows a strategy similar to the '+ SFT' setting in Table 1. Unlike their focus on SFT-driven representation changes, we investigate how RL training influences vision representations.

# B  ADDITIONAL EXPERIMENTS

## B.1  OTHER RL ALGORITHMS VS. SFT

**Rationale for Focusing on DPO.** In our analysis, we focus on DPO as the primary RL method. This choice reflects practical considerations in MLLM post-training. **DPO** (Rafailov et al. 2023) avoids dependence on a reward model, reducing confounding factors when comparing with SFT. It also operates on a data format similar to SFT, $(I_i, q_i, y_i^c, y_i^r)$, enabling a controlled comparison with the same number of image–query–response pairs, as described in Section 3.1. In contrast, **PPO** (Schulman et al. 2017) requires an external reward model trained on a separate dataset and introduces additional RLHF data, making a fair comparison with SFT difficult. **GRPO** (Shao et al. 2024) depends on verifiable signals such as math or coding problems for LLMs, or object counting and bounding-box annotations for MLLMs. These properties differ from those of typical SFT datasets, hindering the construction of a matched evaluation setup. While these constraints motivated our focus

| Code | Task | GRPO data | SFT data | GRPO code | SFT code | Vision encoder |
|---|---|---|---|---|---|---|
| R1-V (2025) | object counting / geometry math | ✓ | ✗ | ✓ | ✓ | update |
| EasyR1 (2025) | geometry math | ✓ | ✗ | ✓ | ✗ | update |
| SimpleVLA-RL (2025b) | robot action planning | ✓ | ✗ | ✓ | ✗ | freeze |
| VLM-R1 (2025) | bounding-box annotation | ✓ | ✓ | ✓ | ✓ | update |

| Code | Task | PPO data | SFT data | PPO code | SFT code | Vision encoder |
|---|---|---|---|---|---|---|
| RL4VLM (2024) | card game / robot action planning | ✓ | ✓ | ✓ | ✗ | freeze |
| RL Generalize (2025) | card game / navigation | ✓ | ✗ | ✓ | ✓ | update |
| LLaVA-RLHF (2024a) | conversation | ✓ | ✓ | ✓ | ✓ | freeze |

Table B: **Overview of GRPO- and PPO-based MLLM GitHub repository.** We summarize open-source implementations for RL-based MLLM training, highlighting their main tasks, data and code availability, and encoder update strategies.

| Model | Post-train | Avg (All) | General | OCR&Chart | Vision | Knowledge |
|---|---|---|---|---|---|---|
| QwenVL-2.5-3B | SFT | 62.8 | 69.4 | 69.0 | 58.0 | 55.0 |
| QwenVL-2.5-3B | GRPO | **65.9** | **72.1** | **73.3** | **61.4** | **56.7** |
| | | +3.1 | +2.7 | +4.3 | +3.4 | +1.7 |

| Model | Post-train | ImageNet | Segmentation |
|---|---|---|---|
| QwenVL-2.5-3B | SFT | 52.01 | 33.71 |
| QwenVL-2.5-3B | GRPO | **53.94** | **35.54** |
| | | +1.93 | +1.8 |

Table C: **Evaluation of QwenVL-2.5-3B under GRPO vs. SFT post-training.** We present results on MLLM benchmarks for MLLMs trained with different objectives (top). We also evaluate the vision encoder updated within MLLMs on vision-only benchmarks (bottom).

on DPO, we also extend our experiments to PPO, GRPO, and a DPO variant (*i.e.*, MPO (Wang et al. 2024b)) to validate that our findings generalize beyond DPO.

**GRPO vs. SFT.** We first survey publicly available implementations of GRPO, as summarized in Table B. Among them, we adopt the codebase from VLM-RL for our GRPO vs. SFT experiments, as it provides a *validated* training strategy for proper GRPO implementation. We use QwenVL-2.5-3B as the base model and perform 1,500 post-training steps with different objectives, while keeping the remaining training configurations unchanged from the original repository. To prevent GPU memory issues during evaluation, we fix the input image resolution to 664×664. For vision-only evaluation, we resize images to 336×336 to extract visual representations and use features prior to the adaptor. The results are reported in Table C. GRPO consistently outperforms SFT across both MLLM and vision-only evaluations. In particular, RL (i.e., GRPO) enhances MLLMs more on vision-intensive tasks than SFT, consistent with Finding 2 in Section 3. Moreover, GRPO yields stronger visual representations, supporting the observations in Findings 3 and 4 of Section 4 that MLLM post-training reshapes the vision encoder and that RL provides more effective visual refinement.

**PPO vs. SFT.** We next turn to the comparison between PPO and SFT and provide a survey of available codebases, as presented in Table B. Based on these conditions, we choose to adopt the LLaVA-RLHF (Sun et al. 2024a) implementation. For PPO, we utilize their publicly released model. For SFT, we post-train a LLaVA-1.0-7B model for one epoch using their SFT data under the original LLaVA framework (Liu et al. 2023a). We note that LLaVA employs the CLIP-L/14-224px encoder, where all input images are automatically resized to 224×224. We present the results in Table D and observe that the PPO-trained MLLM outperforms its SFT counterpart across 16 MLLM benchmarks, with a notable gain on OCR&Chart tasks. We also note that LLaVA-1.0-13B is used as the reward model in this codebase, introducing confounding factors beyond the training objectives alone. Our DPO vs. SFT comparison mitigates this issue by avoiding a reward model and therefore provides a clearer perspective on RL versus SFT, consistent with prior works summarized in Table A.

**MPO vs. DPO vs. SFT.** Finally, we extend our comparison by including MPO (Wang et al. 2024b) as a variant of DPO. Mixed preference optimization (i.e., MPO) combines the objectives of DPO, SFT, and a binary preference classification loss. We integrate their implementation into our codebase and run the experiments. The Table E (top & middle) report MLLM and vision-only evaluation results under different MLLM training algorithms. MPO achieves the highest average score in the MLLM evaluation, outperforming SFT, though its gain over DPO remains modest. On vision-only benchmarks, MPO matches DPO, suggesting that its impact on visual representation learning is limited. Moreover, we present additional results in Table E (bottom), following Section 5.1, where visual representations are evaluated within MLLMs. The results demonstrate that DPO and MPO

| Model | Post-train | Avg (All) | General | OCR&Chart | Vision | Knowledge |
|---|---|---|---|---|---|---|
| LLaVA-1.0-7B | SFT | 33.1 | 43.6 | 26.8 | 26.7 | 35.2 |
| LLaVA-1.0-7B | PPO | **34.6** | **44.0** | **30.0** | **28.0** | **36.4** |
|  |  | +1.5 | +0.4 | +3.2 | +1.3 | +1.2 |

Table D: **Evaluation of LLaVA-1.0-7B under PPO vs. SFT post-training.** We present results on MLLM benchmarks for MLLMs trained with different objectives (left).

| Model | Post-train | Avg | General | OCR&Chart | Vision | Knowledge |
|---|---|---|---|---|---|---|
| Qwen2.5-1.5B + SigLIP2-L/16 | SFT | 59.4 | 68.0 | 60.9 | 56.4 | 52.4 |
| Qwen2.5-1.5B + SigLIP2-L/16 | DPO | 61.0 | 70.0 | **62.6** | 58.4 | 53.0 |
| Qwen2.5-1.5B + SigLIP2-L/16 | MPO | **61.5** | **70.2** | 62.2 | **59.6** | **54.0** |

| Model | Post-train | ImageNet | Segmentation |
|---|---|---|---|
| Qwen2.5-1.5B + SigLIP2-L/16 | SFT | 48.53 | 30.18 |
| Qwen2.5-1.5B + SigLIP2-L/16 | DPO | **50.51** | 30.89 |
| Qwen2.5-1.5B + SigLIP2-L/16 | MPO | 50.18 | **31.07** |

| MLLM = Qwen1.5-1.5B | Avg (All) | General | OCR&Chart | Vision | Knowledge |
|---|---|---|---|---|---|
| + Original SigLIP2-So/16 | 52.4 | 66.2 | 46.6 | 50.6 | 46.1 |
| + Stage2-SFT | 54.6 | 66.9 | 52.2 | 51.7 | 47.7 |
| + Stage2-DPO (i.e., PIVOT) | **55.6** | **68.1** | **53.9** | 52.4 | 48.1 |
| + Stage2-MPO | **55.6** | 67.9 | 53.8 | **52.6** | **48.3** |

Table E: **Evaluation of MLLMs under SFT, DPO, and MPO post-training.** We present results on MLLM benchmarks for models trained with different post-training objectives (top). We further evaluate the updated vision encoder on vision-only benchmarks (middle), and finally assess the same encoder when re-integrated and evaluated within MLLMs (bottom).

both improve visual representation quality over SFT, with MPO showing slight gains in vision-centric and knowledge tasks.

Overall, our experiments with GRPO, PPO, and MPO demonstrate that RL-based training improves MLLM performance and visual representations beyond SFT, confirming that our findings hold across other RL algorithms. The full results across the 16 benchmarks are provided in Table N and Table O.

## B.2 MLLM TRAINING SENSITIVITY

We examine how variations in the learning rate (LR) affect MLLM performance under different post-training strategies. We conduct experiments with Qwen2.5-3B and SigLIP2-So/16 and show the results in Figure A. Since SFT and DPO rely on fundamentally different loss formulations, their optimal learning rates naturally diverge. In practice, we observe that DPO requires substantially smaller LRs than SFT, partly because DPO accounts for both chosen and rejected responses, effectively doubling the batch size per iteration compared to SFT.

## B.3 MLLM PERFORMANCE UNDER NEW DATA DISTRIBUTIONS

**Motivation.** Previous studies, including RLgeneralize (Chu et al. 2025), SimpleVLA-RL (Li et al. 2025b), and RL-Razor (Shenfeld et al. 2025), have posited that RL is beneficial for adapting to new data distributions, mitigating performance degradation and catastrophic forgetting. Unfortunately, they either focused on specialized environments, such as card gaming and robot action planning, or conducted evaluations confined to the knowledge domain like mathematics. Hence, we examine how our MLLMs behave on more common VQA benchmarks when the *Stage 2* post-training data distribution differs from that of *Stage 1* pre-training.

**Experimental setup.** The LLaVA-OneVision samples predominantly contain short answers with fewer than 50 words and lack special tokens such as <think>...</think> and <review>...</review>. In contrast, the MMPR samples occasionally include longer responses and diverse annotation patterns. We exploit this discrepancy by constructing a new *Stage 2* post-training dataset based on MMPR. Specifically, we sample 20K instances from MMPR, where a fraction $r\%$ (0%, 50%, or 80%) consists of samples that either exceed 100 words or contain special tokens. The remaining $100-r\%$ of the dataset is randomly sampled from the rest of MMPR following our original setup.

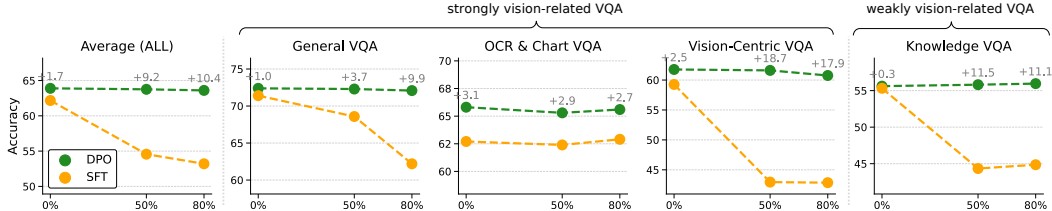

**Figure A: Training sensitivity in Multimodal LLMs.** We conduct an analysis of performance with respect to learning rate under different post-tuning strategies, using Qwen2.5-3B combined with SigLIP2-so400m. The x-axis shows learning rates on a log scale.

**Figure B: MLLM training under new data distribution.** We post-train (*i.e.*, *Stage 2*) an MLLM (Qwen2.5-3B + SigLIP2-So/16) under varying proportions of samples from a shifted distribution (0%, 50%, 80%). DPO remains stable, while SFT shows substantial declines, particularly on general and vision-centric VQA benchmarks.

**Results.** The results, shown in Figure B, reveal that DPO maintains robust performance even as the proportion of new-distribution samples increases. In contrast, SFT-trained MLLMs experience a sharp decline: while achieving 62.2%p with 0% of new-distribution samples, their performance drops to 53.2%p when the ratio increases to 80%. The degradation is especially pronounced in vision-centric VQA tasks, where the accuracy gap between DPO and SFT reaches 17.9%p with 80% new-distribution samples. It demonstrates that the trends observed in earlier RL studies (Chu et al. 2025; Li et al. 2025b; Shenfeld et al. 2025) also generalize across the broad set of 16 benchmarks considered in our evaluation.

### B.4 MLLM PERFORMANCE UNDER MORE SFT-FRIENDLY DATA

**Motivation.** In our main experiments, we use the MPO (Wang et al. 2024b) dataset for post-training. Since this dataset is originally designed for DPO supervision, it may inherently favor DPO. To mitigate this concern and examine whether the relative performance depends on the data characteristics, we additionally construct post-training datasets that are intentionally more SFT-friendly.

**Setup.** We again sample 20K instances from MPO and apply the following constraints to each pair $(I_t, x_t, y_t^c, y_t^r)$. The 'Under30' setting retains only samples where the chosen response $y_t^c$ contains fewer than 30 words, similar in style to 'clear-answer' data. The 'NoThink' setting removes samples containing special reasoning tokens (e.g., <think>, <review>) and restricts $y_t^c$ to fewer than 60 words. These filters yield datasets that align more closely with typical SFT supervision, reducing the implicit advantage that DPO may receive from the original MPO distribution. We repeat the SFT–DPO comparison of Section 3.1 using Qwen2.5-1.5B and Qwen2.5-3B as the LLM backbone, paired with SigLIP2-So/16. The results are summarized in Table F.

**Results.** We observe that these SFT-friendly datasets reduces the data-efficiency advantage of DPO. However, its performance does not fall below that of SFT. We hypothesize that the shorter responses decrease the informational gap between chosen and rejected samples, which diminishes DPO's effectiveness. Nevertheless, DPO does not fall behind SFT; its performance remains better even under these SFT-friendly conditions. The full results across the 16 benchmarks are provided in Table P.

### B.5 MLLM PERFORMANCE ON HALLUCINATION, CAPTIONING, AND ROBUSTNESS BENCHMARKS

**Setup.** We extend our analysis to a broader set of vision tasks to assess the generality of our findings. As summarized in Table G, we evaluate models on hallucination, captioning, and real-world robustness benchmarks. For hallucination, we include HallusionBench (Guan et al. 2024)

| LLM backbone | Train | Data | Avg. (All) | General | OCR&Chart | Vision-cent. | Knowledge |
|---|---|---|---|---|---|---|---|
| Qwen2.5-1.5B | DPO | Origin | **61.0** | **69.8** | **63.1** | **57.7** | **53.3** |
| Qwen2.5-1.5B | SFT | Origin | 59.1 | 68.3 | 60.5 | 54.8 | 52.8 |
| | | | +1.9 | +1.5 | +2.6 | +2.9 | +0.5 |
| Qwen2.5-1.5B | DPO | NoThink | **60.8** | **69.3** | **62.3** | **58.3** | **53.2** |
| Qwen2.5-1.5B | SFT | NoThink | 59.5 | 68.9 | 60.5 | 56.3 | 52.4 |
| | | | +1.2 | +0.4 | +1.7 | +2.0 | +0.7 |
| Qwen2.5-1.5B | DPO | Under30 | **60.1** | **69.6** | **60.0** | **57.4** | 53.3 |
| Qwen2.5-1.5B | SFT | Under30 | 59.3 | 69.0 | 59.2 | 56.1 | 52.8 |
| | | | +0.8 | +0.6 | +0.8 | +1.3 | +0.5 |
| Qwen2.5-3B | DPO | Origin | **63.9** | **72.4** | **66.0** | **61.9** | **55.3** |
| Qwen2.5-3B | SFT | Origin | 62.3 | 71.4 | 62.7 | 59.7 | 55.2 |
| | | | +1.7 | +1.0 | +3.3 | +2.2 | +0.1 |
| Qwen2.5-3B | DPO | NoThink | **63.5** | **71.8** | **64.1** | **62.5** | 55.6 |
| Qwen2.5-3B | SFT | NoThink | 62.9 | 71.6 | 63.8 | 60.0 | **56.2** |
| | | | +0.6 | +0.2 | +0.3 | +2.5 | -0.6 |
| Qwen2.5-3B | DPO | Under30 | **63.0** | **71.4** | **64.1** | **61.6** | 55.0 |
| Qwen2.5-3B | SFT | Under30 | 62.5 | 71.0 | 64.0 | 59.4 | **55.8** |
| | | | +0.5 | +0.5 | +0.1 | +2.2 | -0.8 |

Table F: **MLLM training with SFT-friendly datasets.** We train Qwen2.5-based MLLMs using different data configurations (Origin, NoThink, Under30) and evaluate them on 16 multimodal benchmarks. The vision encoder used in all MLLMs is fixed to SigLIP2-So/16.

| Model | Avg (All) | HallusionBench (2024) | | | POPE (2023b) | | |
|---|---|---|---|---|---|---|---|
| | | AnswerAcc | FaithAcc | QuestionAcc | precision | recall | f1 |
| MLLM-0.5B-SFT | 51.43 | 33.22 | **12.42** | 9.23 | 89.71 | 79.63 | 84.37 |
| MLLM-0.5B-DPO | **54.76** | **37.85** | 11.84 | **15.6** | **93.71** | **82.06** | **87.5** |
| MLLM-1.5B-SFT | 56.31 | 40.79 | 21.09 | 15.6 | 92.89 | **80.98** | 86.53 |
| MLLM-1.5B-DPO | **58.97** | **48.79** | **22.83** | **21.09** | **94.53** | 79.95 | **86.63** |

| Model | Avg (All) | MS COCO (2015) | | | DetailCaps-4870 (2024) | | |
|---|---|---|---|---|---|---|---|
| | | METEOR | ROUGE-L | BERTScore | METEOR | ROUGE-L | CAPTURE |
| MLLM-0.5B-SFT | 41.03 | **28.22** | 50.34 | 67.11 | 16.77 | 26.65 | 57.07 |
| MLLM-0.5B-DPO | **41.47** | 27.63 | **51.07** | **67.81** | **17.27** | **27.37** | **57.64** |
| MLLM-1.5B-SFT | 40.84 | 27.61 | 47.98 | 66.17 | 17.25 | **27.82** | 58.18 |
| MLLM-1.5B-DPO | **42.10** | **28.52** | 50.64 | **68.46** | **18.48** | 27.08 | **59.44** |

| Model | Avg (All) | NaturalBench (2024a) | | | MME_Real. (2025) | VL_RewardB. (2025c) |
|---|---|---|---|---|---|---|
| | | GroupAcc | ImageAcc | QuestionAcc | Avg | Avg |
| MLLM-0.5B-SFT | 12.65 | 12.84 | 42.18 | 0.3695 | 22.07 | **43.27** |
| MLLM-0.5B-DPO | **14.27** | **17.84** | **47.61** | **0.4213** | **27.24** | 43.02 |
| MLLM-1.5B-SFT | 13.75 | 16.53 | 44.03 | 0.4042 | 27.21 | 40.55 |
| MLLM-1.5B-DPO | **14.49** | **25.05** | **54.08** | **0.5087** | **28.35** | **42.82** |

Table G: **Comparison of DPO vs. SFT across extended MLLM benchmarks.** We present results on hallucination, captioning, and real-world/robustness benchmarks. We use MLLMs in Section 3.

and POPE (Li et al. 2023b). For captioning, we use MS COCO (Chen et al. 2015) and DetailCaps-4870 (Dong et al. 2024), which emphasize descriptive completeness. For robustness, we incorporate NaturalBench (Li et al. 2024a), MME-RealWorld (Zhang et al. 2025), and VL-RewardBench (Li et al. 2025c). Specifically, NaturalBench assesses visual grounding robustness by mitigating reliance on language priors, MME-RealWorld evaluates practical utility in complex real-world environments, and VL-RewardBench measures alignment with human preferences across diverse multimodal domains.

**Results.** Across all benchmarks, DPO consistently outperforms SFT. As discussed in Section 4, we attribute these gains to the contrastive nature of DPO, which provides fine-grained gradient signals to the vision encoder and strengthens the model's visual understanding. These enhanced visual representations translate into improved performance on captioning, hallucination mitigation, and real-world robustness evaluations.

### B.6   MLLM PERFORMANCE ON TEXT-ONLY BENCHMARKS

To examine whether multimodal post-training affects language-only abilities, we evaluate the resulting MLLMs on several text-only benchmarks, as summarized in Table H. Our evaluation includes HellaSwag (Zellers et al. 2019), MMLU (Hendrycks et al. 2021), and GSM8K (Cobbe et al. 2021),

| Model | MMLU (2021) | | | | | HellaSwag (2019) | GSM8K (2021) |
|---|---|---|---|---|---|---|---|
| | Overall | Humanities | Other | SocialSci | STEM | Acc | Acc |
| MLLM-0.5B-SFT | 30.7 | 31.2 | 33.9 | 31.7 | 25.8 | **45.8** | 33.4 |
| MLLM-0.5B-DPO | **38.1** | **37.3** | **43.9** | **40.3** | **31.2** | 45.3 | **34.7** |
| | +7.4 | +6.1 | +10.0 | +8.6 | +5.5 | -0.5 | +1.3 |
| MLLM-1.5B-SFT | 34.7 | 32.8 | 39.0 | 38.0 | 29.8 | 55.4 | **57.7** |
| MLLM-1.5B-DPO | **46.8** | **42.7** | **51.6** | **54.5** | **40.5** | **56.9** | 57.3 |
| | +12.1 | +9.9 | +12.5 | +16.5 | +10.7 | +1.6 | -0.4 |

Table H: **Evaluation on text-only benchmarks.** We report MMLU, HellaSwag, and GSM8K performance of SFT vs. DPO trained MLLMs across two model scales.

| Evolving vision encoder | | | | MLLM combining the vision encoder with Qwen2.5 | | | | | |
|---|---|---|---|---|---|---|---|---|---|
| Vision encoder | # Params | PIVOT-proj. | LLM | Add. layer | Total layer | Avg. (All) | General. | OCR&Chart. | Vision-Cen. | Knowledge. |
| SigLIP2-So/16$_{+PIVOT}$ | 400M | 0 | Qwen2.5-0.5B | 2 | 2 | 42.9 | 56.3 | 39.1 | 37.9 | 38.3 |
| SigLIP2-So/16$_{+PIVOT}$ | 400M | 2 | Qwen2.5-0.5B | 2 | 4 | 44.3 | 56.5 | 39.8 | 41.4 | 39.4 |
| SigLIP2-So/16$_{+PIVOT}$ | 400M | 1 | Qwen2.5-0.5B | 1 | 2 | 45.2 | 57.8 | 39.5 | 43.4 | 40.3 |
| SigLIP2-So/16$_{+PIVOT}$ | 400M | 0 | Qwen2.5-1.5B | 2 | 2 | 52.4 | 66.2 | 46.1 | 46.6 | 50.6 |
| SigLIP2-So/16$_{+PIVOT}$ | 400M | 2 | Qwen2.5-1.5B | 2 | 4 | 54.3 | 66.4 | 48.2 | 49.7 | 52.9 |
| SigLIP2-So/16$_{+PIVOT}$ | 400M | 1 | Qwen2.5-1.5B | 1 | 2 | **54.6** | **66.7** | 47.1 | **50.8** | **54.0** |

Table I: **Ablation on reusing the `PIVOT`-trained projector.** 'PIVOT-projector 0, 1, 2' denote configurations that reuse none, only the first layer, or two layers of the frozen `PIVOT`-trained projector, respectively. Additional trainable layers (*Stage 3*-projector) are appended before the LLM to match dimensionality. Among these, the 1+1 setup—reusing the first frozen layer with one new layer—achieves the best downstream MLLM performance during the final *Stage 3* in Figure E.

which are widely used for assessing LLM general language understanding. Results show that DPO-trained models achieve superior performance on MMLU compared to SFT. For HellaSwag and GSM8K, performance remains comparable, aligning with observations in Knowledge VQA. We speculate that this is due to data distribution, MPO (Wang et al. 2024b) data, used for multimodal training. MPO data includes general, science, and document VQA data, which overlaps with the target of MMLU. Otherwise, MPO's math data is highly focused on geometry that GSM8k doesn't cover. Also, MPO doesn't have data related to HellaSwag tasks. So, we conjecture that the data overlap causes different results between MMLU and others. Still, an in-depth study is needed, which we consider future work.

### B.7 PIVOT ABLATION STUDY

Beyond the results in Section 5, we perform further experiments to gain deeper insights into `PIVOT`.

**PIVOT-enhanced projector.** (**setup**) There are two components responsible for visual representation in an MLLM: the vision encoder and the projector. In Section 5, we examined how the vision encoder trained through *Stage 1* and *Stage 2* operates in *Stage 3* of Figure E. In this section, we extend the analysis to the projector, investigating whether reusing the `PIVOT`-tuned projector benefits the model under the same setting. The `PIVOT`-tuned projector obtained after *Stage 2* follows our standard MLLM architecture, consisting of a two-layer MLP. In *Stage 3*, we vary which part of this projector is reused, and denote the configurations as 'PIVOT-tuned 0, 1, and 2,' corresponding to using none, only the first linear layer, or the entire two-layer MLP, respectively.

In *Stage 3*, the `PIVOT`-tuned vision encoder and projector must be connected to a new LLM. To enable this, we introduce a new set of MLP layers, referred to as the Stage 3-projector. We vary the number of layers in this module, and define the total number of multimodal linear layers as the sum of those from the `PIVOT`-tuned projector and the Stage 3-projector. For example, "PIVOT projector 2, Add. layer 2" indicates that the two frozen layers from the `PIVOT`-tuned projector are reused, while two additional randomly initialized layers are appended in the Stage 3-projector, resulting in a total of four layers. (**results**) We observe that using the first linear layer from the `PIVOT`-tuned projector together with one additional layer (i.e., two layers in total) yields the best downstream MLLM performance. Contrary to the expectation that increasing the depth (e.g., 2+2 layers) would leverage more parameters and improve results, this configuration instead leads to inferior performance. Based on this finding, we adopt the 1+1 setting for all subsequent experiments in Section 5.

**Training with more data.** In Section 5.1, our `PIVOT`-enhanced vision encoder is paired with a new LLM and finetuned on the Cambrian-737K dataset. We examine whether the same trend holds with larger-scale data during *Stage 3* of Figure E by using the LLaVA-OV-3M dataset. The results are

| Evolving vision encoder | | MLLM combining the vision encoder with Qwen2.5 | | | | | | |
|---|---|---|---|---|---|---|---|---|
| Vision encoder | #Params | LLM | Data | Avg. (All) | General. | OCR&Chart. | Vision-Cen. | Knowledge. |
| SigLIP2-So/16 | 400M | Qwen2.5-1.5B | Cambrian-737K | 52.4 | 66.2 | 46.6 | 50.6 | 46.1 |
| +PIVOT | 400M | Qwen2.5-1.5B | Cambrian-737K | **55.6** | **68.1** | **53.9** | **52.4** | **48.1** |
| | | | | +3.2 | +1.9 | +7.3 | +1.8 | +2.0 |
| SigLIP2-So/16 | 400M | Qwen2.5-1.5B | LLaVA-OV-3M | 56.9 | 67.9 | 56.4 | 51.3 | 52.0 |
| +PIVOT | 400M | Qwen2.5-1.5B | LLaVA-OV-3M | **59.2** | **68.9** | **59.8** | **54.6** | **53.5** |
| | | | | +2.3 | +1.0 | +3.4 | +3.3 | +1.5 |
| SigLIP2-So/16 | 400M | Qwen2.5-0.5B | LLaVA-OV-3M | 49.0 | 58.8 | 47.2 | 45.2 | 44.6 |
| +PIVOT | 400M | Qwen2.5-0.5B | LLaVA-OV-3M | **50.6** | **59.9** | **51.0** | **46.5** | **45.1** |
| | | | | +1.6 | +1.1 | +3.8 | +1.3 | +0.5 |

Table J: **Effect of larger Stage 3 data on `PIVOT` performance.** Comparison between basic and `PIVOT`-enhanced SigLIP2-So/16 encoders paired with Qwen2.5, trained on either Cambrian-737K (Tong et al. 2024a) or LLaVA-OV-3M (Li et al. 2025a). `PIVOT` consistently improves MLLM performance, and its advantage remains robust even with larger-scale training data.

| Evolving vision encoder | | MLLM combining the vision encoder with Qwen2.5 | | | | | | |
|---|---|---|---|---|---|---|---|---|
| Vision encoder | #Params | LLM | Full train? | Avg. (All) | General. | OCR&Chart. | Vision-Cen. | Knowledge. |
| SigLIP2-So/16 | 400M | Qwen2.5-0.5B | ✓ | 45.1 | 57.1 | 44.6 | 39.6 | 39.2 |
| + PIVOT | 400M | Qwen2.5-0.5B | ✓ | **46.0** | **58.2** | **45.0** | **41.5** | **39.4** |
| | | | | +0.9 | +1.1 | +0.4 | +1.9 | +0.2 |
| SigLIP2-So/16 | 400M | Qwen2.5-1.5B | ✗ | 52.4 | 66.2 | 46.6 | 50.6 | 46.1 |
| SigLIP2-So/16 | 400M | Qwen2.5-1.5B | ✓ | 54.5 | 67.1 | 51.1 | 53.1 | 46.9 |
| + PIVOT | 400M | Qwen2.5-1.5B | ✓ | **55.2** | **67.5** | **52.1** | **54.1** | **47.2** |
| | | | | +0.7 | +0.4 | +1.0 | +1.0 | +0.3 |

Table K: **Effect of full-parameter training.** Although updating all model parameters inevitably alters the intrinsic representations of the vision encoder, we aim to understand its overall effect. MLLMs incorporating `PIVOT`-enhanced encoders exhibit clear and consistent gains over their baseline counterparts, highlighting the robustness of `PIVOT` beyond the controlled evaluation protocol.

reported in Table J. Our findings confirm that the advantage of `PIVOT` persists even with more data. For example, when combined with Qwen2.5-1.5B, the `PIVOT`-enhanced SigLIP2-So/16 achieves an average gain of +2.3%p over the base SigLIP2-So/16 encoder, demonstrating that the benefits of `PIVOT` are robust to data scale in the *Stage 3*.

**Training all parameters.** We adopt the same evaluation protocol of Cambrian (Tong et al. 2024a), DINO-MLLM (Fan et al. 2025), and MLLM-data (Han et al. 2025) in order to directly assess how useful the visual representations of a vision encoder itself are for an MLLM. As noted by Cambrian, it allows us to study visual representations efficiently. Obviously, training all parameters of the model during the *stage 3* of Figure E alters the intrinsic representations of the encoder. Nevertheless, we were interested in understanding how the MLLM performance changes when all parameters—including the vision encoder, LLM, and projector—are updated during *Stage 3*. The results, reported in Table K, show that MLLMs built upon `PIVOT`-enhanced encoders consistently outperform their counterparts.

## B.8 `PIVOT` PERFORMANCE COMPARISON WITH EXISTING MLLMS

To more comprehensively evaluate PIVOT-based MLLMs in Section 5, we compare its performance against a range of existing multimodal models. Our evaluation covers BLIP-2 (Li et al. 2023a), LLaVA-1.5 (Liu et al. 2024a), LLaVA-Next (Li et al. 2024b), LLaVA-OneVision (Li et al. 2025a), and SPHINX (Lin et al. 2024). The results in Table L show that our MLLM with PIVOT achieves strong performance. We have also added the corresponding citations in the related work section. The results show that PIVOT (1.5B) outperforms other MLLMs built on language models under 2B parameters and even exceeds larger models such as LLaVA-1.5B, BLIP-2, and SPHINX. Full results across the 16 benchmarks are provided in Table Q.

## C EXTENDED RESULTS FROM THE MAIN PAPER

### C.1 MLLM TRAINING DATA SCALING

In Figure 4, we analyze the effect of post-training data scale on MLLM performance, focusing on the average scores across all benchmarks. To complement this, we provide the results for each specific domain in Figure C. The results consistently show that the DPO-tuned MLLM outperforms its SFT

| Model | LLM size | Avg. (All) | General | OCR&Chart | Vision-Cent. | Knowledge |
|---|---|---|---|---|---|---|
| BLIP-2 | 7B | 43.7 | 48.5 | 52.8 | 31.0 | 42.5 |
| SPHINX | 7B | 48.7 | 65.6 | 33.2 | 52.5 | 41.4 |
| LLaVA-Next-Vincina | 7B | 50.6 | 68.3 | 38.0 | 52.8 | 43.9 |
| LLaVA-Next-LLaMA-3 | 8B | 60.4 | **71.3** | **64.3** | 57.5 | 49.0 |
| LLaVA-Next-Qwen2 | 7B | **62.0** | 70.5 | 63.3 | **62.3** | **53.3** |
| SPHINX-Tiny | 1.1B | 37.5 | 54.2 | 19.3 | 43.6 | 33.2 |
| LLaVA-OneVision | 0.5B | 50.0 | 60.4 | **57.0** | 42.7 | 39.4 |
| Ours (SigLIP1-So/16)+PIVOT | 1.5B | 53.2 | 67.7 | 46.8 | 51.7 | 46.6 |
| Ours (SigLIP2-So/16)+PIVOT | 1.5B | **55.6** | 68.1 | 53.9 | **52.4** | **48.1** |

Table L: **Comparison between PIVOT-based MLLMs and existing models.** We evaluate several 7B–8B large models and smaller 0.5B–1.5B variants, including our PIVOT-based models.

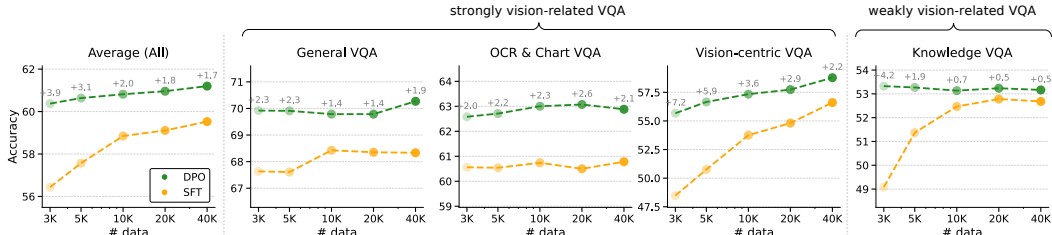

Figure C: **Scaling the amount of post-training data for MLLM.** We vary the size of training data for an MLLM built with Qwen2.5-1.5B and SigLIP2-So/16 and measure its performance.

counterpart as the amount of training data increases. For Knowledge VQA, interestingly, we observe that the performance gap between the two models diminishes from +4.2 to +0.7 as the data size increases from 3K to 10K samples.

Moreover, we examine whether the superiority of DPO persists when training MLLMs with larger datasets. Although SFT exhibits a strong positive slope from 3K to 40K in Figure 4, Table M demonstrates that this trend does not hold as the data scale continues to increase. With more data, the slope becomes negative and SFT fails to catch up with DPO. In our experimental setup, all hyperparameters are fixed except for the dataset size. It is worth noting that selecting hyperparameters suited to the longer training regime can mitigate the performance degradation observed with large amounts of data; for example, adjusting the learning rate or weight decay may help.

### C.2 PERFORMANCE OF MLLMS ON ALL 16 BENCHMARKS

Full results on MLLM performance are reported in Table R, Table S, and Table T. Within our controlled setup, DPO-trained MLLMs consistently surpass their SFT-trained models across different scales of data, vision encoders, and language models. The advantage is evident on 'strongly vision-related tasks' and appears more modest on 'weakly vision-related tasks', corresponding to the experiments in Section 3

### C.3 IMAGENET CLASSIFICATION WITH A VISION ENCODER

We present additional experimental results for ImageNet classification in Figure D. For this analysis, we conduct a linear-probe evaluation using features obtained from the MLLM's vision encoder. This setup differs from the main paper's experiment in Figure 6, which utilizes visual embeddings that have passed through both the vision encoder and the projector. The results reinforce our primary findings: DPO consistently enhances visual representations more effectively than SFT, and the vision encoder's performance improves as the size of the LLM it is trained with increases.

### C.4 GRADIENT VISUALIZATION OF A VISION ENCODER

We provide additional Grad-CAM results in Figure F, where we visualize the gradients on the visual features induced by the SFT and DPO losses. The results show that MLLM post-training yields larger gradients on question-relevant image regions, with DPO providing more concentrated signals than the diffuse gradients from SFT.

| DPO Data Size | Avg. (All) | General | OCR&Chart | Vision-Cent. | Knowledge |
|---|---|---|---|---|---|
| 20K | 61.0 | 70.0 | 62.6 | 58.4 | 53.0 |
| 40K | 61.2 | **70.3** | 62.6 | **58.8** | 53.2 |
| 60K | **61.4** | 69.6 | **64.0** | 57.9 | **54.1** |
| 80K | 60.9 | 69.7 | 62.4 | 58.7 | 53.0 |
| 100K | 59.8 | 69.3 | 60.8 | 55.9 | 53.3 |
| 150K | 54.7 | 65.0 | 50.1 | 52.4 | 51.3 |
| SFT Data Size | Avg. (All) | General | OCR&Chart | Vision-Cent. | Knowledge |
| 20K | 59.4 | 68.0 | 60.9 | 56.4 | 52.4 |
| 40K | 59.5 | 68.0 | 60.8 | **56.6** | 52.7 |
| 60K | **59.6** | **68.4** | **61.1** | 55.0 | **53.9** |
| 80K | 59.1 | 67.8 | 59.5 | 55.8 | 53.1 |
| 100K | 58.1 | 67.2 | 57.7 | 54.5 | 53.1 |
| 150K | 55.7 | 66.4 | 52.3 | 52.5 | 51.8 |

Table M: **Scaling the amount of post-training data for MLLM.** We provide supplementary quantitative results corresponding to Figure C.

| Model | Train | General | | | | OCR & Chart | | | | Vision-Centric | | | | Knowledge | | | |
|---|---|---|---|---|---|---|---|---|---|---|---|---|---|---|---|---|---|
| | | $MME^P$ | MMB | $SEED^I$ | GQA | ChartQA | OCRBench | TextVQA | DocVQA | MMVP | RealWorldQA | $CV\text{-}Bench^{2D}$ | $CV\text{-}Bench^{3D}$ | $SQA^I$ | $MMMU^V$ | $MathVista^M$ | AI2D |
| QwenVL-2.5-3B | SFT | 1536.7 | 73.4 | 71.2 | 56.0 | 62.9 | 64.0 | 62.5 | 86.7 | 44.0 | 50.1 | 67.4 | 70.4 | 78.0 | 44.0 | 21.7 | 76.5 |
| QwenVL-2.5-3B | **GRPO** | 1575.8 | 77.1 | 73.3 | 59.2 | 70.4 | 67.7 | 66.8 | 88.3 | 43.0 | 58.2 | 70.6 | 74.0 | 79.3 | 46.4 | 22.6 | 78.5 |
| LLaVA-1.0-7B | SFT | 1201.1 | 39.3 | 42.5 | 32.5 | 7.1 | 21.6 | 34.8 | 43.8 | 10.9 | 31.3 | 25.0 | 39.5 | 64.8 | 27.9 | 3.1 | 45.0 |
| LLaVA-1.0-7B | **PPO** | 1230.6 | 39.5 | 43.0 | 32.2 | 15.7 | 24.8 | 34.8 | 44.8 | 15.2 | 33.5 | 25.0 | 38.4 | 64.7 | 28.1 | 4.4 | 48.4 |
| Qwen2.5-1.5B + SigLIP2-L/16 | SFT | 1420.5 | 70.9 | 71.7 | 58.5 | 62.4 | 58.9 | 64.5 | 58.0 | 44.6 | 58.2 | 63.6 | 59.3 | 86.0 | 40.2 | 12.1 | 71.1 |
| Qwen2.5-1.5B + SigLIP2-L/16 | DPO | 1485.1 | 72.4 | 72.9 | 60.3 | 65.2 | 61.4 | 64.1 | 59.5 | 51.3 | 58.8 | 63.2 | 60.2 | 86.2 | 40.8 | 14.0 | 71.1 |
| Qwen2.5-1.5B + SigLIP2-L/16 | **MPO** | 1496.7 | 72.3 | 72.4 | 61.1 | 65.2 | 60.4 | 64.1 | 59.1 | 53.3 | 59.6 | 63.3 | 62.2 | 87.0 | 41.8 | 15.3 | 72.0 |

Table N: **Full results of other RL vs. SFT.** The table presents the 16-benchmark performance for MLLMs post-trained with SFT, GRPO, PPO, and MPO.

Furthermore, the bottom two examples in Figure F correspond to a global query (like "Describe the photo in detail."). For this type of query, both DPO and SFT generate similarly distributed gradients across the entire image, a different outcome from the localized queries. As will be further discussed in Section D.6, this supports our hypothesis that the nature of the post-training data can determine how DPO enhances visual representations.

### C.5 SEGMENTATION PROBING WITH A VISION ENCODER

We provide additional qualitative results for segmentation probing in Figure G. For this experiment, a CLIP-L/14 336px vision encoder is post-trained in an MLLM with either SFT or DPO, using a Qwen2.5-3B as the base LLM. The qualitative results indicate that the DPO-trained vision encoder yields segmentation maps more consistent with the ground truth.

## D DISCUSSIONS

### D.1 UNDERSTANDING SFT AND DPO

We elaborate on the post-training techniques discussed in Section 3.1. SFT is a standard approach for equipping LLMs with instruction-following abilities (Radford et al. 2018; Touvron et al. 2023a). In our work, this involves training the MLLM $\pi_\theta$ using a maximum likelihood objective on the post-training dataset $X_{\text{PT}}$. Specifically, for each given image $I_i$ and query $q_i$, the model is optimized to maximize the probability of generating the chosen response $y_i^c$, as formulated in Equation (1). In contrast, DPO (Rafailov et al. 2023) is a prominent RL method that directly aligns the model with human preferences without requiring an explicit reward model. DPO leverages the full preference pair, including both the chosen response $y_i^c$ and the rejected response $y_i^r$. Its objective, also formulated in Equation (1), is to increase the likelihood of the chosen response while simultaneously decreasing

| MLLM = Qwen1.5-1.5B | General | | | | OCR & Chart | | | | Vision-Centric | | | | Knowledge | | | |
|---|---|---|---|---|---|---|---|---|---|---|---|---|---|---|---|---|
| | $MME^P$ | MMB | $SEED^I$ | GQA | ChartQA | OCRBench | TextVQA | DocVQA | MMVP | RealWorldQA | CV-Bench$^{2D}$ | CV-Bench$^{3D}$ | $SQA^I$ | $MMMU^V$ | MathVista$^M$ | AI2D |
| + Original SigLIP2-So/16 | 1403.0 | 65.8 | 68.9 | 59.8 | 43.5 | 39.3 | 54.0 | 49.5 | 36.0 | 55.4 | 58.2 | 52.8 | 72.9 | 38.2 | 8.8 | 64.4 |
| + Stage2-SFT | 1417.0 | 65.9 | 70.4 | 60.5 | 54.4 | 45.6 | 57.8 | 51.0 | 36.3 | 54.8 | 60.0 | 55.8 | 73.3 | 40.9 | 11.6 | 64.9 |
| + Stage2-DPO (i.e., PIVOT) | 1427.5 | 67.8 | 71.8 | 61.4 | 57.0 | 48.3 | 58.5 | 51.7 | 38.3 | 56.2 | 59.1 | 56.2 | 73.9 | 41.5 | 12.3 | 64.7 |
| + Stage2-MPO | 1438.9 | 68.3 | 71.0 | 60.3 | 56.6 | 49.1 | 59.0 | 50.5 | 39.0 | 55.2 | 60.5 | 55.6 | 72.3 | 41.4 | 13.8 | 65.5 |

Table O: **Full results under `PIVOT` evaluation.** We first post-train MLLMs using different algorithms and extract their vision encoders, which are evaluated following Section 5.1. Here, we present the full evaluation results across 16 MLLM benchmarks.

| LLM backbone | Train | Data | General | | | | Knowledge | | | | Vision-Centric | | | | OCR & Chart | | | |
|---|---|---|---|---|---|---|---|---|---|---|---|---|---|---|---|---|---|---|
| | | | $MME^P$ | MMB | $SEED^I$ | GQA | ChartQA | OCRBench | TextVQA | DocVQA | MMVP | RealWorldQA | CV-Bench$^{2D}$ | CV-Bench$^{3D}$ | $SQA^I$ | $MMMU^V$ | MathVista$^M$ | AI2D |
| Qwen2.5-1.5B | DPO | Origin | 1478.7 | 71.9 | 72.5 | 60.9 | 65.4 | 62.7 | 64.7 | 59.5 | 50.0 | 57.4 | 63.6 | 59.9 | 86.7 | 40.1 | 14.7 | 71.5 |
| Qwen2.5-1.5B | SFT | Origin | 1442.1 | 70.6 | 71.9 | 58.7 | 62.1 | 58.8 | 63.4 | 57.7 | 44.0 | 56.0 | 63.1 | 56.1 | 86.6 | 41.0 | 12.6 | 70.9 |
| Qwen2.5-1.5B | DPO | NoThink | 1441.4 | 72.3 | 72.7 | 60.3 | 64.6 | 61.8 | 63.8 | 58.9 | 47.3 | 59.0 | 64.9 | 62.0 | 86.1 | 40.1 | 15.1 | 71.3 |
| Qwen2.5-1.5B | SFT | NoThink | 1471.6 | 71.4 | 71.7 | 59.1 | 62.6 | 58.8 | 64.8 | 56.0 | 46.7 | 58.8 | 63.3 | 56.3 | 85.5 | 41.0 | 13.1 | 70.1 |
| Qwen2.5-1.5B | DPO | Under30 | 1469.8 | 72.2 | 72.3 | 60.3 | 59.8 | 61.7 | 62.7 | 55.8 | 47.3 | 57.9 | 63.1 | 61.4 | 86.2 | 40.8 | 15.1 | 71.0 |
| Qwen2.5-1.5B | SFT | Under30 | 1486.6 | 70.3 | 71.8 | 59.6 | 59.7 | 57.8 | 64.2 | 55.1 | 45.3 | 57.4 | 62.9 | 59.0 | 85.9 | 40.0 | 15.9 | 69.2 |
| Qwen2.5-3B | DPO | Origin | 1553.1 | 76.2 | 74.4 | 61.5 | 67.6 | 62.2 | 64.4 | 69.9 | 52.0 | 59.0 | 67.2 | 69.3 | 87.0 | 42.6 | 15.7 | 75.9 |
| Qwen2.5-3B | SFT | Origin | 1550.0 | 75.2 | 73.7 | 59.3 | 58.5 | 60.1 | 65.1 | 67.1 | 47.5 | 59.0 | 66.3 | 65.8 | 86.8 | 42.3 | 15.9 | 75.9 |
| Qwen2.5-3B | DPO | NoThink | 1511.1 | 76.6 | 73.4 | 61.7 | 64.9 | 61.2 | 64.5 | 65.8 | 52.7 | 60.0 | 67.3 | 70.0 | 88.7 | 43.6 | 14.4 | 75.7 |
| Qwen2.5-3B | SFT | NoThink | 1537.8 | 75.9 | 74.1 | 59.6 | 64.6 | 60.4 | 66.1 | 64.1 | 49.3 | 57.8 | 66.1 | 66.7 | 87.4 | 43.1 | 18.7 | 75.6 |
| Qwen2.5-3B | DPO | Under30 | 1480.2 | 76.1 | 74.1 | 61.4 | 65.2 | 62.5 | 62.9 | 65.9 | 53.3 | 57.8 | 66.3 | 68.9 | 87.6 | 42.4 | 15.1 | 75.0 |
| Qwen2.5-3B | SFT | Under30 | 1509.2 | 75.2 | 73.7 | 59.5 | 65.7 | 59.4 | 65.8 | 65.1 | 50.0 | 57.1 | 65.4 | 65.2 | 88.3 | 43.1 | 17.1 | 74.7 |

Table P: **Full results under SFT-friendly datasets.** We provide full evaluation results on 16 MLLM benchmarks.

that of the rejected one, relative to a reference policy $\pi_{\text{ref}}$, which is typically the initial model before preference alignment.

## D.2 LOW PERFORMANCE GAP ON KNOWLEDGE VQA

As we observe in Section 3, DPO shows a clear advantage over SFT on strongly vision-related tasks, but this performance gap diminishes for Knowledge VQA. This suggests that for knowledge-intensive tasks, leveraging the rejected responses $y_i^r$ provides a less significant benefit compared to the standard SFT approach. We hypothesize that for problems in domains like science and math, the chosen responses $y_i^c$ may already contain sufficient factual knowledge, making the comparative signal from $y_i^r$ less critical. The interplay between preference data characteristics and task domains is a valuable direction for future research.

## D.3 POSITIONING OUR WORK WITHIN RL-BASED MLLM RESEARCH

Prior work has reported performance gains when RL is applied to an MLLM pretrained with SFT (Yu et al. 2024; Wang et al. 2024a; Liao et al. 2025; Fu et al. 2025a). MPO (Wang et al. 2024b) illustrated this by applying DPO to an SFT-trained InternVL model, and RLAIF-V (Yu et al. 2025) similarly compared SFT-trained LLaVA with its DPO-trained variant. Despite this evidence of RL's effectiveness, SFT has remained the dominant training strategy in recent MLLM development (Tong et al. 2024a; Han et al. 2025; Fan et al. 2025). Our study aims to strengthen the RL-based MLLM literature by demonstrating the effectiveness of RL in the post-training stage and motivating its wider adoption in future research. To this end, we design a more fair setup to clearly demonstrate the effect of RL. Rather than comparing a model trained only with SFT to a model trained with SFT followed by DPO, we compare a model trained with SFT in *Stage 1* and DPO in *Stage 2* to a model trained

| Model | LLM size | General | | | | OCR & Chart | | | | Vision-Centric | | | | Knowledge | | | |
|---|---|---|---|---|---|---|---|---|---|---|---|---|---|---|---|---|---|
| | | $MME^P$ | MMB | $SEED^I$ | GQA | ChartQA | OCRBench | TextVQA | DocVQA | MMVP | RealWorldQA | $CV\text{-}Bench^{2D}$ | $CV\text{-}Bench^{3D}$ | $SQA^I$ | $MMMU^V$ | $MathVista^M$ | AI2D |
| BLIP-2 | 7B | 1333.7 | 52.4 | 17.6 | 57.3 | 58.8 | 43.7 | 60.1 | 48.5 | 23.3 | 42.4 | 26.9 | 31.5 | 83.0 | 26.8 | 11.3 | 48.8 |
| SPHINX | 7B | 1515.8 | 57.6 | 68.0 | 61.1 | 37.8 | 29.8 | 38.3 | 26.8 | 38.7 | 48.5 | 61.3 | 61.7 | 68.6 | 31.6 | 9.8 | 55.6 |
| LLaVA-Next-Vincina | 7B | 1504.2 | 66.3 | 68.3 | 63.2 | 18.5 | 33.6 | 61.0 | 38.9 | 30.7 | 55.3 | 62.2 | 62.9 | 72.3 | 36.1 | 7.9 | 59.2 |
| LLaVA-Next-LLaMA-3 | 8B | 1526.0 | 71.0 | 72.6 | 65.3 | 69.2 | 58.1 | 64.7 | 65.1 | 40.0 | 60.1 | 62.1 | 67.7 | 72.9 | 39.6 | 11.8 | 71.6 |
| LLaVA-Next-Qwen2 | 7B | 1453.3 | 72.2 | 73.6 | 63.6 | 67.1 | 54.2 | 63.6 | 68.2 | 49.3 | 60.0 | 64.4 | 75.7 | 73.5 | 36.1 | 30.0 | 73.7 |
| SPHINX-Tiny | 1.1B | 1223.8 | 47.1 | 57.7 | 50.8 | 10.9 | 18.8 | 27.0 | 20.6 | 14.0 | 47.6 | 57.2 | 55.6 | 63.0 | 27.1 | 1.7 | 41.1 |
| LLaVA-OneVision | 0.5B | 1274.9 | 54.9 | 63.3 | 59.5 | 60.9 | 60.4 | 50.9 | 55.8 | 18.7 | 54.1 | 43.8 | 54.3 | 67.5 | 31.0 | 4.8 | 54.3 |
| Ours (SigLIP1-So/16)+PIVOT | 1.5B | 1446.7 | 68.7 | 70.0 | 59.8 | 73.8 | 38.2 | 9.2 | 65.2 | 46.6 | 36.6 | 54.7 | 49.4 | 36.0 | 55.8 | 57.3 | 57.5 |
| Ours (SigLIP2-So/16)+PIVOT | 1.5B | 1427.5 | 67.8 | 71.8 | 61.4 | 57.0 | 48.3 | 58.5 | 51.7 | 38.3 | 56.2 | 59.1 | 56.2 | 73.9 | 41.5 | 12.3 | 64.7 |

Table Q: **Full results of PIVOT-based MLLMs and vs. other MLLMs.** The table presents the 16-benchmark performance for diverse MLLMs.

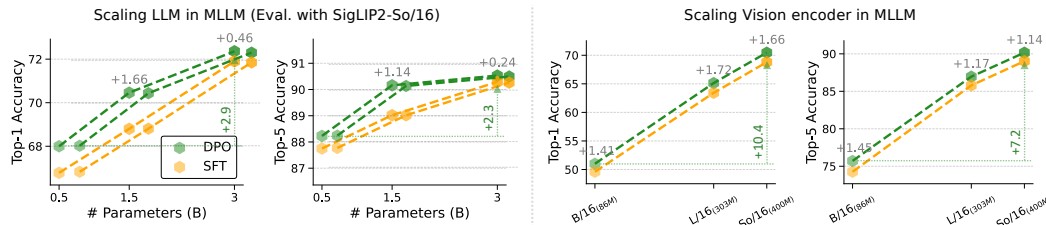

Figure D: **ImageNet classification.** We train MLLMs with different post-training strategies while scaling either the vision encoder (with a fixed Qwen2.5-1.5B) or the LLM (with a fixed SigLIP2-So/16). We utilize features extracted from the MLLM's vision encoder (*i.e.*, SigLIP2-So/16). Note that the features used in Figure 6 are the outputs of the vision encoder and multimodal projector, which are directly used as the LLM's visual embeddings.

with SFT in both stages. Even under this comparison, we find that DPO remains advantageous, and we hope this result encourages further attention to RL-based approaches in MLLM research.

### D.4 PIVOT'S SIGNIFICANCE

PIVOT is a derivative contribution grounded in our analytical findings. Although its design is simple, we view it as an under-explored training regime. Beyond our contributions outlined in the second paragraph of Section 5.1, PIVOT experiments shows that DPO can be effective for training vision encoders, extending prior observations that SFT-based language alignment benefits visual representations (Bolya et al. 2025). Our results further reveal that even state-of-the-art encoders such as SigLIP2, despite large-scale pre-training, retain room for improvement within MLLMs. These insights are valuable for exploring new RL algorithms for MLLMs, and we plan to pursue this direction in future work.

### D.5 LIMITATIONS

Despite our efforts in Section 3.1 to design the experimental setup as fairly as possible, our analysis still has limitations. From a data-scale perspective, each DPO update uses $(I_t, x_t, y_t^c, y_t^r)$, whereas SFT relies only on $(I_t, x_t, y_t^c)$, so the amount of supervision per iteration is not perfectly matched. From a data-type perspective, our comparison relies on preference pairs sourced from the MPO dataset, which may be more favorable to DPO than to SFT. To mitigate these concerns, we make several efforts. Figure 4 shows that even when DPO is limited to fewer samples, it performs strongly, indicating a data-efficiency benefit rather than a data-volume one. In addition, analyses in Section B.3 and Section B.4 examine the impact of different data types and reveal that DPO retains its advantage across these variations. Despite these efforts, further investigation is necessary. A more balanced comparison may require extending SFT to also incorporate negative responses, and it would be valuable to determine whether certain data distributions preferentially benefit SFT or DPO and to identify the properties that underlie such differences.

| Model | Train | Average | General | | | | Knowledge | | | | Vision-Centric | | | | OCR & Chart | | | |
|---|---|---|---|---|---|---|---|---|---|---|---|---|---|---|---|---|---|---|
| | | | $MME^P$ | MMB | $SEED^I$ | GQA | $SQA^I$ | $MMMU^V$ | $MathVista^M$ | AI2D | MMVP | RealWorldQA | $CV\text{-}Bench^{2D}$ | $CV\text{-}Bench^{3D}$ | ChartQA | OCRBench | TextVQA | DocVQA |
| Qwen2.5-3B+SigLIP2-B/16 | DPO | 59.7 | 1438.4 | 72.4 | 72.7 | 60.1 | 83.7 | 44.3 | 15.7 | 73.3 | 45.3 | 56.9 | 64.5 | 65.5 | 60.5 | 49.3 | 59.2 | 59.3 |
| Qwen2.5-3B+SigLIP2-B/16 | SFT | 58.3 | 1509.4 | 72.2 | 71.8 | 57.1 | 83.6 | 42.8 | 15.9 | 73.5 | 42.7 | 55.3 | 63.7 | 60.0 | 57.2 | 47.0 | 57.1 | 57.2 |
| Qwen2.5-3B+SigLIP2-L/16 | DPO | 62.6 | 1498.1 | 75.8 | 73.6 | 61.3 | 85.6 | 42.3 | 15.8 | 74.8 | 48.7 | 58.0 | 66.7 | 69.3 | 65.4 | 58.8 | 64.2 | 66.8 |
| Qwen2.5-3B+SigLIP2-L/16 | SFT | 60.8 | 1547.8 | 74.5 | 73.0 | 58.3 | 85.5 | 42.0 | 15.5 | 74.5 | 43.3 | 57.6 | 66.7 | 65.3 | 55.1 | 56.8 | 62.4 | 64.1 |
| Qwen2.5-3B+SigLIP2-So/16 | DPO | 63.9 | 1553.1 | 76.2 | 74.4 | 61.5 | 87.0 | 42.6 | 15.7 | 75.9 | 52.0 | 59.0 | 67.2 | 69.3 | 67.6 | 62.2 | 64.4 | 69.9 |
| Qwen2.5-3B+SigLIP2-So/16 | SFT | 62.3 | 1550.0 | 75.2 | 73.7 | 59.3 | 86.8 | 42.3 | 15.9 | 75.9 | 47.5 | 59.0 | 66.3 | 65.8 | 58.5 | 60.1 | 65.1 | 67.1 |
| Qwen2.5-3B+SigLIP2-g/16 | DPO | 64.8 | 1548.6 | 77.6 | 75.0 | 62.5 | 87.9 | 43.1 | 17.2 | 75.4 | 52.0 | 59.1 | 68.8 | 70.2 | 66.0 | 65.2 | 67.0 | 72.3 |
| Qwen2.5-3B+SigLIP2-g/16 | SFT | 62.9 | 1558.0 | 75.7 | 74.4 | 60.4 | 86.9 | 43.8 | 17.3 | 75.3 | 48.0 | 59.9 | 67.5 | 65.1 | 57.6 | 63.8 | 65.6 | 67.0 |

Table R: **Scaling the vision encoder in MLLMs.** We analyze the impact of the vision encoder sizes, ranging from 86M (SigLIP-B/16) to 1B (SigLIP-g/16) parameters, in Qwen2.5-3B combined with SigLIP2.

| Model | Train | Average | General | | | | OCR & Chart | | | | Vision-Centric | | | | Knowledge | | | |
|---|---|---|---|---|---|---|---|---|---|---|---|---|---|---|---|---|---|---|
| | | | $MME^P$ | MMB | $SEED^I$ | GQA | ChartQA | OCRBench | TextVQA | DocVQA | MMVP | RealWorldQA | $CV\text{-}Bench^{2D}$ | $CV\text{-}Bench^{3D}$ | $SQA^I$ | $MMMU^V$ | $MathVista^M$ | AI2D |
| Qwen2.5-0.5B+SigLIP2-So/16 | DPO | 51.5 | 1167.7 | 58.1 | 65.7 | 55.9 | 76.2 | 34.1 | 10.8 | 59.7 | 28.0 | 52.5 | 52.3 | 56.5 | 59.1 | 56.2 | 54.8 | 46.1 |
| Qwen2.5-0.5B+SigLIP2-So/16 | SFT | 49.5 | 1170.9 | 55.2 | 63.1 | 55.2 | 75.1 | 33.7 | 10.2 | 59.1 | 25.3 | 50.5 | 45.3 | 49.5 | 57.1 | 54.3 | 54.5 | 45.3 |
| Qwen2.5-1.5B+SigLIP2-So/16 | DPO | 61.0 | 1478.7 | 71.9 | 72.5 | 60.9 | 86.7 | 40.1 | 14.7 | 71.5 | 50.0 | 57.4 | 63.6 | 59.9 | 65.4 | 62.7 | 64.7 | 59.5 |
| Qwen2.5-1.5B+SigLIP2-So/16 | SFT | 59.1 | 1442.1 | 70.6 | 71.9 | 58.7 | 86.6 | 41.0 | 12.6 | 70.9 | 44.0 | 56.0 | 63.1 | 56.1 | 62.1 | 58.8 | 63.4 | 57.7 |
| Qwen2.5-3B+SigLIP2-So/16 | DPO | 63.9 | 1553.1 | 76.2 | 74.4 | 61.5 | 87.0 | 42.6 | 15.7 | 75.9 | 52.0 | 59.0 | 67.2 | 69.3 | 67.6 | 62.2 | 64.4 | 69.9 |
| Qwen2.5-3B+SigLIP2-So/16 | SFT | 62.3 | 1550.0 | 75.2 | 73.7 | 59.3 | 86.8 | 42.3 | 15.9 | 75.9 | 47.5 | 59.0 | 66.3 | 65.8 | 58.5 | 60.1 | 65.1 | 67.1 |
| Qwen2.5-7B+SigLIP2-So/16 | DPO | 68.9 | 1664.0 | 80.3 | 76.0 | 64.0 | 92.4 | 50.2 | 20.4 | 80.7 | 59.3 | 62.2 | 73.0 | 75.8 | 74.2 | 65.6 | 71.1 | 73.5 |
| Qwen2.5-7B+SigLIP2-So/16 | SFT | 66.2 | 1627.7 | 78.6 | 74.9 | 59.9 | 91.8 | 48.7 | 18.7 | 79.9 | 46.0 | 63.5 | 71.9 | 72.3 | 68.8 | 64.0 | 69.0 | 70.0 |

Table S: **Scaling the language model in MLLMs.** Using SigLIP2-So/16 as the vision encoder, we vary the size of the language model (Qwen2.5) and evaluate performance across multiple benchmarks.

## D.6 FUTURE WORK

Beyond the broader impact discussed in Section 6, our study opens several additional avenues for research. While our work primarily utilizes the LLaVA Li et al. (2025a) framework with a Qwen2.5 Yang et al. (2025a) backbone, a natural extension is to investigate if our findings generalize to other MLLM architectures, such as InternVL Chen et al. (2024) and Qwen-VL Bai et al. (2023a), or when using different LLM backbones like LLaMA Dubey et al. (2024) and Gemma Team et al. (2024b). Another promising direction involves exploring whether novel dataset formats could be designed to better leverage DPO for learning stronger visual representations. We have **a particular interest** in this direction and plan to actively pursue it as part of our future work. Furthermore, expanding the evaluation beyond the 16 benchmarks from Cambrian to include traditional hallucination benchmarks Wang et al. (2023); Sun et al. (2024b) could provide deeper insights into the comparative performance of DPO and SFT.

## E EXPERIMENTAL DETAILS

### E.1 PRE-TRAINING & POST-TRAINING

We describe in detail the training strategies of the models used in Section 3. We build our models using the LLaVA-OneVision source code[5]. Our experiments utilize four scales of the SigLIP2 vision encoder (`google/SigLIP2-B/16-patch16-384`, `google/SigLIP2-L/16-patch16-384`, `google/siglip2-So/16-patch16-384`, `google/SigLIP2-g/16-opt-patch16-384`) and four versions of the Qwen2.5-Instruct LLM (`Qwen/Qwen2.5-0.5B-Instruct`, `Qwen/Qwen2.5-1.5B-Instruct`, `Qwen/Qwen2.5-3B-Instruct`, `Qwen/ Qwen2.5-7B-Instruct`), which are connected by a 2-layer MLP projector following the default implementation.

---

[5] https://github.com/LLaVA-VL/LLaVA-NeXT

| | | | General | | | | Knowledge | | | Vision-Centric | | | | OCR & Chart | | | |
|---|---|---|---|---|---|---|---|---|---|---|---|---|---|---|---|---|---|
| Data size | Train | Average | $MME^P$ | MMB | $SEED^I$ | GQA | $SQA^I$ | $MMMU^V$ | $MathVista^M$ | AI2D | MMVP | RealWorldQA | $CV\text{-}Bench^{2D}$ | $CV\text{-}Bench^{3D}$ | ChartQA | OCRBench | TextVQA | DocVQA |
| 3K | DPO | 60.4 | 1490.4 | 72.0 | 72.5 | 60.7 | 86.2 | 40.4 | 15.3 | 71.3 | 46.7 | 56.2 | 62.6 | 57.2 | 65.2 | 61.7 | 64.6 | 58.8 |
| 3K | SFT | 56.4 | 1431.0 | 70.0 | 70.9 | 58.0 | 86.8 | 33.8 | 8.5 | 67.2 | 40.7 | 58.8 | 52.7 | 41.7 | 63.2 | 59.5 | 63.1 | 56.5 |
| 5K | DPO | 60.6 | 1486.3 | 72.2 | 72.4 | 60.7 | 86.0 | 40.6 | 15.5 | 71.0 | 47.3 | 57.1 | 63.2 | 59.0 | 65.6 | 61.9 | 64.4 | 59.0 |
| 5K | SFT | 57.6 | 1409.4 | 70.5 | 71.3 | 58.2 | 86.6 | 39.6 | 8.7 | 70.7 | 38.7 | 59.1 | 62.6 | 42.7 | 63.2 | 59.3 | 63.1 | 56.6 |
| 10K | DPO | 60.8 | 1480.1 | 72.0 | 72.7 | 60.3 | 86.1 | 40.6 | 14.8 | 71.0 | 49.0 | 57.0 | 63.2 | 60.2 | 65.6 | 62.2 | 64.5 | 59.5 |
| 10K | SFT | 58.9 | 1431.5 | 70.9 | 72.1 | 58.9 | 86.0 | 40.2 | 12.5 | 71.1 | 44.6 | 57.9 | 63.6 | 49.3 | 62.4 | 58.1 | 64.5 | 58.0 |
| 20K | DPO | 61.0 | 1478.7 | 71.9 | 72.5 | 60.9 | 86.7 | 40.1 | 14.7 | 71.5 | 50.0 | 57.4 | 63.6 | 59.9 | 65.4 | 62.7 | 64.7 | 59.5 |
| 20K | SFT | 59.1 | 1442.1 | 70.6 | 71.9 | 58.7 | 86.6 | 41.0 | 12.6 | 70.9 | 44.0 | 56.0 | 63.1 | 56.1 | 62.1 | 58.8 | 63.4 | 57.7 |
| 40K | DPO | 61.3 | 1495.4 | 72.2 | 73.2 | 60.9 | 86.4 | 39.0 | 15.6 | 71.7 | 51.9 | 59.0 | 63.0 | 61.3 | 64.5 | 62.6 | 64.8 | 59.7 |
| 40K | SFT | 59.5 | 1423.7 | 70.5 | 72.1 | 58.3 | 85.9 | 40.0 | 13.5 | 71.3 | 45.3 | 57.9 | 64.1 | 59.1 | 63.2 | 57.4 | 64.3 | 58.3 |

Table T: **Scaling data on MLLM performance.** We vary the size of training data for an MLLM built with Qwen2.5-1.5B and SigLIP2-So/16 and measure its performance.

For the training data, we use the BLIP_LAION_CC_SBU_558k dataset[6] for projector-only pretraining and the LLaVA-OneVision-Data-Single 3.2M dataset[7] for Stage 1 pretraining. For Stage 2 post-training, we use a 20K subset randomly sampled from the MMPR-1.2 dataset[8].

The hyperparameters for *Stage 1* are adopted from the standard LLaVA-OneVision finetuning script[9], including a learning rate of $1 \times 10^{-5}$ and a batch size of 256. For *Stage 2* DPO post-training, we largely follow the corresponding script[10] but adjust the learning rate (LR) to $1 \times 10^{-6}$ and use a batch size of 256 for our data scale. To ensure a controlled comparison for *Stage 2* SFT post-training, we use the same finetuning script with a learning rate of $5 \times 10^{-6}$, but remove the vision-encoder-specific LR, mirroring the DPO setup.

Since SFT and DPO rely on fundamentally different loss formulations, their optimal learning rates naturally diverge. In practice, we observe that DPO requires substantially smaller LRs than SFT, partly because DPO accounts for both chosen and rejected responses, effectively doubling the batch size per iteration compared to SFT. This observation aligns with prior settings, such as those in InternVL2.5 (Wang et al. 2024b), where an LR of $2 \times 10^{-7}$ is used for DPO and $4 \times 10^{-5}$ for SFT.

### E.2 EVALUATION BENCHMARKS

As stated in Section 3.1, we adopt the evaluation suite from Cambrian (Tong et al. 2024a) for a comprehensive assessment of MLLM performance. This suite consists of 16 benchmarks categorized into four domains: General, Knowledge, OCR & Chart, and Vision-Centric VQA. A list of these benchmarks, along with their domain assignments and citations, is provided in Table U. Unlike other benchmarks whose scores generally range from 0 to 100, MME produces values on a 0–2000 scale. To ensure comparability within the overall MLLM evaluation, when computing the average score, we rescale the MME results by a factor of 20. We utilize the Cambrian source code, except in the case of DocVQA (Mathew et al. 2021). The Cambrian implementation of DocVQA does not yield numeric outputs automatically; rather, it requires manual submission of result CSV files to the evaluation website. To streamline this process, we employ the lmms-eval (Zhang et al. 2024) source code to obtain DocVQA scores.

---

[6] https://huggingface.co/datasets/liuhaotian/LLaVA-Pretrain/blob/main/blip_laion_cc_sbu_558k.json
[7] https://huggingface.co/datasets/lmms-lab/LLaVA-OneVision-Data
[8] https://huggingface.co/datasets/OpenGVLab/MMPR-v1.2
[9] https://github.com/LLaVA-VL/LLaVA-NeXT/blob/main/scripts/train/finetune_si.sh
[10] https://github.com/LLaVA-VL/LLaVA-NeXT/blob/main/scripts/train/dpo.sh

| Benchmark | Task | Domain | Citation |
|---|---|---|---|
| GQA | all | General VQA | Hudson & Manning (2019) |
| SEED | image-based | General VQA | Ge et al. (2023) |
| MME | perception | General VQA | Fu et al. (2023) |
| MMBench | all | General VQA | Liu et al. (2024b) |
| AI2D | all | Knowledge VQA | Hiippala et al. (2021) |
| ScienceQA | image-based | Knowledge VQA | Lu et al. (2022) |
| MathVista | math | Knowledge VQA | Lu et al. (2023) |
| MMMU | vision | Knowledge VQA | Yue et al. (2024) |
| TextVQA | all | OCR & Chart VQA | Singh et al. (2019) |
| DocVQA | all | OCR & Chart VQA | Mathew et al. (2021) |
| ChartQA | all | OCR & Chart VQA | Masry et al. (2022) |
| OCRBench | all | OCR & Chart VQA | Liu et al. (2023b) |
| MMVP | all | Vision-Centric VQA | Tong et al. (2024b) |
| RealWorldQA | all | Vision-Centric VQA | xAI (2024) |
| CVBench-2D | all | Vision-Centric VQA | Tong et al. (2024a) |
| CVBench-3D | all | Vision-Centric VQA | Tong et al. (2024a) |

Table U: **List of benchmarks used.** To evaluate MLLLMs, we used 16 benchmarks that are assigned to each of the domains proposed in Cambrian (Tong et al. 2024a).

### E.3 IMAGENET CLASSIFICATION

This section details the protocol for the ImageNet classification experiment presented in Section 4. Our approach is based on the linear probe evaluation from the official OpenAI-CLIP repository[11]. As recommended in their public issue[12], we freeze the feature extractor and train a scikit-learn Logistic Regression model with L2 regularization, sweeping over lambda values for a maximum of 1000 iterations. Since evaluating on the full 1M ImageNet dataset is time-consuming, we follow the practice discussed in the community[13] and perform validation on a 50k random subset of the ImageNet data for early-stage validation. In addition, we implement a prototype-based linear classifier for more rapid validation; this is achieved by averaging the features of each class to form the weights of a linear layer. We verify that this faster method yields similar performance trends to the standard Logistic Regression approach.

### E.4 GRAD-CAM

We present here the experimental details for the gradient visualization in Section 4. We construct a training pipeline using *a single sample* and visualize the gradients around the 20th step. This setup alleviates the issue where the cosine learning rate scheduler sets the initial learning rate near zero and produces uninformative gradients at very early steps in the original LLaVA-OneVision code. By focusing on this step range, we obtain meaningful gradient patterns.

### E.5 SEMANTIC SEGMENTATION

We describe here the experimental details for the semantic segmentation study in Section 4. The setup follows the implementation referenced in the codebase of prior work (Covert et al. 2025)[14]. Specifically, we freeze the vision encoder and attach a two-layer MLP head, which is trained on the ADE20K dataset (Zhou et al. 2017). Evaluation is conducted on the validation set, where segmentation is performed at the patch level and recall is used as the primary metric. The training procedure follows the default configuration of the referenced repository, including 5 training epochs and a learning rate of $1 \times 10^{-3}$.

---

[11] https://github.com/openai/CLIP?tab=readme-ov-file#linear-probe-evaluation

[12] https://github.com/openai/CLIP/issues/39#issuecomment-778034767

[13] https://github.com/openai/CLIP/issues/64#issuecomment-804444364

[14] https://github.com/iancovert/patch-seg/tree/main?tab=readme-ov-file

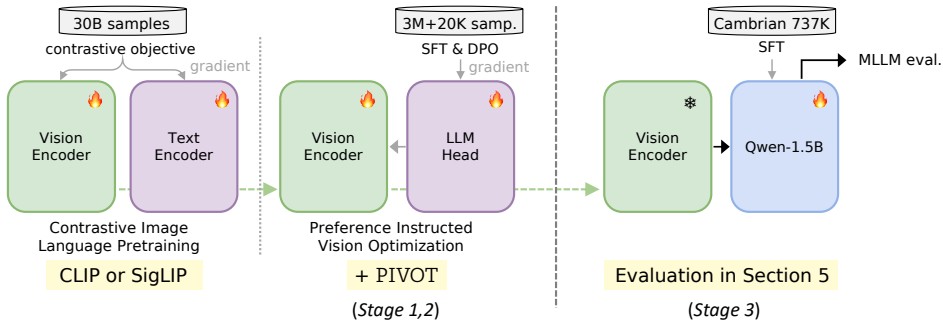

Figure E: We provide additional and more detailed illustrations of the experimental setup from Section 5, in addition to figure 11. More details can be found in Section 5.1. The process begins with contrastive pretraining of the vision encoder using CLIP or SigLIP on large-scale image–text data. Next, the vision encoder is paired with an LLM head and optimized through preference-instructed finetuning (SFT and DPO) with 3M+20K samples (*i.e.*, post-training in Section 3.1). Finally, the tuned vision encoder is frozen and evaluated in an MLLM setting by coupling it with Qwen-1.5B and finetuning on the Cambrian 737K dataset. This setup parallels prior evaluation protocols such as Cambrian (Tong et al. 2024a), DINO-MLLM (Fan et al. 2025), and MLLM-data (Han et al. 2025) and allows direct assessment of the standalone usefulness of vision representations within MLLMs.

### E.6 REPRESENTATION ALIGNMENT

In Section 4, we present results measured against five reference LLMs, including Gemma-2B/7B (Team et al. 2024b), LLaMA-3-8B (Dubey et al. 2024), and Mixtral-8x7B (Jung et al. 2010) and Bloomz-7B (Muennighoff et al. 1786). The vision models under analysis are vision encoders trained within MLLM frameworks alongside three different sizes of LLMs. We evaluate alignment between our vision encoders and the reference LLMs using the implementation provided in the Platonic Representation repository[15]. Scores are computed on the 'minhuh/prh' dataset distributed with the repository. Since this dataset contains only 1,024 examples, the results exhibit variability. To address this, we evaluate vision encoders trained with three different random seeds and report the averaged performance

### E.7 PIVOT-ENHANCED VISION MODEL EVALUATION

To evaluate the effectiveness of PIVOT-enhanced vision models within MLLMs, we follow the pipeline illustrated in Figure E. The middle part of the figure corresponds to the training strategy described in Section 3.1 and Section E.1. In the rightmost part of the figure (*i.e.*, Stage 3), the model is finetuned on a new dataset using the configuration provided in the LLaVA-NeXT repository[16]. This setup includes a batch size of 256, a learning rate of $1 \times 10^{-5}$, and other default hyperparameters.

---

[15]https://github.com/minyoungg/platonic-rep
[16]https://github.com/LLaVA-VL/LLaVA-NeXT/blob/main/scripts/train/finetune_si.sh

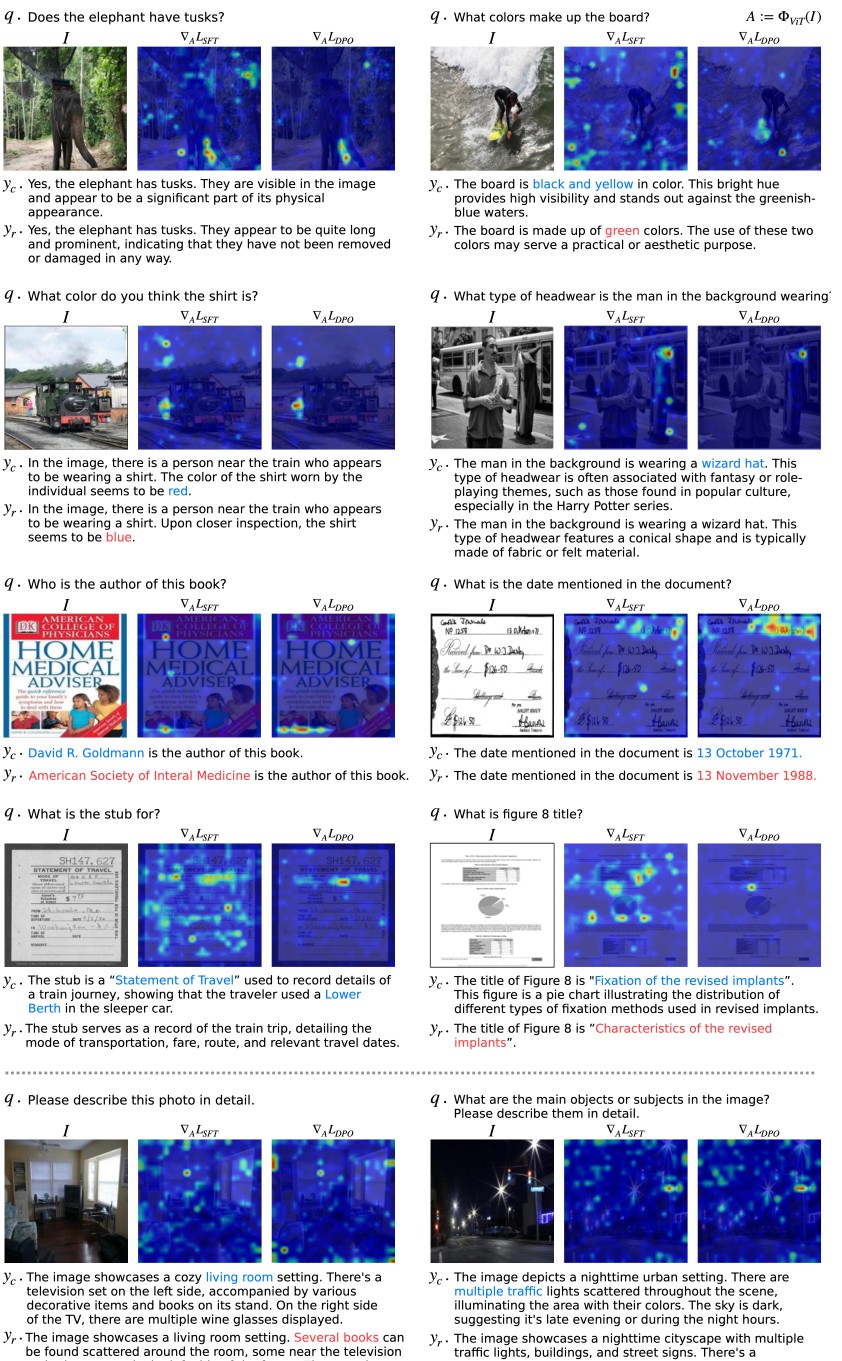

Figure F: **Additional results of Grad-CAM.** We provide additional experimental results of Figure 7, where we illustrate the gradients received by the vision encoder under MLLM post-training approaches, DPO and SFT, using Grad-CAM (Selvaraju et al. 2017).

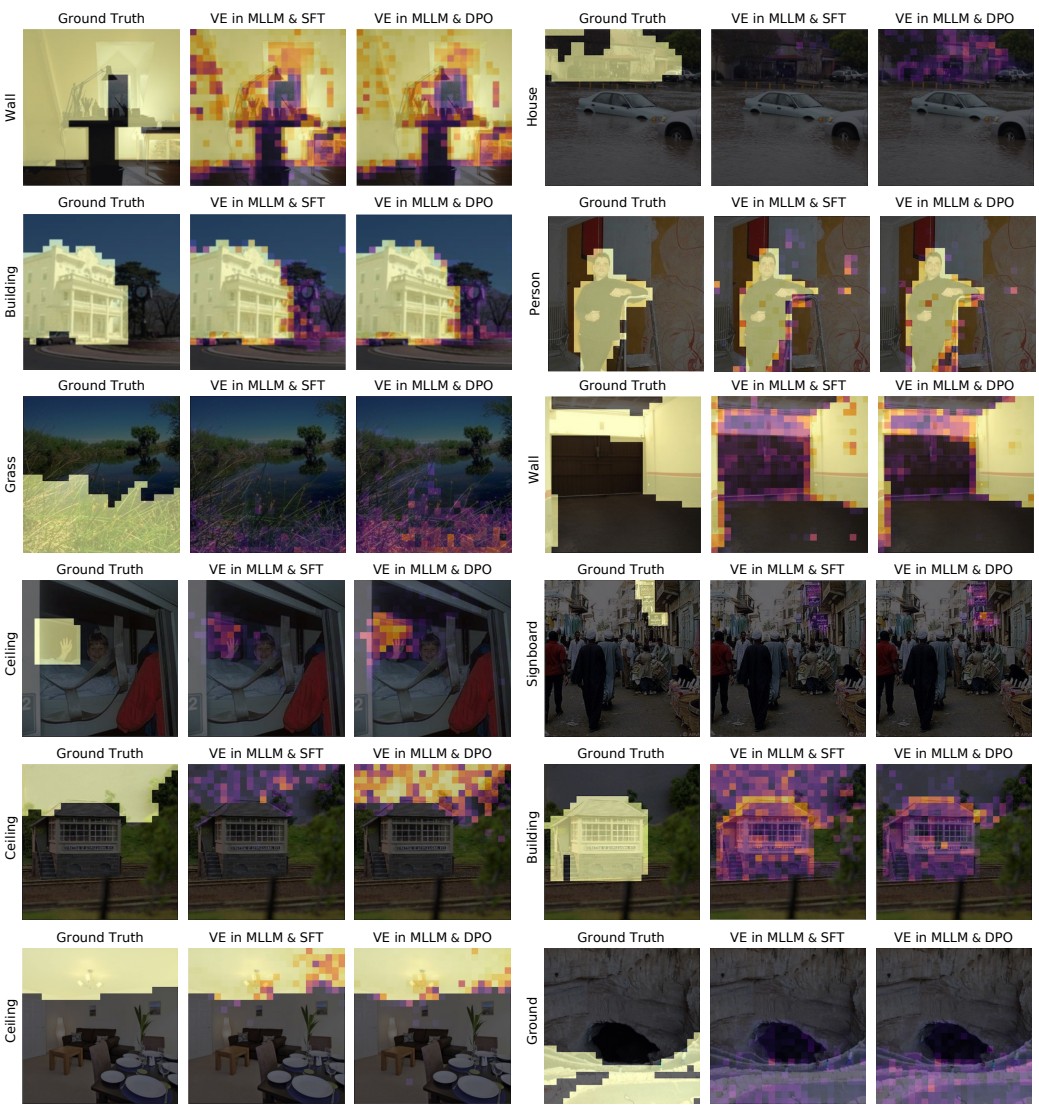

Figure G: **Qualitative results on segmentation probing.** We study segmentation probing of a vision encoder (VE), CLIP-L/14 336px, post-trained in an MLLM with SFT and DPO, where Qwen2.5-3B is a base LLM. The DPO-trained ViT yields segmentation maps consistent with the ground truth, unlike the broader maps from the SFT-trained model.

