# OpenReview forum: "RL makes MLLMs see better than SFT"
_ICLR.cc/2026/Conference — ICLR 2026 Poster_

### Official Review · Reviewer_uppE · 2025-10-26

**Soundness:** 3
**Presentation:** 4
**Contribution:** 3
**Rating:** 8
**Confidence:** 4

**Summary:**

This work studies the impact of the RL technique DPO (Direct Preference Optimization) on the training of MLLM. The finding of the paper can be summarized on DPO being more effective than standard SFT when applied to a proper set of multimodal instruction tuning data containing both a positive and a negative response in text. The authors provide an extensive set of experimental analysis supporting their claims and propose a method called PIVOT to “warm up” visual encoders to be plugged into MLLM and boost their performance. The method boils down to pre-train them with a smaller LLM head using SFT followed by DPO. Only at that point the visual encoder is plugged in the bigger scale LLM to create the final MLLM. This provides a significant boost in performance, in the case where the visual encoder is kept frozen while training the MLLM (ala LLava), and a modest one when everything is trained end-to-end.

**Strengths:**

+ **Presentation:** I really appreciated the presentation in this paper. The work follows a very clear storyline and make an extremelly good case on why the proposed experiments make sense and on how to interpret the results. I found the reading very enjoyable and I would consider this a very good example on how to write a paper in an interesting way.

+ **Novelty:** To the best of my knowledge the effect of SFT vs RL for MLLM is still a quite under-explored area. This work does a nice job at taking a stab at it and starting to characterize the difference among the two. I particularly enjoyed the emphasis not only on the final performance, but also on the effect of the different strategies on the learned representations (e.g., the GRADCam experiments)

+ **Extensive experimental analysys:** The author run a significant number of experiments to explore their hypothesis and support their claim. Most experiments that came to my mind while reading the work were discussed in a subsequent part of the paper. After going through the main paper and supplementary I’m left with almost no doubt or question.

**Weaknesses:**

## Major

I haven’t found major weaknesses in this work.

## Minor

a. **Data advantage for DPO:** The comparison between SFT and DPO is not completely fair since the data used for the comparison are amenable to DPO training that per its own nature will use also negative answers in its optimization process, while SFT will not. This brings to two possible data advantage for DPO:
1. *Data scale:* De facto DPO is technically “seeing” more examples/information within one training run. From this perspective it is not surprising that it is more effective than SFT, because it can rely on the negative examples to pull the optimization away from that path, while SFT cannot. At the same time there isn’t a practical way that comes to my mind for SFT to leverage in any way the negative answers, therefore I don't have a better suggestion on how to compare the two.  Fig. 4 also shows that DPO is vastly more data efficient, hence this argument of “DPO” seeing more data is not the full picture.
2. *Data type:* The data used for Stage 2 are explicitly created to not have a clear ground truth but being more open, i.e., perfect for DPO suboptimal for SFT. SFT on these data will force the model to try to collapse its distribution to only one of the possible many options which are all good answers. DPO instead will only try to rank the good answer above the bad one in terms of likelihood. It would have been interesting to repeat the comparison between SFT and DPO on a more SFT friendly distribution of data (i.e., data with a clear answer). This would help to understand wheter the same findings hold also there or if the story of the paper should be adapted in “pick the best losses for the type of data you have”.

B. **PIVOT contributions/results being weak:** I found the PIVOT section of the paper significantly weaker than the rest. The core issue that I have is that in Tab. 1 the authors compare a visual encoder that has been “warmed up” to work good with a LLM and later kept frozen for the final part of the training to a visual encoder that has never been jointly trained with a LLM. It is not surprisingly that PIVOT encoders obtain significantly higher performance compared to their base model. It is still interesting that lines with +PIVOT are better than lines with +SFT, but this is just the storyline of the rest of the paper. This is even more evident in Tab. D appendix where the gain from PIVOT shrink significantly when going in a “Full train” regime where the visual encoder is allowed to adapt to the LLM during the training phase on Cambrian. All in all I found the PIVOT contribution significantly weaker than the rest and would have liked the paper even without it.

**Questions:**

1. In Fig. 4 the slope of SFT seems significantly more than DPO. Do you have experiment scaling the dataset size even further to verify wheter SFt eventually catch up with DPO?

2. Can you comment on weakness A above?


3. More of a suggestion, Fig. 11 in the main paper is very not informative/somehow misleading. I would suggest to move Fig. D from the appendix to the main paper as it does a way better job in explaining what is going on.

---

> ### Author Response · Authors · 2025-11-22
> **Response to Reviewer uppE (1/3)**
>
> We thank Reviewer uppE for their time and helpful feedback. We appreciate that the reviewer recognizes (1) the clear storyline and presentation, (2) the novelty of exploring SFT vs. RL, and (3) the extensive experimental analysis. We are **deeply grateful** for the positive feedback and are **pleased** that the reviewer found the paper enjoyable to read. We have carefully considered the concerns raised and provide our point-by-point responses below.
>
> ---
> > ### Major-W: I haven’t found major weaknesses in this work.
>
> We are encouraged that the reviewer found no major weaknesses. In addition, the reviewer’s subsequent comments reflect a deep and thorough understanding of our work, which we sincerely appreciate.
>
> ---
> > ### Minor-W1-a: The comparison is not completely fair from a data-scale perspective: rather than simply seeing more data, DPO appears to be more data-efficient.
>
> We appreciate the reviewer raising this important point, and **we agree with the reviewer’s observation**. **The advantage of DPO** cannot be fully attributed to simply seeing more samples; instead, it **may stem from its algorithmic design that enables more efficient data usage**. Figure 4 further supports this point. We present a more detailed discussion below.
>
> 1. In each MLM training iteration, DPO observes $ (I_{t}, q_{t}, y_{t}^{c}, y_{t}^{r}) $, whereas SFT uses only $ (I_{t}, q_{t}, y_{t}^{c}) $ without access to $ y_{t}^{r} $ (Line 142).
> 2. Equalizing this difference in a fair manner is highly challenging. Instead, we conduct an evaluation by varying the number of pairs.
> 3. However, as shown in Figure 4, DPO outperforms SFT even under data disadvantage (e.g., DPO-3K > SFT-40K), suggesting that DPO is more data-efficient.
>
> We would like to note that, beyond the simple intuition that “DPO is better,” our work shows that DPO is particularly effective on vision-intensive tasks and enhances visual representations in MLLMs. We have added these limitations in Section E.3, regarding the lack of fully data-fair comparison. Thank you for the helpful suggestion.

---

> ### Author Response · Authors · 2025-11-22
> **Response to Reviewer uppE (2/3)**
>
> > ### Minor-W1-b: The comparison is not completely fair from a data-type perspective. It would be interesting to repeat the comparison on a more SFT friendly distribution of data.
>
> We are deeply impressed by the reviewer’s thoughtful feedback. **We acknowledge that the relative performance of DPO and SFT may vary depending on the data type.** For example, as shown in Section B.1, reasoning-oriented data tend to favor DPO, and as noted in Line 1229, we believe that novel dataset types may exist that further strengthen visual representations. As suggested by the reviewer, **we construct more SFT-friendly data** **and repeat the comparison**. We observe that **such data reduces the data-efficiency advantage of DPO; however, its performance does not fall below that of SFT**.
>
> We first try to construct an SFT-friendly post-training dataset. When selecting 20k samples from the MPO dataset, we apply the following constraints to each pair, $ (I_{t}, q_{t}, y_{t}^{c}, y_{t}^{r}) $.
>
> - Under30: we select only samples where $y_{t}^{c}$ has fewer than 30 words. (ie, data with a clear answer).
> - NoThink: we select only samples where $y_{t}^{c}$ has fewer than 60 words and constrains no think tokens (e.g., \<think\> and \<review\>).
>
> We then repeat the SFT–DPO comparison using Qwen2.5-1.5B and Qwen2.5-3B, with SigLIP2-So/16 as the vision encoder. The results are shown in the table below.
>
> |Train|LLM|Data|Avg. (All)|General|OCR&Chart|Vision|Knowledge|
> |-|-|-|-|-|-|-|-|
> |DPO|1.5B|Origin|**61.0**|**69.8**|**63.1**|**57.7**|**53.3**|
> |SFT|1.5B|Origin|59.1|68.3|60.5|54.8|52.8|
> |$\Delta$ (DPO-SFT)|||+1.9|+1.5|+2.6|+2.9|+0.5|
> |DPO|1.5B|NoThink|**60.8**|**69.3**|**62.3**|**58.3**|**53.2**|
> |SFT|1.5B|NoThink|59.5|68.9|60.5|56.3|52.4|
> |$\Delta$ (DPO-SFT)|||+1.2|+0.4|+1.7|+2.0|+0.7|
> |DPO|1.5B|Under30|**60.1**|**69.6**|**60.0**|**57.4**|**53.3**|
> |SFT|1.5B|Under30|59.3|69.0|59.2|56.1|52.8|
> |$\Delta$ (DPO-SFT)|||+0.8|+0.6|+0.8|+1.3|+0.5|
> |||||||||
> |DPO|3B|Origin|**63.9**|**72.4**|**66.0**|**61.9**|**55.3**|
> |SFT|3B|Origin|62.3|71.4|62.7|59.7|55.2|
> |$\Delta$ (DPO-SFT)|||+1.7|+1.0|+3.3|+2.2|+0.1|
> |DPO|3B|NoThink|**63.5**|**71.8**|**64.1**|**62.5**|55.6|
> |SFT|3B|NoThink|62.9|71.6|63.8|60.0|56.2|
> |$\Delta$ (DPO-SFT)|||+0.6|+0.2|+0.3|+2.5|-0.6|
> |DPO|3B|Under30|**63.0**|**71.4**|**64.1**|**61.6**|55.0|
> |SFT|3B|Under30|62.5|71.0|64.0|59.4|**55.8**|
> |$\Delta$ (DPO-SFT)|||+0.5|+0.5|+0.1|+2.2|-0.8|
>
>
>
> On datasets similar in style to ‘clear-answer’ data, DPO shows reduced gains compared to the original setting. We hypothesize that the shorter responses decrease the informational gap between chosen and rejected samples, which diminishes DPO’s effectiveness. Nevertheless, DPO does not fall behind SFT; its performance remains better even under these SFT-friendly conditions. We We will faithfully incorporate this insight into our manuscript soon. The full results across the 16 benchmarks are provided at the bottom.
>
> If the reviewer is interested in alternative SFT-friendly data, we would be happy to provide further analyses.
>
> ||||General||||OCR&Chart||||Vision.||||Know.||||
> |-|-|-|-|-|-|-|-|-|-|-|-|-|-|-|-|-|-|-|
> |Train|LLM|Data|mme|mmb|seed|gqa|scienceqa|mmmu|mathvista|ai2d|chartqa|ocrbench|textvqa|docvqa|mmvp|realworldqa|cv2d|cv3d|
> |DPO|1.5B|Origin|1478.7|71.9|72.5|60.9|86.7|40.1|14.7|71.5|65.4|62.7|64.7|59.5|50|57.4|63.6|59.9|
> |SFT|1.5B|Origin|1442.1|70.6|71.9|58.7|86.6|41|12.6|70.9|62.1|58.8|63.4|57.7|44|56|63.1|56.1|
> |DPO|1.5B|NoThink|1441.4|72.3|72.7|60.3|86.1|40.1|15.1|71.3|64.6|61.8|63.8|58.9|47.3|59.0|64.9|62.0|
> |SFT|1.5B|NoThink|1471.6|71.4|71.7|59.1|85.5|41.0|13.1|70.1|62.6|58.8|64.8|56|46.7|58.8|63.3|56.3|
> |DPO|1.5B|Under30|1469.8|72.2|72.3|60.3|86.2|40.8|15.1|71.0|59.8|61.7|62.7|55.8|47.3|57.9|63.1|61.4|
> |SFT|1.5B|Under30|1486.6|70.3|71.8|59.6|85.9|40.0|15.9|69.2|59.7|57.8|64.2|55.1|45.3|57.4|62.9|59.0|
> ||||||||||||||||||||
> |DPO|3B|Origin|1553.1|76.2|74.4|61.5|87|42.6|15.7|75.9|67.6|62.2|64.4|69.9|52|59|67.2|69.3|
> |SFT|3B|Origin|1550|75.2|73.7|59.3|86.8|42.3|15.9|75.9|58.5|60.1|65.1|67.1|47.5|59|66.3|65.8|
> |DPO|3B|NoThink|1511.1|76.6|73.4|61.7|88.7|43.6|14.4|75.7|64.9|61.2|64.5|65.8|52.7|60.0|67.3|70.0|
> |SFT|3B|NoThink|1537.8|75.9|74.1|59.6|87.4|43.1|18.7|75.6|64.6|60.4|66.1|64.1|49.3|57.8|66.1|66.7|
> |DPO|3B|Under30|1480.2|76.1|74.1|61.4|87.6|42.4|15.1|75.0|65.2|62.5|62.9|65.9|53.3|57.8|66.3|68.9|
> |SFT|3B|Under30|1509.2|75.2|73.7|59.5|88.3|43.1|17.1|74.7|65.7|59.4|65.8|65.1|50.0|57.1|65.4|65.2|

---

> ### Author Response · Authors · 2025-11-22
> **Response to Reviewer uppE (3/3)**
>
> > ### Q1: Do you have an experiment scaling the dataset size to verify whether SFT eventually catches up with DPO?
>
> Yes! **When training MLLMs with more data, we observe that SFT does not catch up with DPO and show performance degradation**. The experimental results are presented in the table below.
>
> |DPO|Avg. (All)|General|OCR&Chart|Vision-Cent.|Knowledge|
> |-|-|-|-|-|-|
> |20K|61.0|70.0|62.6|58.4|53.0|
> |40K|61.2|**70.3**|62.6|**58.8**|53.2|
> |60K|**61.4**|69.6|**64.0**|57.9|**54.1**|
> |80K|60.9|69.7|62.4|58.7|53.0|
> |100K|59.8|69.3|60.8|55.9|53.3|
> |150K|54.7|65.0|50.1|52.4|51.3|
>
> |SFT|Avg. (All)|General|OCR&Chart|Vision-Cent.|Knowledge|
> |-|-|-|-|-|-|
> |20K|59.4|68.0|60.9|56.4|52.4|
> |40K|59.5|68.0|60.8|**56.6**|52.7|
> |60K|**59.6**|**68.4**|**61.1**|55.0|**53.9**|
> |80K|59.1|67.8|59.5|55.8|53.1|
> |100K|58.1|67.2|57.7|54.5|53.1|
> |150K|55.7|66.4|52.3|52.5|51.8|
>
>
> Although SFT shows a strong slope from 3K to 40K in Figure 4 and Figure B, this does not persist. With more data, the slope turns negative and SFT fails to catch up with DPO. In our experimental setup, we keep all hyperparameters fixed, except for the dataset size. We appreciate the reviewer’s valuable suggestion and will incorporate it into our manuscript. The full results across the 16 benchmarks are provided below.
>
> |DPO|General||||OCR&Chart||||Vision.||||Know.||||
> |-|-|-|-|-|-|-|-|-|-|-|-|-|-|-|-|-|
> ||mme|mmb|seed|gqa|chartqa|ocrbench|textvqa|docvqa|mmvp|realworldqa|cv2d|cv3d|scienceqa|mmmu|mathvista|ai2d|
> |20K|1485.1|72.4|72.9|60.3|65.2|61.4|64.1|59.5|51.3|58.8|63.2|60.2|86.2|40.8|14.0|71.1|
> |40K|1495.4|72.2|73.2|60.9|64.5|62.1|64.0|59.7|51.9|59.0|63.0|61.3|86.4|39.0|15.6|71.7|
> |60K|1468.6|72.9|73.0|58.9|66.4|66.5|64.7|58.4|47.3|59.0|61.8|63.6|88.3|40.1|14.9|72.9|
> |80K|1480.4|72.0|72.7|60.0|65.2|62.1|63.6|58.5|50.0|58.1|63.8|62.9|87.2|40.4|13.1|71.3|
> |100K|1465.1|71.9|72.3|59.8|64.5|60.9|61.0|56.7|45.7|57.3|61.3|59.4|87.1|41.4|14.6|69.9|
> |150K|1358.2|66.7|70.5|54.8|48.0|44.2|55.1|53.1|36.7|55.3|60.8|56.9|85.0|37.7|11.5|71.1|
>
> |SFT|General||||OCR&Chart||||Vision.||||Know.||||
> |-|-|-|-|-|-|-|-|-|-|-|-|-|-|-|-|-|
> ||mme|mmb|seed|gqa|chartqa|ocrbench|textvqa|docvqa|mmvp|realworldqa|cv2d|cv3d|scienceqa|mmmu|mathvista|ai2d|
> |20K|1420.5|70.9|71.7|58.5|62.4|58.9|64.5|58.0|44.6|58.2|63.6|59.3|86.0|40.2|12.1|71.1|
> |40K|1423.7|70.5|72.1|58.3|63.2|57.4|64.3|58.3|45.3|57.9|64.1|59.1|85.9|40.0|13.5|71.3|
> |60K|1447.9|71.3|72.6|57.3|63.7|59.7|63.2|58.0|44.0|56.1|62.0|58.0|87.8|41.2|14.1|72.4|
> |80K|1425.5|70.9|72.1|56.9|60.7|56.4|63.3|57.7|42.7|58.8|62.8|58.8|86.8|40.1|14.3|71.4|
> |100K|1417.1|69.3|72.1|56.6|59.0|53.6|60.2|57.9|37.3|59.3|61.5|59.8|86.8|39.7|13.5|72.3|
> |150K|1400.8|67.6|70.3|57.5|52.3|48.6|53.6|54.7|39.3|52.4|58.9|59.4|84.3|39.8|13.3|69.6|
>
>
> ---
> > ### Minor-W2: All in all, I found the PIVOT contribution significantly weaker than the rest and would have liked the paper even without it.
>
> We thank the reviewer for acknowledging our primary contribution. As correctly pointed out, **PIVOT is a derivative contribution built upon our findings, which demonstrates the practical utility of our analytics in improving performance. While it is simple, we view it as an under-explored training regime.** Regarding PIVOT’s value:
>
> - It enlightens that DPO can be effective for training vision encoders.
> - It extends prior work: while SFT-based language alignment (Bolya et al., 2025) has been shown to benefit the vision encoder, we further show that DPO yields even stronger visual representations.
> - It reveals an interesting implication: although SOTA encoders like SigLIP2 are often viewed as nearly optimal due to large-scale training, our results show that DPO can improve their performance within MLLMs.
>
> We believe these insights are valuable for exploring new RL algorithms for MLLMs, and we plan to pursue this direction in future work.
>
> \* Daniel Bolya et al. Perception encoder: The best visual embeddings are not at the output of the network, arXiv, 2025.
>
> ---
> > ### Q3: I would suggest moving Fig. D from the appendix to the main paper.
>
> We appreciate the reviewer’s valuable suggestion. We agree that Figure D is better suited for the main paper, and we have therefore modified and relocated it from the appendix.

---

> > ### Comment · Reviewer_uppE · 2025-11-26
> > **Rebuttal feedback**
> >
> > Thanks for the additional results and thank you for coming up with a way of ablating my weakness 1 b even tough my original review was not providing a clear actionable guideline on how to test that.
> > I had a positive rating for the paper before and will likelly maintain it post discussion.
> >
> > RE: Answer to Q1
> >
> > What is your intuition on the decrease of performance for both SFT and DPO when training longer? Is it a matter of picking better hyperparameters for the extended trainign regime or do you feel like there's something else at play?

---

> > > ### Author Response · Authors · 2025-12-02
> > > **Follow-up Response to Reviewer uppE**
> > >
> > > We are pleased that the additional results helped clarify the reviewer's earlier questions. We provide our response to the follow-up question below.
> > >
> > > > ### Q4. What is your intuition on the decrease of performance when training longer? Is it a matter of picking better hyperparameters?
> > >
> > > We thank the reviewer for the careful follow-up and the thoughtful question. Our intuition is that, as the reviewer mentioned, the performance drop can stem from a mismatch between the hyperparameters and the longer post-training regime.
> > >
> > > The post-training is for human preference alignment and is performed with 20K examples in our setup, a scale comparable to earlier studies (e.g., 4K~16K in Yu et al. 2025; Yang et al. 2025; Fu et al. 2025). In our additional experiments, we hold all hyperparameters constant and vary only the data size (i.e., total training steps), leading to a decrease in performance. As the reviewer noted, we also believe that selecting hyperparameters suited to the longer training regime can mitigate the issue. For example, adjusting the learning rate or weight decay may help, as MLLM training is sensitive to them, as shown in Section C.3. We will thoroughly incorporate a discussion of this intuition into our paper.
> > >
> > > \* Tianyu Yu et al. Rlaif-v: Open-source ai feedback leads to super gpt-4v trustworthiness. In CVPR, 2025.
> > >
> > > \* Zhihe Yang et al. Opa-dpo: Mitigating hallucinations in large vision-language models via dpo: On-policy data hold the key. In CVPR, 2025.
> > >
> > > \* Jinlan Fu et al. Chip: Cross-modal hierarchical direct preference optimization for multimodal llms. In ICLR, 2025.

---

### Official Review · Reviewer_FvAp · 2025-10-31

**Soundness:** 3
**Presentation:** 4
**Contribution:** 3
**Rating:** 8
**Confidence:** 4

**Summary:**

This paper investigates how RL post-training compares to the standard STF one in MLLMs on vision-related tasks. Their findings show an advantage of using the RL paradigm for both the quality of visual representation and downstream performance on VQA. The authors propose a new training regime for MLLMs, called Preference-Instructed Vision OpTimization (PIVOT), which consists of a pretraining stage and a post-training stage using DPO on instruction-following samples and 20K preference pairs.

**Strengths:**

- Thorough analysis of the impact of using STF vs RL in the post-training stage showing significant differences-
- Well written paper, easy to follow.

**Weaknesses:**

No experiments on other vision related tasks like image captioning, object hallucinations etc.

**Questions:**

- Significant benefits of using RL are demonstrated on VQA tasks. How about other vision related tasks like image captioning or object hallucinations? Did you perform any experiments to confirm that the performance transfers to these as well?
- I understand that the goal of this paper is to compare the effects of the STF vs RL in post-training on vision encoders. However, it would still be interesting to compare the performance of PIVOT with some of the established MLLM like BLIP2, LLaVa etc and those focusing on vision, like BRAVE: Broadening the Visual Encoding of Vision-Language Models (ECCV 2024), SPHINX: A Mixer of Weights, Visual Embeddings and Image Scales for Multi-modal Large Language Models (ECCV 2024) and similar (which are missing from the related work).

---

> ### Author Response · Authors · 2025-11-22
> **Response to Reviewer FvAp (1/2)**
>
> We thank Reviewer FvAp for their time and constructive feedback. We are pleased that the reviewer recognizes that (1) our analysis is thorough and (2) our paper is well-written. We have carefully considered the concerns raised and provide our point-by-point responses below. We will thoroughly incorporate the experimental results below into our manuscript.
>
> ---
>
> > ### W1: No experiments on other vision related tasks like image captioning, object hallucinations etc.
> >
> > ### Q1: How about other vision related tasks like image captioning or object hallucinations?
>
> We are grateful for the reviewer’s insightful suggestion, and we agree that evaluating our findings on broader vision tasks gives valuable insight. We perform extensive additional experiments that cover image captioning, object hallucination, and real-world robustness. **Through our experiments, we observe that DPO consistently outperforms SFT across various scenarios/benchmarks.** The results and detailed explanations are in the table below.
>
> **A. Image captioning**
>
> In addition, we test on image captioning, a primary task for MLLMs. We utilize the conventional benchmark MS COCO (Chen et al., 2014) and a recent benchmark for evaluating detailed captioning, DetailCaps-4870 (Dong et al., 2024).
>
> | Model | Avg (All) | MS COCO |  |  | DetailCaps-4870 |  |  |
> | - | - | - | - | - | - | - | - |
> |  |  | METEOR | ROUGE-L | BERTScore | METEOR | ROUGE-L | CAPTURE |
> | MLLM-0.5B-SFT | 41.03 | **28.22** | 50.34 | 67.11 | 16.77 | 26.65 | 57.07 |
> | MLLM-0.5B-DPO | **41.47** | 27.63 | **51.07** | **67.81** | **17.27** | **27.37** | **57.64** |
> | MLLM-1.5B-SFT | 40.84 | 27.61 | 47.98 | 66.17 | 17.25 | **27.82** | 58.18 |
> | MLLM-1.5B-DPO | **42.10** | **28.52** | **50.64** | **68.46** | **18.48** | 27.08 | **59.44** |
>
> **B. Hallucination**
>
> We extend our evaluation to include HallusionBench (Guan et al., 2024) and POPE (Li et al., 2023), which are benchmarks for assessing hallucination.
>
> | Model | Avg (All) | HallusionBench |  |  | POPE |  |  |
> | - | - | - | - | - | - | - | - |
> |  |  | AnswerAcc | FaithAcc | QuestionAcc | precision | recall | f1 |
> | MLLM-0.5B-SFT | 51.43 | 33.22 | **12.42** | 9.23 | 89.71 | 79.63 | 84.37 |
> | MLLM-0.5B-DPO | **54.76** | **37.85** | 11.84 | **15.6** | **93.71** | **82.06** | **87.5** |
> | MLLM-1.5B-SFT | 56.31 | 40.79 | 21.09 | 15.6 | 92.89 | **80.98** | 86.53 |
> | MLLM-1.5B-DPO | **58.97** | **48.79** | **22.83** | **21.09** | **94.53** | 79.95 | **86.63** |
>
> **C. Other vision-related tasks**
>
> Furthermore, we test on NaturalBench (Li et al., 2024), MME-RealWorld (Zhang et al., 2025), and VL-RewardBench (Li et al., 2025). As a brief description, NaturalBench assesses visual grounding robustness by mitigating reliance on language priors. MME-RealWorld evaluates practical utility in complex real-world environments. VL-RewardBench measures alignment with human preferences across diverse multimodal domains.
>
> | Model | Avg (All) | NaturalBench |  |  | MME_RealWorld | VL_RewardBench |
> | - | - | - | - | - | - | - |
> |  |  | GroupAcc | ImageAcc | QuestionAcc | Avg | Avg |
> | MLLM-0.5B-SFT | 12.65 | 12.84 | 42.18 | 0.3695 | 22.07 | **43.27** |
> | MLLM-0.5B-DPO | **14.27** | **17.84** | **47.61** | **0.4213** | **27.24** | 43.02 |
> | MLLM-1.5B-SFT | 13.75 | 16.53 | 44.03 | 0.4042 | 27.21 | 40.55 |
> | MLLM-1.5B-DPO | **14.49** | **25.05** | **54.08** | **0.5087** | **28.35** | **42.82** |
>
> **Summary**
>
> Our extended experiments consistently show that DPO achieves superior MLLM performance compared to SFT. As noted in Line 310, we speculate that these gains stem from the contrastive nature of DPO, which provides fine-grained gradient signals to the vision encoder and enhances the MLLM’s visual understanding. These properties translate into improvements across captioning, hallucination, and real-world vision benchmarks.
>
> \* Tianrui Guan et al. HallusionBench: An Advanced Diagnostic Suite for Entangled Language Hallucination and Visual Illusion in Large Vision-Language Models, CVPR 2024.
>
> \* Yifan Li et al. POPE: Evaluating Object Hallucination in Large Vision-Language Models, EMNLP 2023.
>
> \* Baiqi Li et al. NaturalBench: Evaluating Vision-Language Models on Natural Adversarial Samples, NeurIPS 2024.
>
> \* Yi-Fan Zhang et al. MME-RealWorld: Could Your Multimodal LLM Challenge High-Resolution Real-World Scenarios that are Difficult for Humans?, ICLR 2025.
>
> \* Lei Li et al. VLRewardBench: A Challenging Benchmark for Vision-Language Generative Reward Models, CVPR, 2025
>
> \* Xinlei Chen et al. Microsoft COCO Captions: Data Collection and Evaluation Server, arXiv 2014.
>
> \* Tianyi Zhang et al. BERTScore: Evaluating Text Generation with BERT, ICLR, 2020.
>
> \* Hongyuan Dong et al. CAPTURE: Benchmarking and Improving Detail Image Caption, arXiv 2024.

---

> ### Author Response · Authors · 2025-11-22
> **Response to Reviewer FvAp (2/2)**
>
> > ### Q2: It would be interesting to compare PIVOT with BLIP2, LLaVa, and MLLMs focusing on vision, like BRAVE and SPHINX.
>
> We deeply appreciate the reviewer’s valuable suggestion regarding a broader comparison and an expansion of the related work section. **We conduct additional experiments and observe that our MLLM with PIVOT demonstrates strong performance. We have also added the corresponding citations to the related work section.** Our evaluation covers several MLLMs, including BLIP-2 (Li et al., 2023), LLaVA-1.5 (Liu et al., 2024), LLaVA-Next (Li et al., 2024), LLaVA-OneVision (Li et al., 2025), and SPHINX (Lin et al., 2024). Unfortunately, we could not include BRAVE (Kar et al., 2024) because its official implementation is not public.
>
> The experimental results are presented in the table below.
>
> | Model | LLM size | Avg. (All) | General | OCR&Chart | Vision-Cent. | Knowledge |
> | - | - | - | - | - | - | - |
> | BLIP-2 | 7B | 43.7 | 48.5 | 52.8 | 31.0 | 42.5 |
> | SPHINX | 7B | 48.7 | 65.6 | 33.2 | 52.5 | 41.4 |
> | LLaVA-Next-Vincina | 7B | 50.6 | 68.3 | 38.0 | 52.8 | 43.9 |
> | LLaVA-Next-LLaMA-3 | 8B | 60.4 | **71.3** | **64.3** | 57.5 | 49.0 |
> | LLaVA-Next-Qwen2 | 7B | **62.0** | 70.5 | 63.3 | **62.3** | **53.3** |
> | | | | | | | |
> | SPHINX-Tiny | 1.1B | 37.5 | 54.2 | 19.3 | 43.6 | 33.2 |
> | LLaVA-OneVision | 0.5B | 50.0 | 60.4 | **57.0** | 42.7 | 39.4 |
> | Ours (SigLIP1-So/14)+PIVOT | 1.5B | 53.2 | 67.7 | 46.8 | 51.7 | 46.6 |
> | Ours (SigLIP2-So/16)+PIVOT | 1.5B | **55.6** | **68.1** | 53.9 | **52.4** | **48.1** |
>
> The results show that PIVOT (1.5B) outperforms other MLLMs built on language models under 2B parameters, and even exceeds larger models such as LLaVA-Next-Vincina, BLIP-2, and SPHINX. The full results across the 16 benchmarks are provided at the bottom.
>
> \* Junnan Li et al. BLIP-2: Bootstrapping Language-Image Pre-training with Frozen Image Encoders and Large Language Models, ICML 2023.
>
> \* Haotian Liu et al. LLaVA-1.5: Improved Baselines with Visual Instruction Tuning, CVPR 2024.
>
> \* Bo Li et al. LLaVA-NeXT: Stronger LLMs Supercharge Multimodal Capabilities in the Wild, arXiv 2024.
>
> \* Bo Li et al. LLaVA-OneVision: Easy Visual Task Transfer. TMLR, 2025.
>
> \* Ziyi Lin et al. SPHINX: The Joint Mixing of Weights, Tasks, and Visual Embeddings for Multi-modal Large Language Models, ECCV 2024.
>
> \* Oğuzhan Fatih Kar et al. BRAVE: Broadening the Visual Encoding of Vision-Language Models, ECCV 2024.
>
> | Model | LLM | General |  |  |  | OCR&Ch. |  |  |  | Vision. |  |  |  | Know. |  |  |  |
> | - | - | - | - | - | - | - | - | - | - | - | - | - | - | - | - | - | - |
> |                            |      | mme | mmb | seed | gqa | chartqa | ocrbench | textvqa | docVQA | mmvp | realworldqa | cv2d | cv3d | scienceqa | mmmu | mathvista | ai2d |
> | BLIP-2 | 7B | 1333.7 | 52.4 | 17.6 | 57.3 | 58.8 | 43.7 | 60.1 | 48.5 | 23.3 | 42.4 | 26.9 | 31.5 | 83.0 | 26.8 | 11.3 | 48.8 |
> | SPHINX | 7B | 1515.8 | 57.6 | 68.0 | 61.1 | 37.8 | 29.8 | 38.3 | 26.8 | 38.7 | 48.5 | 61.3 | 61.7 | 68.6 | 31.6 | 9.8 | 55.6 |
> | LLaVA-Next-Vincina | 7B | 1504.2 | 66.3 | 68.3 | 63.2 | 18.5 | 33.6 | 61.0 | 38.9 | 30.7 | 55.3 | 62.2 | 62.9 | 72.3 | 36.1 | 7.9 | 59.2 |
> | LLaVA-Next-LLaMA-3 | 8B | 1526.0 | 71.0 | 72.6 | 65.3 | 69.2 | 58.1 | 64.7 | 65.1 | 40.0 | 60.1 | 62.1 | 67.7 | 72.9 | 39.6 | 11.8 | 71.6 |
> | LLaVA-Next-Qwen2 | 7B | 1453.3 | 72.2 | 73.6 | 63.6 | 67.1 | 54.2 | 63.6 | 68.2 | 49.3 | 60.0 | 64.4 | 75.7 | 73.5 | 36.1 | 30.0 | 73.7 |
> |  |  |  |  |  |  |  |  |  |  |  |  |  |  |  |  |  |  |
> | SPHINX-Tiny | 1.1B | 1223.8 | 47.1 | 57.7 | 50.8 | 10.9 | 18.8 | 27.0 | 20.6 | 14.0 | 47.6 | 57.2 | 55.6 | 63.0 | 27.1 | 1.7 | 41.1 |
> | LLaVA-OneVision | 0.5B | 1274.9 | 54.9 | 63.3 | 59.5 | 60.9 | 60.4 | 50.9 | 55.8 | 18.7 | 54.1 | 43.8 | 54.3 | 67.5 | 31.0 | 4.8 | 54.3 |
> | Ours (SigLIP1-So/14)+PIVOT | 1.5B | 1446.7 | 68.7 | 70.0 | 59.8 | 73.8 | 38.2 | 9.2 | 65.2 | 46.6 | 36.6 | 54.7 | 49.4 | 36.0 | 55.8 | 57.3 | 57.5 |
> | Ours (SigLIP2-So/16)+PIVOT | 1.5B | 1427.5 | 67.8 | 71.8 | 61.4 | 57.0 | 48.3 | 58.5 | 51.7 | 38.3 | 56.2 | 59.1 | 56.2 | 73.9 | 41.5 | 12.3 | 64.7 |

---

### Official Review · Reviewer_U9oP · 2025-10-31

**Soundness:** 3
**Presentation:** 3
**Contribution:** 3
**Rating:** 4
**Confidence:** 5

**Summary:**

The research reveals that RL training (e.g., DPO) helps the visual encoder generate stronger and more precisely-localized visual features than SFT. This advantage results in better performance on both MLLM benchmarks (especially vision-heavy VQA) and classic vision tasks such as ImageNet classification and segmentation. Subsequently, the authors then consolidated these findings into PIVOT, a practical recipe, and validated its efficacy across a diverse range of vision encoders.

**Strengths:**

1. The work presents an under-explored area: how RL-based training affects the vision encoder in MLLMs, not just the language model. It  also provides systematic analyses of visual representations in RL-trained MLLMs for the first time.

2. Based on the analytical experiments, this work proposes PIVOT, a recipe that enhances the representational capacity of vision encoders through RL-based training with MLLM. Extensive experiments conducted on visual comprehension tasks consolidate its effectiveness.

**Weaknesses:**

1. The work only compares the performance of  visual encoders after post-training (SFT or RL). Could you provide an estimation of the visual encoder after Stage 1 (without any post-training) training on various visual comprehension tasks?

2. Most benchmarks are academic and may not fully reflect real-world multimodal challenges. Broader evaluation (e.g., on hallucination, robustness, or user-facing tasks) would be beneficial.


3. The study focuses exclusively on DPO as the RL method. While DPO is popular, other RL approaches (e.g., PPO, GRPO) are not explored, limiting the generalizability of the findings.

4. According to the experimental setup of PIVOT, a frozen visual encoder derived from post-training is integrated into a lightweight MLLM for visual understanding. In this context, it is hoped that the authors could provide an optimization approach for the lightweight MLLM through distillation and present corresponding performance evaluations. Specifically, this setting involves using the post-trained VLM(after SFT or RL) to generate corresponding data for LAION/CC/SBU-558K, mixing it with the original data, and then training a lightweight VLM. Only the projector layer would be optimized, and the final comprehension performance would be evaluated.

**Questions:**

Please see the problems listed in the weakness.

---

> ### Author Response · Authors · 2025-11-22
> **Response to Reviewer U9oP (1/3)**
>
> We thank Reviewer U9oP for their time and thorough feedback. We appreciate that the reviewer finds our work (1) provides the first systematic analysis of visual representations in MLLMs and (2) presents extensive experiments validating PIVOT’s effectiveness. We have carefully considered the concerns raised and provide our point-by-point responses below. We will thoroughly incorporate the experimental results below into our manuscript.
>
> ---
>
> > ### W1: Could you provide an estimation of the visual encoder after Stage 1 training on various visual comprehension tasks?
>
> We appreciate the reviewer’s valuable suggestion. To address this point, **we provide additional results below, which indicate that DPO post-training brings benefits to the model’s visual representations**s.
>
> The experimental results are presented in the table below.
>
> |Benchmark|ImageNet|||Segmentation|||VL alignment||
> |-|-|-|-|-|-|-|-|-|
> |LLM|Qwen2.5-1.5B +||Qwen2.5-3B +|Qwen2.5-1.5B +|||Qwen2.5-1.5B +|Qwen2.5-3B +|
> |Vision encoder|SigLIP2-L/16|SigLIP2-So/16|SigLIP2-So/16|CLIP-224px-large|CLIP-L/14 336px|SigLIP1-So/14|SigLIP2-So/16|SigLIP2-So/16|
> |Stage 1|49.17|50.06|50.57|23.91|**24.22**|28.08|0.1662|**0.1712**|
> |Stage 2 with SFT|47.7|48.53|49.15|23.50|23.10|28.41|0.1666|0.1689|
> |Stage 2 with DPO|**49.66**|**50.51**|**50.98**|**24.67**|24.18|**29.36**|**0.1679**|0.1707|
>
> For the VL Alignment evaluation, the reference LLM is Llama-3-8B. We show that DPO consistently outperforms SFT across all post-training settings and generally surpasses the Stage 1 model, with the advantage particularly evident in the ImageNet evaluation. We hope these results contribute to a more comprehensive understanding.

---

> > ### Comment · Reviewer_U9oP · 2025-11-24
> > **Discussion about the 1st training stage**
> >
> > Thanks for your hard work to response my question, here comes my discussion:
> >
> > 1. Based on the experimental results presented in W1, the training of stage-1 (before post-training) already enables the vision encoder to achieve relatively good image comprehension capabilities. That is, the primary benefits of the vision encoder for VLM in image comprehension derive from the 1st training stage, which utilizes 30M training data, rather than the post-training phase. The introduced DPO provides some improvement to the vision encoder's understanding capability, but it is relatively minor. However, I would like to confirm here: during the 1st training phase, are the parameters of the visual encoder updated?

---

> ### Author Response · Authors · 2025-11-22
> **Response to Reviewer U9oP (2/3)**
>
> > ### W2: Broader evaluation (e.g., on hallucination, robustness, or user-facing tasks) would be beneficial.
>
> We thank the reviewer’s constructive suggestion, and we agree that evaluating our findings beyond academic benchmarks is valuable. To address this, we conduct additional experiments on hallucination and robustness/real-world scenarios. **Through our experiments, we observe that DPO consistently outperforms SFT across various scenarios/benchmarks.** The results and detailed explanations are in the table below.
>
> **A. Hallucination**
>
> We evaluate our MLLMs on HallusionBench (Guan et al., 2024) and POPE (Li et al., 2023), which are benchmarks for assessing hallucination.
>
> |Model|Avg (All)|HallusionBench|||POPE|||
> |-|-|-|-|-|-|-|-|
> |||AnswerAcc|FaithAcc|QuestionAcc|precision|recall|f1|
> |MLLM-0.5B-SFT|51.43|33.22|**12.42**|9.23|89.71|79.63|84.37|
> |MLLM-0.5B-DPO|**54.76**|**37.85**|11.84|**15.6**|**93.71**|**82.06**|**87.5**|
> |MLLM-1.5B-SFT|56.31|40.79|21.09|15.6|92.89|**80.98**|86.53|
> |MLLM-1.5B-DPO|**58.97**|**48.79**|**22.83**|**21.09**|**94.53**|79.95|**86.63**|
>
> **B. Robustness & Real-world tasks**
>
> To validate our findings in user-facing tasks, we extend our evaluation to include NaturalBench (Li et al., 2024), MME-RealWorld (Zhang et al., 2025), and VL-RewardBench (Li et al., 2025). As a brief description, NaturalBench assesses visual grounding robustness by mitigating reliance on language priors. MME-RealWorld evaluates practical utility in complex real-world environments. VL-RewardBench measures alignment with human preferences across diverse multimodal domains.
>
> |Model|Avg (All)|NaturalBench|||MME_RealWorld|VL_RewardBench|
> |-|-|-|-|-|-|-|
> |||GroupAcc|ImageAcc|QuestionAcc|Avg|Avg|
> |MLLM-0.5B-SFT|12.65|12.84|42.18|0.3695|22.07|**43.27**|
> |MLLM-0.5B-DPO|**14.27**|**17.84**|**47.61**|**0.4213**|**27.24**|43.02|
> |MLLM-1.5B-SFT|13.75|16.53|44.03|0.4042|27.21|40.55|
> |MLLM-1.5B-DPO|**14.49**|**25.05**|**54.08**|**0.5087**|**28.35**|**42.82**|
>
> **C. Image captioning**
>
> In addition, we test on image captioning, a task widely used in applications. We utilize the conventional benchmark MS COCO (Chen et al., 2014) and a recent benchmark for evaluating detailed captioning, DetailCaps-4870 (Dong et al., 2024).
>
> |Model|Avg (All)|MS COCO|||DetailCaps-4870|||
> |-|-|-|-|-|-|-|-|
> |||METEOR|ROUGE-L|BERTScore|METEOR|ROUGE-L|CAPTURE|
> |MLLM-0.5B-SFT|41.03|**28.22**|50.34|67.11|16.77|26.65|57.07|
> |MLLM-0.5B-DPO|**41.47**|27.63|**51.07**|**67.81**|**17.27**|**27.37**|**57.64**|
> |MLLM-1.5B-SFT|40.84|27.61|47.98|66.17|17.25|**27.82**|58.18|
> |MLLM-1.5B-DPO|**42.10**|**28.52**|**50.64**|**68.46**|**18.48**|27.08|**59.44**|
>
> **Summary**
>
> Our extended experiments consistently show that DPO achieves superior MLLM performance compared to SFT. As noted in Line 310, we speculate that these gains stem from the contrastive nature of DPO, which provides fine-grained gradient signals to the vision encoder and enhances the MLLM’s visual understanding. These properties translate into improvements across captioning, hallucination, and real-world vision benchmarks.
>
> \* Tianrui Guan et al. HallusionBench: An Advanced Diagnostic Suite for Entangled Language Hallucination and Visual Illusion in Large Vision-Language Models, CVPR 2024.
>
> \* Yifan Li et al. POPE: Evaluating Object Hallucination in Large Vision-Language Models, EMNLP 2023.
>
> \* Baiqi Li et al. NaturalBench: Evaluating Vision-Language Models on Natural Adversarial Samples, NeurIPS 2024.
>
> \* Yi-Fan Zhang et al. MME-RealWorld: Could Your Multimodal LLM Challenge High-Resolution Real-World Scenarios that are Difficult for Humans?, ICLR 2025.
>
> \* Lei Li et al. VLRewardBench: A Challenging Benchmark for Vision-Language Generative Reward Models, CVPR, 2025
>
> \* Xinlei Chen et al. Microsoft COCO Captions: Data Collection and Evaluation Server, arXiv 2014.
>
> \* Satanjeev Banerjee et al. METEOR: An Automatic Metric for MT Evaluation with Improved Correlation with Human Judgments, ACL, 2005.
>
> \* Chin-Yew Lin et al. ROUGE: A Package for Automatic Evaluation of Summaries , ACL, 2004.
>
> \* Tianyi Zhang et al. BERTScore: Evaluating Text Generation with BERT, ICLR, 2020.
>
> \* Hongyuan Dong et al. CAPTURE: Benchmarking and Improving Detail Image Caption, arXiv 2024.
>
> \* RoboEXP: Action-Conditioned Scene Graph via Interactive Exploration for Robotic Manipulation.

---

> ### Author Response · Authors · 2025-11-22
> **Response to Reviewer U9oP (3/3)**
>
> > ### W3: Other RL approaches (e.g., PPO, GRPO) are not explored, limiting the generalizability of the findings.
>
> Thank you for raising an important question. We agree that applying our findings to other RL approaches could strengthen and generalize our claim. **We are currently conducting additional experiments with PPO, GRPO.** Due to the computational demands, results are expected within the next week. In addition, we would like to clarify our considerations on the choice of RL methods for analysis:
>
> 1. We select DPO as a primary RL method for the following reasons:
>    1. DPO does not rely on an external reward model, which minimizes confounding factors when comparing with SFT.
>    2. DPO allows controlled comparison with SFT on the same data format, $ (I_{t}, q_{t}, y_{t}^{c}, y_{t}^{r}) $ (Line 142).
> 2. PPO requires a reward model trained on different data $ ({I_{t}, q_{t}, y_{t}^{1}, r_{t}^{1}, y_{t}^{2}, r_{t}^{2}, \ldots}) $ (where $r_{t}$ represents a reward value or a rank), as well as additional data for training the policy model. Thus, it is challenging to build a data-fair setting for comparison between PPO and SFT.
> 3. GRPO relies on 'verifiable' data, such as math or coding problems for LLMs, or object counting and bounding-box annotations for MLLMs. As SFT and DPO data do not contain such verifiable signals, a fair comparison setting for GRPO and SFT presents additional challenges from those of PPO.
>
> Despite these limitations, we are making our best effort to extend our analysis to RL beyond DPO. We appreciate the reviewer’s patience and look forward to sharing them soon.
>
>
>
> ---
>
> > ### W4:  It is hoped that the authors could provide an optimization approach for the lightweight MLLM through distillation.
>
> The reviewer provides an insightful and reasonable suggestion. We agree that the suggested distillation complements our proposed PIVOT. **Therefore, we conduct additional experiments and observe that combining the distillation with PIVOT yields further improvements.**
>
> We set the experimental setup as follows: The teacher model is an MLLM (Qwen2.5-1.5B + SigLIP2-So/16 and Qwen2.5-1.5B + SigLIP1-So/14) that has completed Stage 1 (pre-training) and Stage 2 (DPO post-training). They are utilized to generate detailed captions for the LAION/CC/SBU-558K datasets. This data is then incorporated into the projector-only pre-training phase to develop the lightweight MLLM in Section 5. We refer to this strategy as +distillation.
>
> The experimental results are presented in the table below.
>
> |Model|Avg (All)|General|OCR&Chart|Vision-Cent.|Knowledge|
> |-|-|-|-|-|-|
> |MLLM-SigLIP1-So/14|50.9|65.4|42.3|49.8|46|
> |+ distillation|51.4|66.2|42.8|50.3|46.4|
> |$\Delta$|+0.5|+0.8|+0.5|+0.5|+0.4|
> |+ PIVOT|53.2|67.7|46.8|51.7|46.6|
> |**+ PIVOT + distillation**|**54**|**68.4**|**47.8**|**52.5**|**47.2**|
> |$\Delta$|+0.8|+0.7|+1|+0.8|+0.6|
> |MLLM-SigLIP2-So/16|52.4|66.2|46.6|50.6|46.1|
> |+ distillation|52.8|66.9|47|51|46.3|
> |$\Delta$|+0.4|+0.7|+0.4|+0.4|+0.2|
> |+ PIVOT|55.6|68.1|53.9|52.4|48.1|
> |**+ PIVOT + distillation**|**56.4**|**68.8**|**54.9**|**53.3**|**48.7**|
> |$\Delta$|+0.8|+0.7|+1|+0.9|+0.6|
>
> Regarding the experimental results, applying '+distillation' to an MLLM using the original SigLIP yields performance gains, though the improvements are smaller than those achieved by PIVOT. Notably, combining '+distillation with PIVOT leads to additional performance boosts.
>
> We believe that the reviewer’s suggestion targets an aspect that is highlighted in LLaVA-OneVision (Li et al., 2025) and Cambrian (Tong et al., 2024), which demonstrate the benefits of utilizing curated data during the pre-training phase. We appreciate the reviewer for pointing out this opportunity. The full results across the 16 benchmarks are provided at the bottom.
>
> \* Bo Li et al. LLaVA-OneVision: Easy Visual Task Transfer. TMLR, 2025.
>
> \* Shengbang Tong et al. Cambrian-1: A Fully Open, Vision-Centric Exploration of Multimodal LLMs. NeurIPS, 2024.
>
> |Model|General||||OCR&Ch.||||Vision.||||Know.||||
> |-|-|-|-|-|-|-|-|-|-|-|-|-|-|-|-|-|
> ||mme|mmb|seed|gqa|chartqa|ocrbench|textvqa|docVQA|mmvp|realworldqa|cv2d|cv3d|scienceqa|mmmu|mathvista|ai2d|
> |MLLM-SigLIP1-So/14|1384.1|66.0|67.7|58.8|36.7|33.1|51.9|47.5|32.7|54.8|55.9|55.9|72.9|39.0|8.5|63.5|
> |+ distillation|1402.9|66.7|68.5|59.5|36.9|33.2|52.8|48.1|34.0|55.0|56.5|55.9|74.1|39.4|8.3|63.9|
> |+ PIVOT|1446.7|68.7|70.0|59.8|46.6|36.6|54.7|49.4|36.0|55.8|57.3|57.5|73.8|38.2|9.2|65.2|
> |+ PIVOT + distillation|1457.4|70.2|70.1|60.4|47.8|37.1|55.6|50.6|36.7|56.3|58.4|58.7|75.3|38.5|8.8|66.1|
> |MLLM-SigLIP2-So/16|1403.0|65.8|68.9|59.8|43.5|39.3|54.0|49.5|36.0|55.4|58.2|52.8|72.9|38.2|8.8|64.4|
> |+ distillation|1425.1|66.3|69.4|60.6|44.8|39.6|54.2|49.4|35.0|56.1|58.8|54.1|72.4|38.8|8.9|65.2|
> |+ PIVOT|1427.5|67.8|71.8|61.4|57.0|48.3|58.5|51.7|38.3|56.2|59.1|56.2|73.9|41.5|12.3|64.7|
> |+ PIVOT + distillation|1443.8|68.9|72.0|62.1|58.1|49.2|59.6|52.6|39.0|56.7|60.1|57.4|74.6|42.4|12.4|65.4|

---

> > ### Comment · Reviewer_U9oP · 2025-11-24
> > **Discussion about the 1st stage training.**
> >
> > 2. If the visual encoder parameters are updated during the first phase of training, based on the comparative observations in point 1, how does the vision encoder trained in the first phase (without any post-training) perform when connected to Qwen2.5-1.5B + SigLIP2-So/16 and Qwen2.5-1.5B + SigLIP1-So/14, compared to PIVOT on the tasks mentioned in W4 - General, OCR&Chart, Vision-Centric and Knowledge tasks?

---

> ### Author Response · Authors · 2025-11-26
> **Follow-up Response to Reviewer U9oP**
>
> > ### Q1. During the 1st training phase, are the parameters of the visual encoder updated? If so, how does the vision encoder trained in Stage 1 perform when integrated in MLLMs?
> >
> > ### W5. DPO provides some improvement to the vision encoder when evaluated on vision-only tasks, but it is relatively minor.
>
> We appreciate the reviewer’s prompt engagement and thoughtful comments. First, we clarify that **the vision encoder parameters are updated during Stage 1** in our setup. To resolve the concern related to Stage 1, **we conduct the suggested experiment and observe that DPO post-training shows clear benefits over only pre-training (i.e., Stage 1)** in MLLM evaluation.
>
> To assess visual representations in the MLLM context, we compare encoders trained ‘Stage1-only’ alongside ‘Stage2-SFT’ and ‘Stage2-DPO’ (i.e., PIVOT). The results are in the table below.
>
> |MLLM =  Qwen1.5-1.5B +||Avg (All)|General|OCR&Chart|Vision-Cent.|Knowledge|
> |-|-|-|-|-|-|-|
> |+ Original SigLIP1-So/14||50.9|65.4|42.3|49.8|46.0|
> |+ Stage1-only||51.8|66.1|44.7|50.3|46.2|
> |+ Stage2-SFT||52.2|66.5|45.2|50.8|46.3|
> |+ Stage2-DPO||**53.2**|**67.7**|**46.8**|**51.7**|**46.6**|
> |$\Delta$ (Stage2-DPO -  Stage1-only)||+1.4|+1.6|+2.1|+1.4|+0.4|
> |$\Delta$ (Stage2-DPO -  Stage2-SFT)||+1|+1.2|+1.6|+0.9|+0.3|
> ||||||||
> |+ Original SigLIP2-So/16||52.4|66.2|46.6|50.6|46.1|
> |+ Stage1-only||54.4|66.6|51.8|51.4|48.0|
> |+ Stage2-SFT||54.6|66.9|52.2|51.7|47.7|
> |+ Stage2-DPO||**55.6**|**68.1**|**53.9**|**52.4**|**48.1**|
> |$\Delta$ (Stage2-DPO -  Stage1-only)||+1.2|+1.5|+2.1|+1.0|+0.1|
> |$\Delta$ (Stage2-DPO -  Stage2-SFT)||+1|+1.2|+1.7|+0.7|+0.4|
>
> Based on the results above, we would like to highlight that:
>
> - Applying Stage2-DPO provides meaningful improvements over the Stage1-only.
> - The gains by Stage2-DPO are particularly notable on vision-intensive tasks (e.g., General and OCR&Chart), as noted in Finding 2 of our paper.
> - The performance gap between Stage‑2 SFT and Stage‑2 DPO continues to support our main finding as presented in our paper.
>
> We will include the stage-1-only setup (i.e., without any post-training) as a baseline to explicitly demonstrate the impact of post-training. The full results across the 16 benchmarks are provided at the bottom.
>
> Furthermore, we discuss the rationale and validity of our focus on post-training below:
>
> - The post-training phase is used specifically for human preference alignment, a process widely regarded as important in LLM/MLM development.
> - We perform a controlled comparison between SFT and DPO under the same training phase and data scale, enabling a focused assessment of post-training objectives.
> - We utilize 20K post-training data, which is comparable to the scale used in prior works (e.g., 4K–16K), providing a reasonable frame for interpreting the results.
>
> The reviewer’s valuable feedback offers insight for future directions, such as designing new post-training data to strengthen the effect of DPO on vision-only tasks. We sincerely thank the reviewer.
>
> For reference, we present the post-training DPO data sizes from prior works, as follows. Full references for each abbreviation are provided in Table A of our paper.
>
> |Related work||RLHF-V (2024)|RLAIF-V (2025)|OPA-DPO (2025)|CHiP (2025)|LPOI (2025)|mDPO (2024)|V-DPO (2024)|
> |-|-|-|-|-|-|-|-|-|
> |DPO data size||10k|16K|4.8K|5K|10K|10K|5K|
>
> \* Tianyu Yu et al. Rlaif-v: Open-source ai feedback leads to super gpt-4v trustworthiness. In CVPR, 2025.
>
> \* Zhihe Yang et al. Opa-dpo: Mitigating hallucinations in large vision-language models via dpo. In CVPR, 2025.
>
> \* Jinlan Fu et al. Chip: Cross-modal hierarchical direct preference optimization for multimodal llms. In ICLR, 2025.
>
> \* Fatemeh Pesaran Zadeh et al. Lpoi: Listwise preference optimization for vision language models. In ACL, 2025.
>
> |MLLM =  Qwen1.5-1.5B +|General||||OCR&Chart||||Vision-Cent.||||Knowledge||||
> |-|-|-|-|-|-|-|-|-|-|-|-|-|-|-|-|-|
> ||mme|mmb|seed|gqa|chartqa|ocrbench|textvqa|docVQA|mmvp|realworldqa|cv-2d|cv-3d|scienceqa|mmmu|mathvista|ai2d|
> |+ Original SigLIP1-So/14|1384.1|66.0|67.7|58.8|36.7|33.1|51.9|47.5|32.7|54.8|55.9|55.9|72.9|39.0|8.5|63.5|
> |+ Stage1-only|1371.5|67.7|69.0|59.1|43.1|35.0|52.7|47.9|35.0|53.0|57.2|56.1|73.2|38.3|9.0|64.9|
> |+ Stage2-SFT|1389.7|67.8|68.9|59.7|44.3|34.5|53.1|48.9|37.3|52.7|56.5|56.8|73.1|37.7|9.4|64.9|
> |+ Stage2-DPO (i.e., PIVOT)|1446.7|68.7|70.0|59.8|46.6|36.6|54.7|49.4|36.0|55.8|57.3|57.5|73.8|38.2|9.2|65.2|
> |+ Original SigLIP2-So/16|1403.0|65.8|68.9|59.8|43.5|39.3|54.0|49.5|36.0|55.4|58.2|52.8|72.9|38.2|8.8|64.4|
> |+ Stage1-only|1402.1|66.4|69.6|60.2|55.3|44.5|56.2|51.2|35.7|54.4|57.9|57.8|74.4|41.2|11.8|64.6|
> |+ Stage2-SFT|1417.0|65.9|70.4|60.5|54.4|45.6|57.8|51|36.3|54.8|60.0|55.8|73.3|40.9|11.6|64.9|
> |+ Stage2-DPO (i.e., PIVOT)|1427.5|67.8|71.8|61.4|57.0|48.3|58.5|51.7|38.3|56.2|59.1|56.2|73.9|41.5|12.3|64.7|

---

> > ### Comment · Reviewer_U9oP · 2025-11-28
> > **Response to the authors**
> >
> > I appreciate the authors for providing comprehensive experimental results during the rebuttal period, which effectively demonstrated the validity of the proposed approach and fully addressed all the concerns I had raised. Therefore, I'm happy to raise my score to 8 with strong accept.

---

> > > ### Author Response · Authors · 2025-12-02
> > > **Follow-up Response to Reviewer U9oP (2/2)**
> > >
> > > **3. MPO vs. DPO vs. SFT**
> > >
> > > Finally, we extend our comparison by including MPO (Wang et al. 2024) as a variant of DPO. Mixed preference optimization (i.e., MPO) combines the objectives of DPO, SFT, and a binary preference classification loss. We integrate their implementation into our codebase and run the experiments. The tables below report MLLM and vision-only evaluation results under different MLLM training algorithms.
> > >
> > > |MLLM eval. (Section 3)|Post-train|Avg|General|OCR&Chart|Vision|Knowledge|
> > > |-|-|-|-|-|-|-|
> > > |Qwen2.5-1.5B + SigLIP2-L/16|SFT|59.4|68.0|60.9|56.4|52.4|
> > > |Qwen2.5-1.5B + SigLIP2-L/16|DPO|61.0|70.0|**62.6**|58.4|53.0|
> > > |Qwen2.5-1.5B + SigLIP2-L/16|MPO|**61.5**|**70.2**|62.2|**59.6**|**54.0**|
> > >
> > > MPO achieves the highest average score in the MLLM evaluation, outperforming SFT, though its gain over DPO remains modest. On vision‑only benchmarks, MPO matches DPO, suggesting that its impact on visual representation learning is limited. Moreover, we present additional results below, following the setup in Section 5, where visual representations are evaluated within MLLMs.
> > >
> > > |Vision-only eval. (Section4)|Post-train|ImageNet|Segmentation|
> > > |-|-|-|-|
> > > |Qwen2.5-1.5B + SigLIP2-L/16|SFT|48.53|30.18|
> > > |Qwen2.5-1.5B + SigLIP2-L/16|DPO|**50.51**|30.89|
> > > |Qwen2.5-1.5B + SigLIP2-L/16|MPO|50.18|**31.07**|
> > >
> > > The results demonstrate that DPO and MPO both improve visual representation quality over SFT, with MPO showing slight gains in vision-centric and knowledge tasks.
> > >
> > > |Vision eval. in MLLMs (Section 5)|Avg (All)|General|OCR&Chart|Vision-Cent.|Knowledge|
> > > |-|-|-|-|-|-|
> > > |MLLM = Qwen1.5-1.5B +||||||
> > > |+ Original SigLIP2-So/16|52.4|66.2|46.6|50.6|46.1|
> > > |+ Stage2-SFT|54.6|66.9|52.2|51.7|47.7|
> > > |+ Stage2-DPO (i.e., PIVOT)|**55.6**|**68.1**|**53.9**|52.4|48.1|
> > > |+ Stage2-MPO|**55.6**|67.9|53.8|**52.6**|**48.3**|
> > >
> > >
> > >
> > > **4. Summary**
> > >
> > > Our additional experiments with GRPO, PPO, and MPO demonstrate that RL-based training improves MLLM performance and visual representations beyond SFT, confirming that our findings hold across other RL algorithms. As noted in Section E.4, we plan to develop new RL algorithms for stronger visual representations, which we believe will help MLLMs better “see.”
> > >
> > >
> > >
> > > \* Liang Chen et al. R1-V: Reinforcing Super Generalization Ability in Vision-Language Models with Less Than \$3. GitHub repository, 2025.
> > >
> > > \* Yaowei Zheng et al. EasyR1: An Efficient, Scalable, Multi-Modality RL Training Framework. In GitHub repository, 2025.
> > >
> > > \* Haozhan Li et al. SimpleVLA-RL: Scaling VLA Training via Reinforcement Learning. In arXiv, 2025.
> > >
> > > \* Haozhan Shen et al. VLM-R1: A Stable and Generalizable R1-style Large Vision-Language Model. In arXiv, 2025.
> > >
> > > \* Yuexiang Zhai et al. RL4VLM: Fine-Tuning Large Vision-Language Models as Decision-Making Agents via Reinforcement Learning. In NeurIPS, 2025.
> > >
> > > \* Tianzhe Chu et al. SFT Memorizes, RL Generalizes: A Comparative Study of Foundation Model Post-training. In ICML, 2025.
> > >
> > > \* Zhiqing Sun et al. LLaVA-RLHF: Aligning Large Multimodal Models with Factually Augmented RLHF. In arXiv, 2023.
> > >
> > > \* Weiyun Wang et al. MPO: Enhancing the Reasoning Ability of Multimodal Large Language Models via Mixed Preference Optimization. In arXiv, 2024.
> > >
> > >
> > >
> > >
> > > The full results across the 16 benchmarks are provided below.
> > >
> > > |Base model|post-train|General||||OCR&Chart||||Vision-Cent.||||Knowledge||||
> > > |-|-|-|-|-|-|-|-|-|-|-|-|-|-|-|-|-|-|
> > > |||mme|mmb|seed|gqa|scienceqa|mmmu|mathvista|ai2d|chartqa|ocrbench|textvqa|docvqa|mmvp|realworldqa|cv-2d|cv-3d|
> > > |QwenVL-2.5-3B|SFT|1536.7|73.4|71.2|56.0|78.0|44.0|21.7|76.5|62.9|64.0|62.5|86.7|44.0|50.1|67.4|70.4|
> > > |QwenVL-2.5-3B|**GRPO**|1575.8|77.1|73.3|59.2|79.3|46.4|22.6|78.5|70.4|67.7|66.8|88.3|43.0|58.2|70.6|74.0|
> > > |||||||||||||||||||
> > > |LLaVA-1.0-7B|SFT|1201.1|39.3|42.5|32.5|64.8|27.9|3.1|45.0|7.1|21.6|34.8|43.8|10.9|31.3|25.0|39.5|
> > > |LLaVA-1.0-7B|**PPO**|1230.6|39.5|43.0|32.2|64.7|28.1|4.4|48.4|15.7|24.8|34.8|44.8|15.2|33.5|25.0|38.4|
> > > |||||||||||||||||||
> > > |Qwen2.5-1.5B + SigLIP2-L/16|SFT|1420.5|70.9|71.7|58.5|86.0|40.2|12.1|71.1|62.4|58.9|64.5|58.0|44.6|58.2|63.6|59.3|
> > > |Qwen2.5-1.5B + SigLIP2-L/16|DPO|1485.1|72.4|72.9|60.3|86.2|40.8|14.0|71.1|65.2|61.4|64.1|59.5|51.3|58.8|63.2|60.2|
> > > |Qwen2.5-1.5B + SigLIP2-L/16|**MPO**|1496.7|72.3|72.4|61.1|87.0|41.8|15.3|72.0|65.2|60.4|64.1|59.1|53.3|59.6|63.3|62.2|
> > >
> > >
> > >
> > >
> > > |Vision eval. in MLLMs (Section 5)|General||||OCR&Chart||||Vision-Cent.||||Knowledge||||
> > > |-|-|-|-|-|-|-|-|-|-|-|-|-|-|-|-|-|
> > > |MLLM = Qwen1.5-1.5B +|mme|mmb|seed|gqa|chartqa|ocrbench|textvqa|docVQA|mmvp|realworldqa|cv-2d|cv-3d|scienceqa|mmmu|mathvista|ai2d|
> > > |+ Original SigLIP2-So/16|1403.0|65.8|68.9|59.8|43.5|39.3|54.0|49.5|36.0|55.4|58.2|52.8|72.9|38.2|8.8|64.4|
> > > |+ Stage2-SFT|1417.0|65.9|70.4|60.5|54.4|45.6|57.8|51|36.3|54.8|60.0|55.8|73.3|40.9|11.6|64.9|
> > > |+ Stage2-DPO (i.e., PIVOT)|1427.5|67.8|71.8|61.4|57.0|48.3|58.5|51.7|38.3|56.2|59.1|56.2|73.9|41.5|12.3|64.7|
> > > |+ Stage2-MPO|1438.9|68.3|71|60.3|56.6|49.1|59|50.5|39|55.2|60.5|55.6|72.3|41.4|13.8|65.5|

---

> ### Author Response · Authors · 2025-12-02
> **Follow-up Response to Reviewer U9oP (1/2)**
>
> We are pleased that our responses fully addressed the reviewer’s concerns, and we deeply appreciate their decision to raise the score to a strong accept.
>
> We had committed to providing additional experimental results, and we are happy to share them now. The Reviewer raised the concern about whether our findings hold when comparing other RL algorithms with SFT. Accordingly, **we have completed additional experiments with PPO, GRPO, and a DPO variant, and we observe that our findings generalize across them.** We noted the following considerations as the rationale for choosing DPO as our primary RL method:
>
> - DPO enables direct comparison with SFT using identical data formats and does not require an external reward model.
> - PPO relies on external reward models and uses data formats that are not aligned with those of SFT.
> - GRPO depends on verifiable data types, making it difficult to ensure comparable training setups.
>
> Despite these limitations, we are making our best effort to extend our analysis to other RL algorithms, and we present the results below.
>
>
>
> **1. GRPO vs. SFT**
>
> We first survey publicly available implementations of the GRPO, as summarized below.
>
> |Code|Main task|GRPO data|SFT data|GRPO code|SFT code|Vision encoder|
> |-|-|-|-|-|-|-|
> |R1-V (2025)|object counting / geometry math|o|x|o|o|update|
> |EasyR1 (2025)|geometry math|o|x|o|x|update|
> |SimpleVLA-RL (2025)|robot action planning|o|x|o|x|freeze|
> |VLM-R1 (2025)|bounding-box annotation|**o**|**o**|**o**|**o**|**update**|
>
> Among these options, we select the codebase from VLM-RL for our GRPO vs. SFT experiments, as it offers a validated training strategy for proper GRPO implementation. Here, QwenVL-2.5-3B is used as the base model, and we perform 1,500 post-training steps using different objectives while adopting the remaining training parameters directly from the repository. We will describe the experimental setup in full detail in the manuscript.
>
> Below, we present results on **16 MLLM benchmarks** for MLLMs trained with different objectives. We also evaluate the vision encoder updated within MLLMs on vision-only benchmarks.
>
> |MLLM eval. (Section 3)|Post-train|Avg (All)|General|OCR&Chart|Vision|Knowledge|
> |-|-|-|-|-|-|-|
> |QwenVL-2.5-3B|SFT|62.8|69.4|69|58|55|
> |QwenVL-2.5-3B|GRPO|**65.9**|**72.1**|**73.3**|**61.4**|**56.7**|
> ||$\Delta$|+3.1|+2.7|+4.3|+3.4|+1.7|
>
> |Vision-only eval. (Section4)|Post-train|ImageNet|Segmentation|
> |-|-|-|-|
> |QwenVL-2.5-3B|SFT|52.01|33.71|
> |QwenVL-2.5-3B|GRPO|**53.94**|**35.54**|
> ||$\Delta$|+1.93|+1.8|
>
> The results show that GRPO consistently outperforms SFT on both evaluations. In particular, RL (i.e., GRPO) strengthens MLLMs more on vision-intensive tasks than SFT (Finding 2 in Section 3). MLLM training reshapes the vision encoder, and RL yields more effective visual representations (Findings 3 & 4 in Section 4). These observations confirm that our findings generalize beyond DPO to GRPO.
>
>
>
> **2. PPO vs. SFT**
>
> We next turn to the comparison between PPO and SFT and provide a survey of available codebases below.
>
> |Code|Data type|PPO data|SFT data|PPO code|SFT code|Vision encoder|
> |-|-|-|-|-|-|-|
> |RL4VLM (2024)|card game / robot action planning|o|o|o|x|freeze|
> |SFT Memo., RL Generalize (2025)|card game / navigation|o|x|o|o|update|
> |LLaVA-RLHF (2023)|conversation|**o**|**o**|**o**|**o**|freeze|
>
> Based on these conditions, we choose to adopt the LLaVA-RLHF implementation. For PPO, we utilize their publicly released model, and for SFT, we post-train a LLaVA-1.0-7B for one epoch using their SFT data and the original LLaVA framework (Liu et al. 2023). Further details will be described in our paper. Although the vision encoder is frozen in this setup, which prevents isolated evaluation of visual representations, we conduct this comparison to examine the influence of different objectives on MLLM performance. The results are presented as follows:
>
> |MLLM eval. (Section 3)|Post-train|Avg  (All)|General|OCR&Chart|Vision|Knowledge|
> |-|-|-|-|-|-|-|
> |LLaVA-1.0-7B|SFT|33.1|43.6|26.8|26.7|35.2|
> |LLaVA-1.0-7B|PPO|34.6|44|30|28|36.4|
> ||$\Delta$|+1.5|+0.4|+3.2|+1.3|+1.2|
>
> We observe that the PPO-trained MLLM outperforms its SFT counterpart on 16 MLLM benchmarks, with a notable gain on OCR&Chart tasks (Finding 2). We note that LLaVA-1.0-13B is used as the reward model in this codebase, which introduces confounding factors beyond the training objectives alone. Our DPO vs. SFT comparison mitigates this issue and provides a clearer perspective on RL vs. SFT, consistent with prior works in Table A.

---

### Official Review · Reviewer_ncjS · 2025-11-01

**Soundness:** 3
**Presentation:** 3
**Contribution:** 3
**Rating:** 6
**Confidence:** 4

**Summary:**

This work aims to understand the effect of reinforcement learning versus supervised fine-tuning post-training on the visual encoders of large vision-language models. Accordingly, the work shows that, RL (via DPO) tends to improve vision-centric MLLM performance and yields stronger, more localized visual features than SFT. Following from this intuition, the work then proposes the upgrade the visual encoder quality by first training it with DPO in an MLLM post-training setting. Several experimental results support both the initial analyses and the usefulness of PIVOT.

**Strengths:**

Here are the primary strengths of the work:

- The motivation is very clear. The work isolates SFT vs. DPO for MLLM post-training and explicitly examines effects on both the MLLM and the vision encoder rather than only downstream scores.
- The work includes several encoder-centric analysis, complementing VQA results with ImageNet linear probes, segmentation probing, and qualitative localization, arguing that RL strengthens and localizes visual features.
- The work then proposes a simple extension of DPO to achieve stronger visual encoders, called "PIVOT", while showing reusing an RL-trained encoder in new MLLMs gives decent gains.

**Weaknesses:**

Here are the primary weaknesses of the work:

**W1: Limited Methodological Novelty:** The paper’s primary contribution is an analysis of SFT vs. DPO under a direct, unmodified application of DPO without any new objectives or reward. The proposed visual encoder-upgrading method, PIVOT, also follows straightly from this.

**W2: Limited to a Single RL Variant:** The field is shifting quickly and there are many newer RL variants utilized in MLLMs, e.g. GRPO, as also noted by the authors. However, the conclusions are demonstrated for DPO-style RL within a specific Stage-2 pipeline and it’s unclear how broadly they hold across other RL algorithms, reward definitions, or different post-training recipes.

**Questions:**

- Do the main findings (encoder strengthening and vision-centric gains) persist with other RL approaches (e.g., PPO/GRPO) or alternative pairwise objectives beyond DPO?

- Where does DPO help least (e.g., knowledge VQA)? Though the work is vision-centric, how does the end MLLM perform under text-only benchmarks, such as HellaSwag? Do you observe any depreciation?

- For the models leveraging a combination of SFT and DPO [A, B], where does your work stand? Would your conclusions hold for a practical pipeline with SFT→DPO (and DPO→SFT) under equal compute? There is a mention of this literature in the Appendix, stating how this work only focuses on the differences between SFT vs DPO, but I believe that a more detailed discussion is required here.

I believe that the work could be improved further by addressing the aforementioned questions.

---
[A] Yu, T., Zhang, H., Li, Q., Xu, Q., Yao, Y., Chen, D., ... & Sun, M. (2025). Rlaif-v: Open-source ai feedback leads to super gpt-4v trustworthiness. In Proceedings of the Computer Vision and Pattern Recognition Conference (pp. 19985-19995).

[B] Sun, Z., Shen, S., Cao, S., Liu, H., Li, C., Shen, Y., ... & Darrell, T. (2023). Aligning large multimodal models with factually augmented rlhf. arXiv preprint arXiv:2309.14525.

---

> ### Author Response · Authors · 2025-11-22
> **Response to Reviewer ncjS (1/2)**
>
> We thank Reviewer ncjS for their time and valuable feedback. We are glad that the reviewer values (1) the clear motivation and (2) the several encoder-centric analysis. We have carefully considered the concerns raised and provide our point-by-point responses below.
>
> ---
>
> > ### W1: Methodological Novelty: The paper’s primary contribution is an analysis of SFT vs. DPO under a direct, unmodified application of DPO without any new objectives or reward.
>
> We are grateful to reviewer ncjS for raising the important point. As the reviewer mentioned, our main contribution is analytical rather than methodological. We would like to emphasize our analytical results, which demonstrate that methodological novelty is an additive benefit and not a primary focus: (1) **a systematic analysis of SFT vs. RL** **that reveals overlooked effects on vision representations**, (2) **novel findings derived from this analysis**, and (3) **illuminating DPO’s value for vision encoder training**. We present further details below.
>
> 1. Our work analyzes RL vs. SFT, offering not only a distinct attempt compared to prior MLLM studies but also the first understanding of how MLLM training influences the vision encoder.
>    1. Unlike prior studies limited to specialized settings (e.g., card games and robot planning), our analysis provides a broader comparison across 16 MLLM benchmarks.
>    2. Prior works lacked a fair comparison across training algorithms (Line 140), whereas our study provides a controlled evaluation using the same data and full-parameter updates.
>    3. The MLLM field has largely been LLM-centric; in contrast, we provide the first analysis of this previously overlooked vision-encoder component.
> 2. Our study presents several novel and interesting findings. The vision-level evidence is grounded in extensive vision-only benchmarks. We provide:
>    1. [MLLM-1] The vision encoder’s capacity significantly affects performance, as much as the LLM.
>    2. [MLLM-2] DPO is consistently more effective than SFT on strongly vision-related tasks.
>    3. [Vision-1] MLLM training actually reshapes the visual representations!
>    4. [Vision-2] DPO induces stronger and more precisely localized visual features than SFT.
>    5. [Vision-3] The vision encoder benefits from a larger LLM, receiving more informative optimization signals.
> 3. Our work enlightens that an unmodified DPO can be effective for training vision encoders.
>    1. While SFT-based language alignment (Bolya et al., 2025) has been shown to benefit the vision encoder, we demonstrate that DPO yields stronger visual representations than SFT.
>    2. SOTA encoders like SigLIP2 are often assumed to be nearly optimal due to their training on massive data and computation; however, our study shows that even a simple DPO step leaves substantial room for improvement within MLLMs.
>
> We believe this novel analysis constitutes a valuable contribution to the MLLM community. We appreciate the suggestion about new RL methods and plan to explore this in future research.
>
> \* Daniel Bolya et al. Perception encoder: The best visual embeddings are not at the output of the network, arXiv, 2025.
>
>
>
>
>
> ---
>
>
> > ### W2: RL variants: It’s unclear how broadly they hold across other RL algorithms or different post-training recipes.
> > ### Q1: Do the main findings persist with other RL approaches or alternative pairwise objectives beyond DPO?
>
> Thank you for raising an important question. We agree that applying our findings to other RL approaches could strengthen and generalize our claim. **We are currently conducting additional experiments with PPO, GRPO, and alternatives to DPO.** Due to the computational demands, results are expected within the next week. In addition, we would like to clarify our considerations on the choice of RL methods for analysis:
>
> 1. We select DPO as a primary RL method for the following reasons:
>    1. DPO does not rely on an external reward model, which minimizes confounding factors when comparing with SFT.
>    2.  DPO allows controlled comparison with SFT on the same data format, $ (I_{t}, q_{t}, y_{t}^{c}, y_{t}^{r}) $ (Line 142).
> 2. PPO requires a reward model trained on a different data $ ({I_{t}, q_{t}, y_{t}^{1}, r_{t}^{1}, y_{t}^{2}, r_{t}^{2}, \ldots}) $ (where $r_{t}$ represents a reward value or a rank), as well as additional data for training the policy model. Thus, it is challenging to build a data fair setting for comparison between PPO and SFT.
> 3. GRPO relies on 'verifiable' data, such as math or coding problems for LLMs, or object counting and bounding-box annotations for MLLMs. As SFT and DPO data do not contain such verifiable signals, a fair comparison setting for GRPO and SFT presents additional challenges from those of PPO.
>
> Despite these limitations, we are making our best effort to extend our analysis to RL beyond DPO. We appreciate the reviewer’s patience and look forward to sharing them soon.

---

> ### Author Response · Authors · 2025-11-22
> **Response to Reviewer ncjS (2/2)**
>
> > ### Q2: How does the end MLLM perform under text-only benchmarks, such as HellaSwag? Do you observe any depreciation?
>
> We appreciate that the reviewer raises an important point. Investigating text-only performance provides valuable insights even for multimodal models. **Through our experiments, we observe that DPO does not degrade text-only capabilities while enhancing vision representations.** Our evaluation includes HellaSwag (Zellers et al., 2019), as suggested, and additionally on MMLU (Hendrycks et al., 2021) and GSM8K (Cobbe et al., 2021), widely used benchmarks for LLM evaluation.
>
> The experimental results are presented in the table below.
>
> | Model | MMLU |  |  |  |  | HellaSwag | GSM8K |
> | - | - | - | - | - | - | - | - |
> |  | Overall | Humanities | Other | SocialSci | STEM | Acc | Acc |
> | MLLM-0.5B-SFT | 30.68 | 31.18 | 33.89 | 31.69 | 25.78 | **45.8** | 33.4 |
> | MLLM-0.5B-DPO | **38.05** | **37.28** | **43.93** | **40.27** | **31.24** | 45.28 | **34.7** |
> | $\Delta$ (DPO-SFT) | +7.37 | +6.1 | +10.04 | +8.58 | +5.46 | -0.52 | +1.3 |
> | MLLM-1.5B-SFT | 34.65 | 32.84 | 39.04 | 37.99 | 29.75 | 55.38 | **57.7** |
> | MLLM-1.5B-DPO | **46.75** | **42.7** | **51.56** | **54.53** | **40.47** | **56.94** | 57.3 |
> | $\Delta$ (DPO-SFT) | +12.1 | +9.86 | +12.52 | +16.54 | +10.72 | +1.56 | -0.4 |
>
> Results show that DPO-trained models achieve superior performance on MMLU compared to SFT. For HellaSwag and GSM8K, performance remains comparable, aligning with observations in Knowledge VQA. We speculate that this is due to data distribution, MPO data, used for multimodal training. MPO data includes general, science, and document VQA data, which overlaps with the target of MMLU. Otherwise, MPO’s math data is highly focused on geometry that GSM8k doesn’t cover. Also, MPO doesn’t have data related to HellaSwag tasks. So, we conjecture that the data overlap causes different results between MMLU and others. Still, an in-depth study is needed, which we consider future work, as discussed in Section E.1.
>
> \* Rowan Zellers et al. HellaSwag: Can a Machine Really Finish Your Sentence?, ACL, 2019.
>
> \* Karl Cobbe et al. GSM8K: Training Verifiers to Solve Math Word Problems, arXiv, 2021
>
> \* Dan Hendrycks et al. MMLU: Measuring Massive Multitask Language Understanding, ICLR, 2021.
>
>
>
>
>
>
> ---
>
> > ### Q3: For models leveraging practical pipelines such as SFT→DPO [A, B], where does your work stand
>
> The reviewer raises an important point regarding the positioning of our work, and we agree that addressing it strengthens the paper. In short, **we share the same stance as (Yu et al., 2024; Sun et al., 2024), who demonstrate the benefits of the SFT→RL pipeline, and we aim to further support their conclusions.**
>
> 1. Previous works show that the pipeline of `SFT (pretraining) → RL (post-training)` yields superior performance compared to `SFT-only baselines`. We reinforce this consensus by showing that `SFT (pretraining) → RL (post-training)` outperforms `SFT (pretraining) → SFT (post-training)`.
> 2. Distinct from prior work, our contributions are: (1) a controlled comparison with the same computational budget and amount of post-training data (Line 140), and (2) a novel analysis of visual representations within the MLLM.
>
> Until recently, SFT remained the dominant approach in MLLM training (Tong et al., 2024; Fan et al., 2025; Han et al., 2025), even after the insights provided by (Yu et al., 2024; Sun et al., 2024). Our work reinforces their findings and advocates for broader consideration of RL methods. To ensure clarity, we have added a discussion of this point in Section E.2.
>
> \* Tianyu Yu et al. RLAIF-V: Aligning MLLMs through Open-Source AI Feedback for Super GPT-4V Trustworthiness, CVPR, 2024.
>
> \* Zhiqing Sun et al. LLaVA-RLHF: Aligning Large Multimodal Models with Factually Augmented RLHF, ACL, 2024.
>
> \* Peter Tong et al. Cambrian-1: A Fully Open, Vision-Centric Exploration of Multimodal LLMs, NeurIPS, 2024.
>
> \* David Fan et al. Scaling Language-Free Visual Representation Learning, ICCV, 2025.
>
> \* Junlin Han et al. Learning to See Before Seeing: Demystifying LLM Visual Priors from Language Pre-training, arXiv, 2025.

---

> > ### Comment · Reviewer_ncjS · 2025-11-25
> > **Initial Response to the Rebuttal**
> >
> > I would like to thank the authors for their efforts and their rebuttal. My primary concerns pre-rebuttal phase were three-folds:
> >
> > - Limited methodological novelty
> > - Limited analysis to a single RL variant
> > - Performance analysis for text-only benchmarks
> > - Broader stance of the work on approaches utilizing SFT _and_ DPO instead of SFT _or_ DPO.
> >
> > Following the response by the authors, the first limitation above still persists to an extent though I do not believe that it is major enough to make this work a reject. I particularly feel this way as I believe that the analysis and other findings of the work are very interesting and could be so for the broader community.
> >
> > Regarding the second weakness, the authors have stated that they are still running experiments. I fully understand this and thank them for trying to complete these experiments in such short notice.
> >
> > Finally, regarding the third and fourth weaknesses, the authors have provided compelling quantitative evidence that DPO does not underperform against SFT even under text-only benchmarks and provided additional discussions around where their work stands with respect to the newer works utilizing SFT and DPO sequentially. I believe that the results presented by the authors on text-heavy benchmarks is very interesting and I acknowledge the comments of the authors on the fourth weakness. Accordingly, my concerns on these two weaknesses are fully resolved.
> >
> > As suggested by the authors, I will be waiting for the results with different RL variants and will revisit my review then. For the time being, I am maintaining my rating but I am open to increasing it.

---

> ### Author Response · Authors · 2025-12-02
> **Follow-up Response to Reviewer ncjS (1/2)**
>
> We appreciate that our responses addressed the reviewer’s concerns, and we are grateful for their openness to increasing the score.
>
> We had committed to providing additional experimental results, and we are happy to share them now. The Reviewer raised the concern about whether our findings hold when comparing other RL algorithms with SFT. Accordingly, **we have completed additional experiments with PPO, GRPO, and a DPO variant, and we observe that our findings generalize across them.** We noted the following considerations as the rationale for choosing DPO as our primary RL method:
>
> - DPO enables direct comparison with SFT using identical data formats and does not require an external reward model.
> - PPO relies on external reward models and uses data formats that are not aligned with those of SFT.
> - GRPO depends on verifiable data types, making it difficult to ensure comparable training setups.
>
> Despite these limitations, we are making our best effort to extend our analysis to other RL algorithms, and we present the results below.
>
>
>
> **1. GRPO vs. SFT**
>
> We first survey publicly available implementations of the GRPO, as summarized below.
>
> |Code|Main task|GRPO data|SFT data|GRPO code|SFT code|Vision encoder|
> |-|-|-|-|-|-|-|
> |R1-V (2025)|object counting / geometry math|o|x|o|o|update|
> |EasyR1 (2025)|geometry math|o|x|o|x|update|
> |SimpleVLA-RL (2025)|robot action planning|o|x|o|x|freeze|
> |VLM-R1 (2025)|bounding-box annotation|**o**|**o**|**o**|**o**|**update**|
>
> Among these options, we select the codebase from VLM-RL for our GRPO vs. SFT experiments, as it offers a validated training strategy for proper GRPO implementation. Here, QwenVL-2.5-3B is used as the base model, and we perform 1,500 post-training steps using different objectives while adopting the remaining training parameters directly from the repository. We will describe the experimental setup in full detail in the manuscript.
>
> Below, we present results on **16 MLLM benchmarks** for MLLMs trained with different objectives. We also evaluate the vision encoder updated within MLLMs on vision-only benchmarks.
>
> |MLLM eval. (Section 3)|Post-train|Avg (All)|General|OCR&Chart|Vision|Knowledge|
> |-|-|-|-|-|-|-|
> |QwenVL-2.5-3B|SFT|62.8|69.4|69|58|55|
> |QwenVL-2.5-3B|GRPO|**65.9**|**72.1**|**73.3**|**61.4**|**56.7**|
> ||$\Delta$|+3.1|+2.7|+4.3|+3.4|+1.7|
>
> |Vision-only eval. (Section4)|Post-train|ImageNet|Segmentation|
> |-|-|-|-|
> |QwenVL-2.5-3B|SFT|52.01|33.71|
> |QwenVL-2.5-3B|GRPO|**53.94**|**35.54**|
> ||$\Delta$|+1.93|+1.8|
>
> The results show that GRPO consistently outperforms SFT on both evaluations. In particular, RL (i.e., GRPO) strengthens MLLMs more on vision-intensive tasks than SFT (Finding 2 in Section 3). MLLM training reshapes the vision encoder, and RL yields more effective visual representations (Findings 3 & 4 in Section 4). These observations confirm that our findings generalize beyond DPO to GRPO.
>
>
>
> **2. PPO vs. SFT**
>
> We next turn to the comparison between PPO and SFT and provide a survey of available codebases below.
>
> |Code|Data type|PPO data|SFT data|PPO code|SFT code|Vision encoder|
> |-|-|-|-|-|-|-|
> |RL4VLM (2024)|card game / robot action planning|o|o|o|x|freeze|
> |SFT Memo., RL Generalize (2025)|card game / navigation|o|x|o|o|update|
> |LLaVA-RLHF (2023)|conversation|**o**|**o**|**o**|**o**|freeze|
>
> Based on these conditions, we choose to adopt the LLaVA-RLHF implementation. For PPO, we utilize their publicly released model, and for SFT, we post-train a LLaVA-1.0-7B for one epoch using their SFT data and the original LLaVA framework (Liu et al. 2023). Further details will be described in our paper. Although the vision encoder is frozen in this setup, which prevents isolated evaluation of visual representations, we conduct this comparison to examine the influence of different objectives on MLLM performance. The results are presented as follows:
>
> |MLLM eval. (Section 3)|Post-train|Avg  (All)|General|OCR&Chart|Vision|Knowledge|
> |-|-|-|-|-|-|-|
> |LLaVA-1.0-7B|SFT|33.1|43.6|26.8|26.7|35.2|
> |LLaVA-1.0-7B|PPO|34.6|44|30|28|36.4|
> ||$\Delta$|+1.5|+0.4|+3.2|+1.3|+1.2|
>
> We observe that the PPO-trained MLLM outperforms its SFT counterpart on 16 MLLM benchmarks, with a notable gain on OCR&Chart tasks (Finding 2). We note that LLaVA-1.0-13B is used as the reward model in this codebase, which introduces confounding factors beyond the training objectives alone. Our DPO vs. SFT comparison mitigates this issue and provides a clearer perspective on RL vs. SFT, consistent with prior works in Table A.

---

> ### Author Response · Authors · 2025-12-02
> **Follow-up Response to Reviewer ncjS (2/2)**
>
> **3. MPO vs. DPO vs. SFT**
>
> Finally, we extend our comparison by including MPO (Wang et al. 2024) as a variant of DPO. Mixed preference optimization (i.e., MPO) combines the objectives of DPO, SFT, and a binary preference classification loss. We integrate their implementation into our codebase and run the experiments. The tables below report MLLM and vision-only evaluation results under different MLLM training algorithms.
>
> |MLLM eval. (Section 3)|Post-train|Avg|General|OCR&Chart|Vision|Knowledge|
> |-|-|-|-|-|-|-|
> |Qwen2.5-1.5B + SigLIP2-L/16|SFT|59.4|68.0|60.9|56.4|52.4|
> |Qwen2.5-1.5B + SigLIP2-L/16|DPO|61.0|70.0|**62.6**|58.4|53.0|
> |Qwen2.5-1.5B + SigLIP2-L/16|MPO|**61.5**|**70.2**|62.2|**59.6**|**54.0**|
>
> MPO achieves the highest average score in the MLLM evaluation, outperforming SFT, though its gain over DPO remains modest. On vision‑only benchmarks, MPO matches DPO, suggesting that its impact on visual representation learning is limited. Moreover, we present additional results below, following the setup in Section 5, where visual representations are evaluated within MLLMs.
>
> |Vision-only eval. (Section4)|Post-train|ImageNet|Segmentation|
> |-|-|-|-|
> |Qwen2.5-1.5B + SigLIP2-L/16|SFT|48.53|30.18|
> |Qwen2.5-1.5B + SigLIP2-L/16|DPO|**50.51**|30.89|
> |Qwen2.5-1.5B + SigLIP2-L/16|MPO|50.18|**31.07**|
>
> The results demonstrate that DPO and MPO both improve visual representation quality over SFT, with MPO showing slight gains in vision-centric and knowledge tasks.
>
> |Vision eval. in MLLMs (Section 5)|Avg (All)|General|OCR&Chart|Vision-Cent.|Knowledge|
> |-|-|-|-|-|-|
> |MLLM = Qwen1.5-1.5B +||||||
> |+ Original SigLIP2-So/16|52.4|66.2|46.6|50.6|46.1|
> |+ Stage2-SFT|54.6|66.9|52.2|51.7|47.7|
> |+ Stage2-DPO (i.e., PIVOT)|**55.6**|**68.1**|**53.9**|52.4|48.1|
> |+ Stage2-MPO|**55.6**|67.9|53.8|**52.6**|**48.3**|
>
>
>
> **4. Summary**
>
> Our additional experiments with GRPO, PPO, and MPO demonstrate that RL-based training improves MLLM performance and visual representations beyond SFT, confirming that our findings hold across other RL algorithms. As noted in Section E.4, we plan to develop new RL algorithms for stronger visual representations, which we believe will help MLLMs better “see.”
>
>
>
> \* Liang Chen et al. R1-V: Reinforcing Super Generalization Ability in Vision-Language Models with Less Than \$3. GitHub repository, 2025.
>
> \* Yaowei Zheng et al. EasyR1: An Efficient, Scalable, Multi-Modality RL Training Framework. In GitHub repository, 2025.
>
> \* Haozhan Li et al. SimpleVLA-RL: Scaling VLA Training via Reinforcement Learning. In arXiv, 2025.
>
> \* Haozhan Shen et al. VLM-R1: A Stable and Generalizable R1-style Large Vision-Language Model. In arXiv, 2025.
>
> \* Yuexiang Zhai et al. RL4VLM: Fine-Tuning Large Vision-Language Models as Decision-Making Agents via Reinforcement Learning. In NeurIPS, 2025.
>
> \* Tianzhe Chu et al. SFT Memorizes, RL Generalizes: A Comparative Study of Foundation Model Post-training. In ICML, 2025.
>
> \* Zhiqing Sun et al. LLaVA-RLHF: Aligning Large Multimodal Models with Factually Augmented RLHF. In arXiv, 2023.
>
> \* Weiyun Wang et al. MPO: Enhancing the Reasoning Ability of Multimodal Large Language Models via Mixed Preference Optimization. In arXiv, 2024.
>
>
>
>
> The full results across the 16 benchmarks are provided below.
>
> |Base model|post-train|General||||OCR&Chart||||Vision-Cent.||||Knowledge||||
> |-|-|-|-|-|-|-|-|-|-|-|-|-|-|-|-|-|-|
> |||mme|mmb|seed|gqa|scienceqa|mmmu|mathvista|ai2d|chartqa|ocrbench|textvqa|docvqa|mmvp|realworldqa|cv-2d|cv-3d|
> |QwenVL-2.5-3B|SFT|1536.7|73.4|71.2|56.0|78.0|44.0|21.7|76.5|62.9|64.0|62.5|86.7|44.0|50.1|67.4|70.4|
> |QwenVL-2.5-3B|**GRPO**|1575.8|77.1|73.3|59.2|79.3|46.4|22.6|78.5|70.4|67.7|66.8|88.3|43.0|58.2|70.6|74.0|
> |||||||||||||||||||
> |LLaVA-1.0-7B|SFT|1201.1|39.3|42.5|32.5|64.8|27.9|3.1|45.0|7.1|21.6|34.8|43.8|10.9|31.3|25.0|39.5|
> |LLaVA-1.0-7B|**PPO**|1230.6|39.5|43.0|32.2|64.7|28.1|4.4|48.4|15.7|24.8|34.8|44.8|15.2|33.5|25.0|38.4|
> |||||||||||||||||||
> |Qwen2.5-1.5B + SigLIP2-L/16|SFT|1420.5|70.9|71.7|58.5|86.0|40.2|12.1|71.1|62.4|58.9|64.5|58.0|44.6|58.2|63.6|59.3|
> |Qwen2.5-1.5B + SigLIP2-L/16|DPO|1485.1|72.4|72.9|60.3|86.2|40.8|14.0|71.1|65.2|61.4|64.1|59.5|51.3|58.8|63.2|60.2|
> |Qwen2.5-1.5B + SigLIP2-L/16|**MPO**|1496.7|72.3|72.4|61.1|87.0|41.8|15.3|72.0|65.2|60.4|64.1|59.1|53.3|59.6|63.3|62.2|
>
>
>
>
> |Vision eval. in MLLMs (Section 5)|General||||OCR&Chart||||Vision-Cent.||||Knowledge||||
> |-|-|-|-|-|-|-|-|-|-|-|-|-|-|-|-|-|
> |MLLM = Qwen1.5-1.5B +|mme|mmb|seed|gqa|chartqa|ocrbench|textvqa|docVQA|mmvp|realworldqa|cv-2d|cv-3d|scienceqa|mmmu|mathvista|ai2d|
> |+ Original SigLIP2-So/16|1403.0|65.8|68.9|59.8|43.5|39.3|54.0|49.5|36.0|55.4|58.2|52.8|72.9|38.2|8.8|64.4|
> |+ Stage2-SFT|1417.0|65.9|70.4|60.5|54.4|45.6|57.8|51|36.3|54.8|60.0|55.8|73.3|40.9|11.6|64.9|
> |+ Stage2-DPO (i.e., PIVOT)|1427.5|67.8|71.8|61.4|57.0|48.3|58.5|51.7|38.3|56.2|59.1|56.2|73.9|41.5|12.3|64.7|
> |+ Stage2-MPO|1438.9|68.3|71|60.3|56.6|49.1|59|50.5|39|55.2|60.5|55.6|72.3|41.4|13.8|65.5|

---

### Author Response · Authors · 2025-12-02
**Summary of Rebuttal Discussion**

Dear ACs and Reviewers,

We thank all the Reviewers for their time and thoughtful comments. Our discussion can be summarized as follows:

| Reviewer | Initial Score | Update             | Notes (Quotes from discussions)                              |
| -------- | ------------- | ------------------ | ------------------------------------------------------------ |
| ncjS     | 6             | Open to increasing | Three of four concerns resolved. **"Open to increasing the rating"** after other RL results are provided |
| U9oP     | 4             | 8                  | All concerns fully addressed. **"Happy to raise my score to 8 with strong accept"** |
| FvAp     | 8             | -                  | -                                                            |
| uppE     | 8             | 8                  | Satisfied with additional experiments. "**Maintain a positive rating**" |



We are particularly grateful for the acknowledgment of the strengths of our work, including:

- Our paper is *well written* (FvAp) and has an *enjoyable storyline* (ncjS).
- Our analysis covers *under-explored* (U9oP) and *novel* areas (uppE).
- Our analysis is *clearly motivated* (ncjS), *systematical* (U9oP), and *thorough* (FvAp).
- Our experiment is *extensive* (ncjS, U9oP, and uppE).
- Our presented training regime is *practical* (U9oP) and *strong* (ncjS).



We have carefully considered the Reviewers’ feedback on the weaknesses and provided our responses as follows.

**Clarifications & discussions**

- We clarified that our contribution lies in a novel analysis of RL vs. SFT that reveals overlooked effects on vision encoders. (ncjS-W1)
- We detailed that our work aligns with prior works favoring SFT pretraining followed by RL post-training, rather than SFT for both stages. (ncjS-Q3)
- We discussed the inherent challenges of fully fair DPO vs. SFT comparisons, where DPO may seem naturally advantaged. But, we clarified that our analysis goes beyond the simple “DPO is better” intuition. (uppE-MinorW1-a)
- We clarified that PIVOT, while simple, provides a valuable insight that DPO training can benefit not only LLMs but also vision encoders. (uppE-MinorW2)
- We moved Figure D from the appendix to the main paper. (uppE-Q3)

**Additional experiments**

- We validated that our findings generalize beyond DPO vs. SFT, extending to **other RL algorithms** such as GRPO, PPO, and MPO. (ncjS-W2 and U9oP-W3)
- We showed that DPO-trained MLLMs also outperform SFT on text-only benchmarks like HellaSwag and MMLU. (ncjS-Q2)
- We showed that DPO post-training provides clear improvements over pre-training alone for visual representation learning. (U9oP-W1+W5)
- We found that DPO is beneficial to MLLMs’ visual understanding, yielding better performance on hallucination, captioning, and robustness/real-world benchmarks. (U9oP-W2 and FvAp-W1)
- We evaluated the reviewer’s suggested distillation method and confirmed that it is compatible with our PIVOT training regime. (U9oP-W4)
- We tested additional MLLMs, including BLIP-2 and SPHINX, and observed that our PIVOT model shows strong performance compared to them. (FvAp-Q2)
- We showed that constructing SFT-friendly training data narrows DPO’s advantage, yet DPO remains competitive with SFT. (uppE-MinorW1-b)
- We conducted comparisons with larger training data sizes and observed that SFT still does not surpass DPO. (uppE-Q1)



We hope that our clarifications and additional experimental results address all of the concerns. We again appreciate the constructive feedback and the opportunity to improve our paper.

---

### Meta-Review · Area_Chair_U8Tq · 2025-12-09

**Summary:**

This work aims to understand the effect of reinforcement learning versus supervised fine-tuning post-training on the visual encoders of large vision-language models. Accordingly, the work shows that, RL (via DPO) tends to improve vision-centric MLLM performance and yields stronger, more localized visual features than SFT. Following from this intuition, the work then proposes the upgrade the visual encoder quality by first training it with DPO in an MLLM post-training setting. Most reviewers are positive to this submission, and the authors' rebuttal seems strong. I read the paper also, and believe the submission is above borderline acceptance.

**Reviewer Concerns:**

Most reviewers are positive to this submission, and the authors' rebuttal seems strong.

**Reviewer Scores:**

Most reviewers are positive to this submission, and the authors' rebuttal seems strong.

---

### Decision · Program_Chairs · 2026-01-26

Accept (Poster)